# SHARP RESULTS FOR NIEP AND NMF

## ABSTRACT

The orthodox Non-negative Inverse Eigenvalue Problem (oNIEP) has challenged mathematicians for over 70 years. Motivated by applications in non-negative matrix factorization (NMF) and network modeling, we consider an NIEP as follows. Consider a $K \times K$ diagonal matrix $J_{K,m} = \text{diag}(1 + a_{K,m}, 1, \ldots, 1, -1, \ldots, -1)$, where exactly $m$ entries are $-1$ and $a_{K,m} = \max\{0, (2m - K)\}$. We wish to determine for which $(K, m)$, there is a $K \times K$ orthogonal matrix $Q$ such that $Q J_{K,m} Q'$ is doubly stochastic. Using several approaches (especially a combined Haar and Discrete Fourier Transform (DFT) approach) we developed, we show that in most of the cases, the NIEP is solvable. We show that these results are sharp. Also, since these are construction approaches, they automatically provide an explicit way for computing matrix $Q$. As a result, these approaches give rise to both a computable NMF algorithm and sharp results for NMF. We also discuss the implication of our results for social network modeling.

## 1 INTRODUCTION

The Non-negative Inverse Eigenvalue Problem (NIEP) goes back to a question posed by Kolmogorov in 1937 Kolmogorov (1937): given a complex number $z$, when is $z$ an eigenvalue of a non-negative matrix? Suleimanova Suleimanov (1949) extended Kolmogorov's question in 1949 to what is now called the orthodox NIEP (oNIEP): find necessary and sufficient conditions such that the list of $K$ complex numbers $\sigma = (\lambda_1, \lambda_2, \ldots, \lambda_K)$ is the spectrum of a $K \times K$ non-negative matrix $A$. It turns out that Kolmogorov's question is relatively easy to answer, but the oNIEP is much harder and has challenged mathematicians for 70+ years. For this reason, many sub-problems of oNIEP has been proposed, among them are the Real NIEP (assume $\lambda_k$ are real-valued), the Symmetric Real NIEP (assume $\lambda_k$ are real-valued and $A$ is symmetric), and the Double Stochastic Symmetric NIEP (assume $\lambda_k$ are real-valued and $A$ is doubly stochastic). Existing approaches to NIEP have been largely focused on construction approaches (e.g., see (Fiedler, 1974; G.W.Soules, 1983)). Unfortunately, both oNIEP and these sub-problems are extremely hard ones, and all remain unsolved for $K \geq 5$. See (Egleston et al., 2004) and also (Johnsona et al., 2018) for a nice survey.

Motivated by applications in non-negative matrix factorization (NMF) and network modeling (see Sections 1.1 and 4), we consider a special NIEP in this paper. We need the following definition.

**Definition 1.1** *We call a matrix non-negative if all its entries are non-negative. We call a non-negative matrix double stochastic if all of its rows and columns sum to the same number.*

Conventionally, in a doubly stochastic matrix, all rows/columns sum to 1; our definition is slightly different. Fix integers $(K, m)$ such that $K \geq 2$ and $1 \leq m \leq K - 1$. Let $a_{K,m} = \max\{0, (2m - K)\}$ and let $J_{K,m} \in \mathbb{R}^{K,K}$ be the diagonal matrix $J_{K,m} = \text{diag}(1 + a_{K,m}, 1, \ldots, 1, -1, \ldots, -1)$ (where exactly $m$ of them are $-1$). We are interested in a special case of the NIEP as follows:

> The NIEP: for which $(m, K)$, there is an easy-to-compute $K \times K$ orthogonal matrix $Q$ such that $Q J_{K,m} Q'$ is doubly stochastic (and so non-negative by definition). (1)

When such a $Q$ exists, we say Problem (1) is solvable for $(K, m)$. In this paper, we identify many pairs of $(K, m)$ where Problem (1) is solvable and $K$ be arbitrarily large. These results address a hard problem and shed interesting new light on the NIEP.

### 1.1 Motivation: NMF and social network modeling

The NIEP is motivated by an NMF problem as follows (where $K \ll n$ as in most applications).

> The NMF problem: Given an $n \times n$ non-negative matrix $\Omega$ with rank $K$, when can we write $\Omega = \Theta \Pi P \Pi' \Theta$, where $\Theta$ is an $n \times n$ positive diagonal matrix, $\Pi$ is an $n \times K$ matrix where each row is a weight vector, and $P$ is a $K \times K$ non-negative matrix? $\qquad$ (2)

The NMF is motived by recent interests on social network modeling. Especially, if the NMF problem is solvable, then we can rewrite a so-called rank-$K$ network model as a *Degree-Corrected Mixed-Membership (DCMM)* model (Jin et al., 2022+; Zhang et al., 2020). See Section L in the supplement and Jin (2022) for detailed discussion on how the NMF problem (2) is related to network modeling.

To see how NIEP in (1) is motivated by the NMF problem (2), we fix $a \geq 0$ and introduce a $K \times K$ matrix $J_{K,m,a} = \text{diag}(1 + a, 1, \ldots, 1, -1, \ldots, -1)$ (where $m$ entries are $-1$). Let $\lambda_k$ be the $k$-th largest (in absolute values) eigenvalue of $\Omega$ and let $\xi_k$ be the corresponding eigenvector. Extending (Jin, 2022), we can show that the NMF problem (2) is solvable if the Conditions (A)-(B) below hold:

Condition (A). $\quad \exists$ a $K \times K$ orthogonal matrix $Q$ such that $Q' J_{K,m,a} Q$ is doubly stochastic,

$$\text{Condition (B).} \quad \sum_{k=1}^{K-1} |\lambda_{k+1}| \cdot (\xi_{k+1}(i)/\xi_1(i))^2 \leq \frac{1}{K-1} \frac{|\lambda_1|}{(1+a)}, 1 \leq i \leq n. \qquad (3)$$

In fact, once Conditions (A)-(B) hold, we can use $Q$ and the first $K$ eigen-pairs of $\Omega$ to construct $(\Theta, \Pi, P)$ such that $\Omega = \Theta \Pi P \Pi' \Theta$ and $(\Theta, \Pi, P)$ are non-negative matrices as we desire to have in (2). See Algorithm 1 for details. Seeming, the parameter $a \geq 0$ plays a key role. For the special case of $m \leq K/2$, we can take $a$ as mall as $a = 0$. For all $(K, m)$ in this case, we can construct a $Q$ with an explicit formula (so it is easy-to-compute) such that Condition (A) holds. At the same time, Condition (B) is known as the sharpest as we can have (see (Jin, 2022) and the 551 page book on NMF (Shaked-Monderer & Berman, 2021)).

For the more challenging case of $m > K/2$, the problem remains largely unsolved. In fact, to use Algorithm 1, we must input an $a \geq 0$, and check Conditions (A) and (B) (both depend on $a$). Note that Condition (A) does not depend on $\Omega$, and for each $a \geq 0$, Condition (B) defines a class of $\Omega$. Therefore, in order to have a better algorithm, we wish to identify the smallest $a$ (denoted by $a_{K,m}^*$) such that Condition (A) is satisfied. If we can identify $a_{K,m}^*$, then we have two benefits.

- We can improve Algorithm 1 to a new algorithm, where there is no need to input an $a$ or to check Condition (A); all we need is to check if Condition (B) holds for $a = a_{K,m}^*$.
- The class of $\Omega$ defined by Condition (B) is as broad as possible.

These motivate the following problem.

> Optimal $a$: Fix $K$ and $1 \leq m \leq K - 1$. What is the smallest $a$ (denoted by $a_{K,m}^*$) such that there is a $K \times K$ orthogonal matrix $Q$ where $Q' J_{K,m,a} Q$ is doubly stochastic? $\qquad$ (4)

Recall that $a_{K,m} = \max\{0, 2m - K\}$. We have the following lemma (the proof is omitted).

**Lemma 1** *For Problem (4) to be solvable, we must have $a \geq a_{K,m}$. Therefore, $a_{K,m}^* \geq a_{K,m}$.*

By Lemma 1, the smallest $a$ we may consider is $a = a_{K,m}$. This raises the following question: for $a = a_{K,m}$, is there a $K \times K$ orthogonal matrix $Q$ such that $Q' J_{K,m,a} Q$ is doubly stochastic? Compared with (1), this is nothing but the NIEP stated earlier; see details therein.

In summary, fix a $(K, m)$ and suppose NIEP (1) is solvable for them. In this case, first, $a_{K,m}^* = a_{K,m}$, so we have completely solved Problem (4). Second, Condition (A) is automatically satisfied. Last, Condition (B) defines a class of $\Omega$ that is as broad as possible. Combining these gives an improved algorithm for NMF (e.g., Algorithm 2) and sharp results on NMF.

This explains how NIEP (1) is related to the NMF problem (2), and also why NIEP (1) is of interest.

**The matrix $Q$.** Note that to solve the NMF problem, we must solve the NIEP (1) with an *easy-to-compute* orthogonal matrix $Q$. This is exactly what we provide in this paper: in Section 2, for all $(K, m)$ where we show that the NIEP is solvable, we use a construction approach for the proof, so the matrix $Q$ is provided explicitly in our proof and is easy-to-compute.

---

**Algorithm 1** The NMF algorithm

---

**Input**: a number $a > 0$, an integer $K \geq 1$, and an $n \times n$ non-negative matrix with rank $K$.

- For $k = 1, 2, \ldots, K$, obtain the $k$-th eigen-pair $(\lambda_k, \xi_k)$ of $\Omega$.
- If Conditions (A)-(B) hold (note that both conditions depend on $a$; e.g., see (3)), obtain $U = [\xi_1, \xi_2, \ldots, \xi_K] \mathrm{diag}(|\lambda_1|/\sqrt{1 + a}, |\lambda_2|, \ldots, |\lambda_K|)Q$. Write $U = [u_1, u_2, \ldots, u_n]$ and factor $u_i$ as $\theta_i \pi_i$ where $\theta_i = \|u_1\|_1$. Claim the NMF problem (2) is solvable:

$$\Omega = \Theta \Pi P \Pi' \Theta, \quad \text{where } \Theta = \mathrm{diag}(\theta_1, \ldots, \theta_n), \Pi = [\pi_1, \ldots, \pi_n]', P = Q J_{K,m,a} Q'.$$

- Otherwise, conclude that it is unclear whether the NMF problem (2) is solvable or not.

**Output**: $(\Theta, \Pi, P)$ (if Condition (A)-(B) hold) or Unclear otherwise.

---

## 1.2 RESULTS AND CONTRIBUTIONS

We tackle Problem (1) using the construction approach (which is frequently used in the literature on NIEP). Our construction uses (a) a Haar basis approach, (b) a Discrete Fourier Transformation (DFT) approach, (c) a combined Haar-DFT approach, and (d) a Fiedler's approach. Compared with existing approaches in NIEP, all (a)-(d) are new. Note that (d) is inspired by well-known Fiedler's approach (Fiedler, 1974), but we need new tricks to make it useful in our setting).

In Section 2, first, we show in 9 different cases, (S1)-(S9), Problem (1) is solvable. For example, one such case is when $4 \mid K$, and we show Problem (1) is always solvable when $4 \mid K$. We also show that in two different cases, (N1)-(N2), Problem (1) is not solvable. For illustration, we present our results for all eligible $(K, m)$ with $K \leq 20$ in Table 1: in all 190 pairs of $(K, m)$ in this range, 169 of them are solvable, 9 of them are not solvable, and 12 of them remain unclear. In Section 3, to complement with our study in Section 2, we propose a novel optimization approach and further identify 3 of the 12 "unclear" cases as *solvable*. Therefore, only 9 out 190 cases remain unclear.

In Section 4, we discuss connections to NMF and network modeling, and propose Algorithm 2 as an improved version of Algorithm 1. In detail, we construct a quantity $\tilde{a}_{K,m}$ which equals to $a_{K,m}$ for the 9 solvable cases S1-S9 above, and is slightly larger than $a_{K,m}$ otherwise. We show that if

$$\sum_{k=1}^{K-1} |\lambda_{k+1}| [\xi_{k+1}(i)/\xi_1(i)]^2 \leq \lambda_1 / [(1 + \tilde{a}_{K,m})(K - 1)], \qquad 1 \leq i \leq n, \tag{5}$$

then there exist non-negative matrices $(\Theta, \Pi, P)$ as desired in (2) such that $\Omega = \Theta \Pi P \Pi' \Theta$. Compared with Condition (B), the above condition is nearly optimal in the sense that $\tilde{a}_{K,m}$ is nearly equal to $a^*_{K,m}$ (recall that $a^*_{K,m}$ denotes the optimal $a$; see (4)).

Our contributions are two fold. First, we obtained many interesting results on Problem (1). By far, for $K \geq 5$, the oNIEP Suleimanov (1949) and many of its sub-problems remain unsolved. Our problem is a special NIEP motivated by NMF and network modeling. Compared to oNIEP and many of its subproblems, our problem is still quite challenging, but fortunately, the study of which is more fruitful (e.g., our $K$ can be arbitrarily large). The main technical challenge is, we need an array of innovative ideas and tricks to construct orthogonal matrices $Q$ with desired properties. We tackle this by developing a Haar basis approach, a DFT approach, and a combined Haar-DFT approach. Compared with existing works on oNIEP (see Egleston et al. (2004); Johnsona et al. (2018) for surveys), our setting is quite different and our results and approaches are new.

Second, we propose a new algorithm (e.g., Algorithm 2). In this new algorithm, once condition (5) is satisfied, then we can have an NMF $\Omega = \Theta \Pi P \Pi' \Theta$ as desired. Our NMF results are sharp: when the NIEP is solvable for $(K, m)$, then $a_{K,m}$ solves Problem (4) and Condition (B) defines a class of $\Omega$ that is as broad as possible. Our work is related to Jin (2022), but the focus of Jin (2022) is on how to rotate an $n$-dimensional vector to the first quadrant of $\mathbb{R}^n$ and on the case of $m \leq K/2$, and our focus is on NIEP and the much more challenging case of $m \geq K/2$. As a result, our study is technically more demanding than Jin (2022), requiring many new ideas and tricks. Our work also sheds new light on network modeling. There are many different network models, and we must understand their advantages and disadvantages and pick the most suitable one; our findings are helpful in identifying the most suitable models in analyzing real networks.

**Notations**. In this paper, $c_{K,0} = K^{-1/2}\mathbf{1}_K = K^{-1/2}(1, 1, \ldots, 1)'$ (where " $'$ " is the transpose). For a vector $a \in \mathbb{R}^n$, $\|a\|$ and $\|a\|_\infty$ denote the $\ell^2$-norm and the $\ell^\infty$-norm, respectively, and $\mathrm{diag}(a)$ denotes the $n \times n$ diagonal matrix where the $i$-th diagonal entry is $a_i$, $1 \leq i \leq n$. When $\Omega$ is an $n \times n$ matrix, $\mathrm{diag}(\Omega)$ denotes the $n \times n$ diagonal matrix where the $i$-th entry is $\Omega(i, i)$, $1 \leq i \leq n$.

## 2 MAIN RESULTS

We tackle the NIEP with the construction approach and a frequently used idea is as follows.

**Definition 2.1** *Fix $K \geq 2$, let $c_{K,0} \in \mathbb{R}^K$ be the vector $K^{-1/2}(1, 1, \ldots, 1)'$.*

Suppose there is a $K \times K$ orthogonal matrix $Q$ such that $M = QJ_{K,m}Q'$ is doubly stochastic. Partition $Q = [q_1, Q_1, G]$, where $q_1, Q_1, G$ consisting 1, $(K - m - 1)$, and $m$ columns of $Q$, respectively. Since $M$ is doubly stochastic, we can always assume $q_1 = c_{K,0}$ (e.g., Lemma A.1 and especially the remark below it). By the definition of $J_{K,m}$ and direct calculations,

$$M = (2 + a_{K,m})c_{K,0}c_{K,0}' + 2Q_1Q_1' - I_K. \tag{6}$$

Therefore, all we need is to construct a matrix $Q \in \mathbb{R}^{K,K-m-1}$ such that ($H = Q_1Q_1'$ for short)

$$Q_1'c_{K,0} = 0, \; Q_1'Q_1 = I_{K-m-1}, \; H(i,j) \geq \begin{cases} (K - 2 - a_{K,m})/(2K), & i = j, \\ -(2 + a_{K,m})/(2K), & \text{otherwise.} \end{cases} \tag{7}$$

Especially, if $m \geq K/2$, then $a_{K,m} = 2m - K$ and $M$ is non-negative and traceless ($\mathrm{trace}(M) = \mathrm{trace}(J_{K,m}) = 0$). In this case, the last requirement of (7) can be strengthened into

$$H(i,j) = (K - m - 1)/K \text{ if } i = j \text{ and } H(i,j) \geq -(2 + 2m - K)/(2K) \text{ if } i \neq j. \tag{8}$$

When $(K - m - 1) \leq 1$ or $K \leq 4$ is small, the above problem is not hard to solve.

**Theorem 2** *Problem (1) is not solvable when $m = K - 2$ and $K \geq 5$ is an odd number.*

In this case, first, $(K - m - 1) = 1$ so $Q_1$ reduces to a unit-norm vector $q \in \mathbb{R}^K$ and so $H = qq'$. Second, $m \geq K/2$, so $H$ must satisfy (8). Combining these, each entry of $q$ is $\pm\sqrt{1/K}$, but since $K$ is odd, none of such $q$ is orthogonal to $\mathbf{1}_K$; compare this with the first requirement (7). This proves Theorem 2 (see the appendix for detailed proof). Similarly, we have the theorem below.

**Theorem 3** *Fix integers $(K, m)$ with $1 \leq m \leq K - 1$. Problem (1) is solvable in either of the following cases: (a) $m = K - 1$, (b) $m = K - 2$ and $K$ is an even number, (c) $K \leq 4$.*

When $m \leq (K - 3)$, $Q_1$ has at least two columns, and the study is much harder. This includes the case of $(K, m) = (7, 4)$; the theorem below is long and it takes a long time to figure out.

**Theorem 4** *Problem (1) is not solvable when $(K, m) = (7, 4)$.*

See Section B of the supplement for proofs for Theorem 2-4. We now discuss more general cases. In many of these cases, we follow the idea above and focus on how to construct $Q_1$. Since the columns of $Q_1$ is part of an orthogonal basis of $\mathbb{R}^K$, we borrow ideas from wavelets (e.g., Mallat (1999)), where we have many ideas for constructing an orthogonal basis (e.g., the Haar basis, the Discreet Fourier Transformation (DFT) basis).

### 2.1 THE HAAR BASIS APPROACH

**Definition 2.2** *Fix $K \geq 2$. For any $1 \leq j \leq K/2$, let $h_{K,j} \in \mathbb{R}^K$ be the vector where the $(2j-1)$-th row is $1/\sqrt{2}$, the $2j$-th row is $-1/\sqrt{2}$, and all other rows are $0$.*

Fixing $K \geq 2$ and $m \leq K/2$, let $H^* = [h_{K,1}, h_{K,2}, \ldots, h_{K,m}] \in \mathbb{R}^{K,m}$ and $U = [c_{K,0}, H^*]$. It is seen that $U'U = I_{m+1}$. Therefore, we can always find a matrix $G \in \mathbb{R}^{K,K-m-1}$ such that the matrix $Q = [c_{K,0}, G, H^*]$ is a $K \times K$ orthogonal matrix. Theorem 5 is proved in the supplement.

**Theorem 5** *Fix integers $(K, m)$ with $K \geq 2$ and $1 \leq m \leq K/2$. In this case, $a_{K,m} = 0$ and $J_{K,m} = \text{diag}(1, \ldots, 1, -1, \ldots, -1)$. For the matrix $Q$ above, the matrix $Q J_{K,m} Q'$ is non-negative doubly stochastic. Therefore, Problem (1) is solvable when $1 \leq m \leq K/2$ and $K \geq 2$.*

The first part of Theorem 5 overlaps with (Jin, 2022, Theorem 2.2), but not exactly the same. Theorem 5 is proved in the supplement. The above construction is inspired by the well-known Haar basis Mallat (1999). In fact, when $K$ is dyadic, (e.g., $K = 2^n$ for some integer $n$), $\{c_{k,0}, h_{K,1}, \ldots, h_{K,m}\}$ is part of a Haar basis Mallat (1999). Note that Theorem 5 is valid for any $K$ (dyadic or not).

When $K$ is dyadic, Problem (1) is always solvable. Since the case $m \leq K/2$ is covered above, we assume $(K/2) < m \leq K - 1$. In this case, let $H = [v_1, v_2, \ldots, v_K]$ be the $K \times K$ standard Haar basis matrix (Mallat, 1999), where $v_i$ are arranged in the descending order of $\|v_i\|_0$ (number of nonzero entries in $v_i$). Write $K = 2^s$ and $H = [v_1, U_0, U_1, \ldots, U_{s-1}]$, where for $0 \leq j \leq s - 1$, $U_j$ consists of $2^j$ consecutive columns of $H$. Write $m - K = \sum_{j=0}^{s-2} a_j 2^j$ where $a_j \in \{0, 1\}$ (such an expression exists and is unique). We construct a matrix $Q_1 \in \mathbb{R}^{K,m}$ as follows: we start with $Q_1 = [U_0, U_1, \ldots, U_{s-1}]$. Then for $j = 0, \ldots, s - 2$, we remove block $U_j$ in $H$ if and only if $a_j = 0$. Denote the resultant $K \times m$ matrix by $Q_1$. Theorem 6 is proved in the supplement

**Theorem 6** *Fix integers $(K, m)$ where $K$ is dyadic and $1 \leq m \leq K - 1$. For the matrix $Q_1$ above, there is a $K \times K$ orthogonal matrix $Q$ with the form of $Q = [c_{K,0}, G, Q_1]$ where $Q J_{K,m} Q'$ is non-negative doubly stochastic. Therefore, Problem (1) is always solvable when $K$ is dyadic.*

Theorem 6 is a special case of Theorem 10 below, but they use very different approaches to constructing $Q$. For this reason, both results are interesting and may lead to different future extensions.

## 2.2 THE DISCRETE FOURIER TRANSFORMATION (DFT) APPROACH

In this section, we focus on the case of $(m + 1) \geq 3K/4$. Let $Q = [q_1, Q_1, G]$ as in (6). We wish to construct a $Q_1$ in a way so that all diagonal entries of $Q_1 Q_1'$ are $(K - m - 1)/K$. Once we have such a $Q_1$, then $M = Q J_{K,m} Q'$ is doubly stochastic. To see the point, first, note that by (6) and direct calculations, all diagonal entries of $M$ are 0. Second, by Cauchy-Schwartz equality, when all diagonal entries of $QQ'$ are $(K - m - 1)/K$, then all off-diagonal entries of $Q_1 Q_1'$ are $\geq -(K - m - 1)/K$. It follows from (6) and direct calculations that all off-diagonal entries of $M$ are $\geq [3(m + 1) - 4K]/K$, which is $\geq 0$ since $(m + 1) \geq 3K/4$.

We now construct such a $Q_1$. Our idea is to use the well-known DFT orthogonal basis vectors (Mallat, 1999), which serve well for our purpose. Recall that $c_{K,0} = K^{-1/2}(1, 1, \ldots, 1)' \in \mathbb{R}^K$. Consider four cases: (a) $K$ is even and $m$ is odd, (b) $K$ is odd and $m$ is even, (c) both $K$ and $m$ are even, and (d) both $K$ and $m$ are odd. Note that $(K - m - 1)$ is even in (a)-(b) and is odd in (c)-(d).

**Definition 2.3** *Fix $K \geq 2$. When $K$ is even, let $s_{K,0} = \sqrt{1/K}(1, -1, 1, -1, \ldots, 1, -1)' \in \mathbb{R}^K$. Moreover, for any $1 \leq j \leq (K - 2)/2$, let*

$$c_{K,j} = \sqrt{2/K}\left(1, \cos(\frac{2j\pi}{K}), \cos(\frac{2 \cdot 2j\pi}{K}), \cos(\frac{3 \cdot 2j\pi}{K}), \ldots, \cos(\frac{(K-1) \cdot 2j\pi}{K})\right)',$$

$$s_{K,j} = \sqrt{2/K}\left(0, \sin(\frac{2j\pi}{K}), \sin(\frac{2 \cdot 2j\pi}{K}), \sin(\frac{3 \cdot 2j\pi}{K}), \ldots, \sin(\frac{(K-1) \cdot 2j\pi}{K})\right)'.$$

Now, let $n$ be the largest integer such that $2n \leq (K - m - 1)$ and let

$$Q_1 = \begin{cases} [c_{K,1}, s_{K,1}, \ldots, c_{K,n}, s_{K,n}] \in \mathbb{R}^{K,2n}, & \text{in Cases (a)-(b)}, \\ [s_{K,0}, c_{K,1}, s_{K,1}, \ldots, c_{K,n}, s_{K,n}] \in \mathbb{R}^{K,2n+1}, & \text{in Case (c)}. \end{cases}$$

For Case (d), let $Q_1^* = [q_0, q_1, q_2, \ldots, q_{2n}] \in \mathbb{R}^{N,2n+1}$, where for $1 \leq j \leq n$,

$$q_0 = \begin{bmatrix} \mathbf{1}_{(K+1)/2} \\ -\mathbf{1}_{(K-1)/2} \end{bmatrix}, \qquad q_{2j-1} = \begin{bmatrix} c_{(K+1)/2,j} \\ c_{(K-1)/2,j} \end{bmatrix}, \qquad q_{2j} = \begin{bmatrix} s_{(K+1)/2,j} \\ s_{(K-1)/2,j} \end{bmatrix}.$$

Let $c_0 = \sqrt{(K-1)/(K(K+1))}$, $d_0 = \sqrt{(K+1)/(K(K-1))}$, $c = \sqrt{\frac{1}{K}(\frac{K+1}{2} + \frac{1}{2n})}$ and $d = \sqrt{\frac{1}{K}(\frac{K-1}{2} - \frac{1}{2n})}$. We partition $Q_1^*$ as follows (left) and introduce a new matrix $Q_1$ (right) by

$$Q_1^* = \begin{bmatrix} \mathbf{1}_{(K+1)/2} & B_1 \\ -\mathbf{1}_{(K-1)/2} & B_2 \end{bmatrix}, \qquad Q_1 = \begin{bmatrix} c_0 \mathbf{1}_{(K+1)/2} & cB_1 \\ -d_0 \mathbf{1}_{(K-1)/2} & dB_2 \end{bmatrix}. \tag{9}$$

The following lemma is proved in the supplement.

**Lemma 7** *Fix integers $(K, m)$ such that $(3K/4) \leq m + 1 \leq K$. Suppose when $K$ is odd, $m \neq K - 2$. In all the four cases (a)-(d), the matrix $Q_1$ we constructed above satisfies $Q_1'Q_1 = I_{K-m-1}$ and all diagonal entries of $Q_1Q_1'$ are $(K - m - 1)/K$.*

Combining Lemma 7 with earlier arguments in this section gives the following theorem.

**Theorem 8** *Fix integers $(K, m)$ with $(3K/4) \leq m + 1 \leq K$ and $m \neq K - 2$ when $K$ is odd. For $Q_1 \in \mathbb{R}^{K,K-m-1}$ as above, there is a matrix $G \in \mathbb{R}^{K,m}$ such that (a) $Q = [c_{K,0}, Q_1, G]$ is a $K \times K$ orthogonal matrix, and (b) $QJ_{K,m}Q'$ is non-negative and doubly stochastic.*

Theorem 8 is proved in the supplement. Theorem 8 can be further improved with more delicate analysis. Consider Case (b) for example, where $1 \leq m \leq K - 1$, $K \geq 5$, $K$ is odd, and $m$ is even. Let $a_K = (1/2) + 1/(K \sin(4\pi/K))$, and let $1/2 < b_K < 3/4$ be the unique solution of the equation $x - 1/2 + \sin(3\pi x)/(K \sin(3\pi/K)) = 0$. Theorem 9 is proved in the supplement.

**Theorem 9** *Fix integers $(K, m)$ and $1 \leq m \leq K - 1$. Suppose $K$ is odd, $m$ is even, and $m/K \geq \max\{a_K, b_K\}$. Let $Q_1$ be constructed as above in Case (b). Then there is a matrix $G \in \mathbb{R}^{K,m}$ such that $Q = [c_{K,0}, Q_1, G]$ is a $K \times K$ orthogonal matrix and $QJ_{K,m}Q'$ is non-negative and doubly stochastic. Therefore, Problem (1) is solvable in this case.*

As $K \to \infty$, $\min\{a_K, b_K\} \to x_0$, where $x_0 \approx 0.58$ is the unique solution of $x - 1/2 + \frac{\sin(3\pi x)}{3\pi} = 0$. Compared with Theorem 8, the result in Theorem 9 is much stronger. Theorem 9 is for Case (b). For Case (d), improvements are possible, but we need more delicate analysis, so we omit it. For Case (a) and (c) (where $K$ is even), improvements are also possible. However, with a new approach in next section, we have much stronger results in these two cases; see Theorems 10-11 below. For these reasons, we skip further discussion along this line.

**Two algorithms (Section J of the supplement)**. Alternatively, we may directly check for which $(K, m)$, $QJ_{K,m}Q'$ is doubly stochastic for the $Q$ constructed above. In Section J of the supplement, we introduce two algorithms for Case (b) and (d), respectively. Se details therein. For Case (a) and (c), such algorithms are not necessary for the problem is already solved (almost) completely.

## 2.3 THE COMBINED HAAR-DFT APPROACH

The combined Haar-DFT approach gives much stronger results. Since the case $m \leq K/2$ is solved (e.g., Theorem 5), we assume $m > K/2$. We consider two cases (a) $4 \mid K$ and (b) $2 \mid K$ but $4 \nmid K$. Consider case (a). Let $n$ be the largest integer such that $2n \leq (m - K/2)$. Introduce

$$M_1 = \begin{cases} [s_{K/2,0}, M_1^*], & \text{if } m \text{ is odd,} \\ M_1^*, & \text{if } m \text{ is even,} \end{cases} \quad \text{where} \quad M_1^* = [c_{K/2,0}, s_{K/2,0}, \ldots, c_{K/2,n}, s_{K/2,n}].$$

We construct $Q_1 \in \mathbb{R}^{K,m-K/2}$ such that (a) the first two rows of $Q_1$ equal to each other, the next two rows of $Q_1$ equal to each other, and so on and so forth, and (b) row $1, 3, \ldots, (K-1)$ of $Q_1$ equal to row $1, 2, \ldots, K/2$ of $M_1^*$, respectively. Let $H_0 = [h_{K,1}, \ldots, h_{K,K/2}]$ and $Q_0 = [c_{K,0}, Q_1, H_0]$. It is seen that $Q_0 \in \mathbb{R}^{K,m+1}$ and $Q_0'Q_0 = I_p$. The following theorem is proved in the supplement (compared with Theorem 6, the result here is much stronger, but constructions are very different, so both of them may be valuable for future extensions).

**Theorem 10** *Fix integers $(K, m)$ where $K \geq 5$, $1 \leq m \leq K - 1$, and $4 \mid K$. There is a matrix $G \in \mathbb{R}^{K,K-m-1}$ such that $Q = [c_{K,0}, Q_1, G, H_0]$ is a $K \times K$ orthogonal matrix and $QJ_{K,m}Q'$ is non-negative and doubly stochastic. Therefore, Problem (1) is solvable when $4 \mid K$.*

Consider Case (b). In this case, $K/2$ is odd. When $m$ is odd, let $n = (m - K/2)/2$, and let $M_1^* = [c_{K/2,0}, s_{K/2,0}, \ldots, c_{K/2,n}, s_{K/2,n}]$. When $m$ is even and $m - K/2 \geq 3$, let $n = (m - (K/2) - 1)/2$, and define $M_1^*$ in a way similar to that of $Q_1^*$ in (9) by replacing $K$ there by $K/2$. Similarly, let $Q_1 \in \mathbb{R}^{K,m-K/2}$ be the matrix such that a) the first two rows of $Q_1$ equal to each other, the next two rows of $Q_1$ equal to each other, and so on and so forth, and (b) row $1, 3, \ldots, (K-1)$ of $Q_1$ equal to row $1, 2, \ldots, K/2$ of $M_1^*$, respectively. Finally, with the same $H_0$ as above, let $Q_0 = [c_{K,0}, Q_1, H_0]$. It is seen $Q_0 \in \mathbb{R}^{K,m+1}$ and $Q_0'Q_0 = I_{m+1}$. The following theorem is proved in the supplement.

**Theorem 11** *Fix integers $(K, m)$ where $K \geq 5$, $1 \leq m \leq K - 1$. Suppose $2 \mid K$ but $4 \nmid K$, and $m \leq (K/2) + 1$. There is a matrix $G \in \mathbb{R}^{K, K-m-1}$ such that $Q = [c_{K,0}, Q_1, G, H_0]$ is a $K \times K$ orthogonal matrix and $Q J_{K,m} Q'$ is non-negative and doubly stochastic. Therefore, Problem (1) is solvable when $K$ is even, $4 \nmid K$, and $m \neq (K/2) + 1$.*

Somewhat unexpected, the case of $m = (K/2) + 1$ is a challenging corner case. The reason why this is a corner case is that, when $m$ is even, our construction above requires $m - K/2 \geq 3$.

## 2.4 CONNECTIONS TO LITERATURE ON NIEP, ESPECIALLY FIEDLER'S APPROACH

Soules' approach G.W.Soules (1983) and and Fiedler's approach Fiedler (1974) are two well-known NIEP approaches. For our setting, Soules' approach does not produce any non-trivial result. Fiedler's approach inspires an approach below, which produces some non-trivial result that are not covered in the previous sections.

The main idea is to relate Problem (1) for a given $(K, m)$ to Problem 1) for $(K_1, m_1)$ and $(K_2, m_2)$, with $(K_1, K_2, m_1, m_2)$ to be determined. In detail, suppose $A \in \mathbb{R}^{K_1, K_1}$ is a traceless non-negative doubly stochastic matrix with spectrum $\sigma(A_1) = \{(2m_1 - K_1 + 1, 1, \ldots, 1, -1, \ldots, -1\}$ (where exactly $m_1$ of them are $-1$), and $A_2 \in \mathbb{R}^{K_2, K_2}$ is traceless non-negative doubly stochastic matrix with spectrum $\sigma(A_2) = \{(2m_2 - K_2 + 1, 1, \ldots, 1, -1, \ldots, -1\}$ (where exactly $m_2$ of them are $-1$). Suppose $(2m_1 - K_1) > 0$ and $2m_2 - K_2 > 0$, so the Perron roots of two matrices are $(2m_1 - K_1 + 1)$ and $(2m_2 - K_2 + 1)$, respectively, and the corresponding Perron eigenvectors are $u_1 = (K_1)^{-1/2} \mathbf{1}_{K_1}$ and $u_1 = (K_2)^{-1/2} \mathbf{1}_{K_2}$, respectively. Introduce

$$A = \begin{bmatrix} A_1 & \rho u_1 v_1' \\ \rho v_1 u_1' & A_2 \end{bmatrix}, \qquad \text{where } \rho = \sqrt{(2m_1 - K_1)(2m_2 - K_2)}. \tag{10}$$

**Theorem 12** *Fix integers $(K_1, m_1)$ and $(K_2, m_2)$ such that $K_1/2 < m_1 \leq K_1 - 1$, $K_2/2 < m_2 \leq K_2 - 1$, and $K_1/K_2 = m_1/m_2$. For $A_1$ and $A_2$ as above, the matrix $A$ defined as in (10) is doubly stochastic, and the spectrum of $A$ is $\sigma(A) = (2(m_1 + m_2) - (K_1 + K_2) + 1, 1, \ldots, 1, -1, \ldots, -1)$, where exactly $m_1 + m_2$ of them are $-1$. As a result, if Problem (1) is solvable for $(K, m)$ with $K/2 < m \leq K - 1$, then for any integer $N \geq 1$, Problem (1) is also solvable for $(NK, Nm)$.*

**Example 2**. It is not hard to see that Problem (1) is solvable for $(K, m) = (3, 2)$. By Theorem 12, Problem (1) is also solvable for $(K, m) = (6, 4)$, $(K, m) = (9, 6)$, etc.

## 2.5 A SUMMARY AND ESPECIALLY A PYTHON CODE FOR USING OUR THEOREMS

In summary, we have three cases: (N), (S), and (U), where Problem (1) is not solvable, solvable, and unclear so, respectively. Moreover, Case (N) includes all cases (N1) where $K \geq 5$ is odd and $m = K - 2$ and (N2) where $(K, m) = (7, 4)$. Case (S) includes 9 different cases: (S1). $K \leq 4$, (S2). $m = K - 1$, (S3). $m \leq K/2$, (S4). $4 \mid K$, (S5). $2 | K$ but $4 \nmid K$ and $m \neq (K/2) + 1$, (S6). $K$ is odd, $m$ is even, and $m/K \geq \max\{a_K, b_K\}$, (S7) $K$ is odd, $m$ is odd, and $(m + 1)/K \geq 3/4$ but $m \neq K - 2$, (S8). $K$ is odd, and $Q J_{K,m} Q'$ is doubly stochastic for the $Q$ constructed in cases (b) or (d) in Section 2.2 respectively (this can be checked by the algorithms in Section H of the supplement), and (S9). Problem (1) is solvable for $(K_1, m_1)$ where $K_1 = K/N$ and $m_1 = m/N$ for some integer $N \geq 2$. Case (U) includes two cases: (U1) the corner case where $K$ is even and $m = (K/2) + 1$, and (U2) all other cases not covered by Cases (N), (S), and (U1).

To use these results, we have written a python code, where if you input a $K$, then for each $m = 1, 2, \ldots, K - 1$, we output three possible outcomes: (N), (S), (U), where Problem (1) is not solvable, solvable, and unclear for $(K, m)$. Using the code, we have produced Table 1, where we present our results for all $(K, m)$ with $K \leq 20$. The results suggest when $K \leq 20$, Problem (1) is solvable for most cases. The code is available upon request (and will be made public after the review process).

## 3 AN OPTIMIZATION APPROACH TO THEORETICALLY UNSOLVED CASES

The results in Section 2 are based on construction approaches, where for each $(K, m)$ where Problem (1) is solvable, we have provided the the orthogonal matrix $Q$. See the proofs in detail. At the same

Table 1: Our results on Problem (1 ) for all possible $(K, m)$ with $1 \le m < K \le 20$. ■: outside the range of interest; ✓: the problem is solvable; ✗: the problem is not solvable; $\Delta$: solvable suggested by the numerical approach in Section 3, but not yet validated in theory; □: whether Problem (1) is solvable or not remains unclear.

| $m$ | 1 | 2 | 3 | 4 | 5 | 6 | 7 | 8 | 9 | 10 | 11 | 12 | 13 | 14 | 15 | 16 | 17 | 18 | 19 |
|---|---|---|---|---|---|---|---|---|---|---|---|---|---|---|---|---|---|---|---|
| $K = 2$ | ✓ | ■ | ■ | ■ | ■ | ■ | ■ | ■ | ■ | ■ | ■ | ■ | ■ | ■ | ■ | ■ | ■ | ■ | ■ |
| $K = 3$ | ✓ | ✓ | ■ | ■ | ■ | ■ | ■ | ■ | ■ | ■ | ■ | ■ | ■ | ■ | ■ | ■ | ■ | ■ | ■ |
| $K = 4$ | ✓ | ✓ | ✓ | ■ | ■ | ■ | ■ | ■ | ■ | ■ | ■ | ■ | ■ | ■ | ■ | ■ | ■ | ■ | ■ |
| $K = 5$ | ✓ | ✓ | ✗ | ✓ | ■ | ■ | ■ | ■ | ■ | ■ | ■ | ■ | ■ | ■ | ■ | ■ | ■ | ■ | ■ |
| $K = 6$ | ✓ | ✓ | ✓ | ✓ | ✓ | ■ | ■ | ■ | ■ | ■ | ■ | ■ | ■ | ■ | ■ | ■ | ■ | ■ | ■ |
| $K = 7$ | ✓ | ✓ | ✓ | ✗ | ✗ | ✓ | ■ | ■ | ■ | ■ | ■ | ■ | ■ | ■ | ■ | ■ | ■ | ■ | ■ |
| $K = 8$ | ✓ | ✓ | ✓ | ✓ | ✓ | ✓ | ✓ | ■ | ■ | ■ | ■ | ■ | ■ | ■ | ■ | ■ | ■ | ■ | ■ |
| $K = 9$ | ✓ | ✓ | ✓ | ✓ | □ | ✓ | ✗ | ✓ | ■ | ■ | ■ | ■ | ■ | ■ | ■ | ■ | ■ | ■ | ■ |
| $K = 10$ | ✓ | ✓ | ✓ | ✓ | ✓ | □ | ✓ | ✓ | ✓ | ■ | ■ | ■ | ■ | ■ | ■ | ■ | ■ | ■ | ■ |
| $K = 11$ | ✓ | ✓ | ✓ | ✓ | ✓ | □ | $\Delta$ | ✓ | ✗ | ✓ | ■ | ■ | ■ | ■ | ■ | ■ | ■ | ■ | ■ |
| $K = 12$ | ✓ | ✓ | ✓ | ✓ | ✓ | ✓ | ✓ | ✓ | ✓ | ✓ | ✓ | ■ | ■ | ■ | ■ | ■ | ■ | ■ | ■ |
| $K = 13$ | ✓ | ✓ | ✓ | ✓ | ✓ | ✓ | □ | ✓ | ✓ | ✓ | ✗ | ✓ | ■ | ■ | ■ | ■ | ■ | ■ | ■ |
| $K = 14$ | ✓ | ✓ | ✓ | ✓ | ✓ | ✓ | ✓ | □ | ✓ | ✓ | ✓ | ✓ | ✓ | ■ | ■ | ■ | ■ | ■ | ■ |
| $K = 15$ | ✓ | ✓ | ✓ | ✓ | ✓ | ✓ | ✓ | □ | $\Delta$ | ✓ | ✓ | ✓ | ✗ | ✓ | ■ | ■ | ■ | ■ | ■ |
| $K = 16$ | ✓ | ✓ | ✓ | ✓ | ✓ | ✓ | ✓ | ✓ | ✓ | ✓ | ✓ | ✓ | ✓ | ✓ | ■ | ■ | ■ | ■ | ■ |
| $K = 17$ | ✓ | ✓ | ✓ | ✓ | ✓ | ✓ | ✓ | ✓ | □ | ✓ | ✓ | ✓ | ✓ | ✓ | ✗ | ✓ | ■ | ■ | ■ |
| $K = 18$ | ✓ | ✓ | ✓ | ✓ | ✓ | ✓ | ✓ | ✓ | ✓ | □ | ✓ | ✓ | ✓ | ✓ | ✓ | ✓ | ✓ | ■ | ■ |
| $K = 19$ | ✓ | ✓ | ✓ | ✓ | ✓ | ✓ | ✓ | ✓ | ✓ | □ | $\Delta$ | ✓ | ✓ | ✓ | ✓ | ✓ | ✗ | ✓ | ■ |
| $K = 20$ | ✓ | ✓ | ✓ | ✓ | ✓ | ✓ | ✓ | ✓ | ✓ | ✓ | ✓ | ✓ | ✓ | ✓ | ✓ | ✓ | ✓ | ✓ | ✓ |

time, for some of the cases (e.g., see Table 1), it is unclear whether Problem (1) is solvable or not. To complement our study in Section 2, we propose an optimization approach with further insights.

Since Problem 1 is always solvable when $m \le K/2$, we assume $m > K/2$. Consider a $K \times K$ orthogonal matrix $Q = [c_{K,0}, Q_1, Q_2]$, where $Q_1 \in \mathbb{R}^{K, K-m-1}$ and $Q_2 \in \mathbb{R}^{K,m}$, and we recall $c_{K,0} = K^{-1/2}(1, 1, \ldots, 1)'$. By the definition, $Q J_{K,m} Q' = [(a_{K,m}+2)/K]\mathbf{1}_K \mathbf{1}_K' + 2 Q_1 Q_1' - I_K$, where $Q_1$ is known as a Stiefel manifold. Therefore, at a heart of Problem (1), it is a Stiefel manifold optimization problem, which is known to be a hard non-convex problem Li et al. (2020). We propose an approach which combines random Stiefel manifolds sampling with convex optimization, where the key is the following lemma (Lemma 13 is proved in the supplement).

**Lemma 13** *Fix $(m, K)$ such that $m > K/2$ and let $Q = [c_{K,0}, Q_1, Q_2]$ be a $K \times K$ orthogonal matrix as above. $Q J_{K,m} Q'$ is doubly stochastic if and only if there exists a matrix $M \in \mathbb{R}^{K,K}$ that satisfies the following constraints: (a). $\mathrm{rank}(M) = K - m - 1$, (b) $\lambda_{\max}(M) \le 1$ and $\mathbf{1}_K' M \mathbf{1}_K = 0$, and (c). $M_{ii} = (K-m-1)/K$ and $M_{ij} \ge -(2m-K+2)/(2K), 1 \le i \ne j \le K$.*

Note in all constraints in Lemma 13, all except the first one are convex. This motivates the following algorithm. (A). Repeat (A1)-(A2) below for $N$ times: (A1). Generate $M_0 \in \mathbb{R}^{K \times (K-m-1)}$ with iid $N(0,1)$ entries. Compute its SVD: $M_0 = U D V'$, and let $\widehat{M_0} = U U'$. (A2). Let $\mathcal{M}$ be the feasible region defined by (b)-(c) in Lemma 13. Solve the **convex** program: $\widehat{M} = \arg \max_{M \in \mathcal{M}, M \succeq 0} \mathrm{trace}(M \widehat{M_0})$, where $M \succeq 0$ constrains $M$ to be positive semidefinite. (B). Declare Problem (1) solvable if $\mathrm{rank}(\widehat{M}) = K - m - 1$ for **any** of the $N$ repetitions.

The main idea is to randomly sample a projection matrix of the desired rank, and find $\widehat{M} \in \mathcal{M}$ that maximally aligns with it. We apply the approach to all 12 unsolved cases in Table 1. The algorithm declares that Problem (1) is solvable for $(K, m) = (11, 7)$ and $(K, m) = (15, 9)$ with $N = 10^3$, and declares that Problem (1) is solvable for $(K, m) = (19, 15)$ with $N = 2 \times 10^5$. See Table 1 as well as the supplement (Section K) for details. Since the solution accuracy is limited by floating-point precision, whether these cases are theoretically solvable remains open. Assuming they are, all $(K, m)$ pairs such that $K = 2m - 3, m > 5$, and $K < 20$ are solvable. This suggests an interesting conjecture: Problem 1 is solvable for all $(K, m)$ where $K = (2m - 3)$ and $m > 5$.

Our problem involves the optimization over the Stiefel manifolds and is known as a hard non-convex problem. We approach it by running $N$ different random sampling, and for each of them we run a

convex optimization. The convex optimization part is relatively fast, but we may need a large $N$. How to develop a more effective approach to our problem is a very interesting future question.

---

**Algorithm 2** The NMF algorithm (as an improved version of Algorithm 1 in Section 1.1)

---

**Input**: An integer $K \geq 1$ and an $n \times n$ non-negative matrix $\Omega$ with rank $K$.

- For $k = 1, 2, \ldots, K$, obtain the $k$-th eigen-pair $(\lambda_k, \xi_k)$ of $\Omega$, and obtain $m$ (the number of strictly negative eigenvalues of $\Omega$). Following our constructions, set $a = \tilde{a}_{K,m}$ and obtain the $K \times K$ orthogonal matrix $Q$ such that $Q J_{K,m,a} Q'$ is doubly stochastic (see the proofs in the supplement for explicit formula for $Q$).

- If Condition (B*) holds, let $U = [\xi_1, \xi_2, \ldots, \xi_K] \text{diag}(|\lambda_1|/\sqrt{1 + \tilde{a}_{K,m}}, |\lambda_2|, \ldots, |\lambda_K|) Q$. Write $U = [u_1, u_2, \ldots, u_n]$ and factor $u_i$ as $\theta_i \pi_i$ where $\theta_i = \|u_1\|_1$. Claim that the NMF problem (2) is solvable and $\Omega = \Theta \Pi P \Pi' \Theta$, where $\Theta = \text{diag}(\theta_1, \theta_2, \ldots, \theta_n)$, $\Pi = [\pi_1, \pi_2, \ldots, \pi_n]'$ and $P = Q J_{K,m,a} Q'$ are non-negative matrices as in (2).

- Otherwise, conclude that it is unclear whether the NMF problem (2) is solvable or not.

**Output**: $(\Theta, \Pi, P)$ (if Condition (B*) holds) or Unclear otherwise.

---

## 4 APPLICATION TO NMF AND NETWORK MODELING

Recall that $a_{K,m} = \max\{0, 2m - K\}$ and $a^*_{K,m}$ is the smallest value of $a$ such that there is a $K \times K$ orthogonal matrix $Q$ such that $Q J_{K,m,a} Q'$ is doubly stochastic; see (4). Also, recall that we have 9 different solvable cases (S1)-(S9), 2 not-solvable cases (N1)-(N2), and some unclear cases, denoted by (U); see Section 2.5. We further divide $U$ as (U1) and (U2), where (U1) is the case where $m = (K/2) + 1$ and $2 \mid K$ but $4 \nmid K$, and (U2) contains all other cases in (U). Recall that $(\lambda_k, \xi_k)$ is the $k$-th eigen-pair of $\Omega$ (e.g., Section 1.1). Introduce

$$\tilde{a}_{K,m} = \begin{cases} a_{K,m}, & \text{Case (S1)-(S9)}, \\ K - \frac{4(K-1)}{K}, & \text{Case (N1) } (a_{K,m} = K - 4 \text{ in this case}), \\ 4\cos(\pi/7) - 2 \approx 1.6 & \text{Case (N2) } (a_{K,m} = 1 \text{ in this case}), \\ \frac{2(K+2)}{K-2}, & \text{Case (U1) } (a_{K,m} = 2 \text{ in this case}), \\ \frac{K}{K-1}(a_{K,m} - 1) + \frac{2K}{\sqrt{K-1}}, & \text{Case (U2)}. \end{cases} \quad (11)$$

Similar to Condition (B) in Section 1.1, we introduce

$$\text{Condition (B*).} \qquad \sum_{k=1}^{K-1} |\lambda_{k+1}| (\xi_{k+1}(i)/\xi_1(i))^2 \leq \frac{1}{(1 + \tilde{a}_{K,m})\sqrt{K-1}}, \qquad 1 \leq i \leq n. \quad (12)$$

**Theorem 14** *Fix $(K, m)$ such that $K \geq 2$ and $1 \leq m \leq K - 1$. We have $a_{K,m} \leq a^*_{K,m} \leq \tilde{a}_{K,m}$ and especially $a^*_{K,m} = a_{K,m}$ if Problem (1) is solvable for $(K, m)$. Also, for $a = \tilde{a}_{K,m}$, there is an explicit $K \times K$ orthogonal matrix $Q$ such that $Q' J_{K,m,a} Q$ is doubly stochastic. Finally, if Condition (B*) holds, then there are non-negative matrix $(\Theta, \Pi, P)$ in (2) as desired such that $\Omega = \Theta \Pi P \Pi' \Theta$.*

Theorem 14 is proved in the supplement. For the cases (S1)-(S9), the result follows directly from our results on NIEP. For other cases, we need a new approach. Note that for each case of $(K, m)$, our proofs provide an explicit formula for $Q$ so $Q$ is easy-to-compute. See the supplement for details.

Combining Theorem 14 and our discussion in Section 1.1, we have an improved version of Algorithm 1: Algorithm 2. Note that in Algorithm 1, we need to check both Conditions (A) and (B). In Algorithm 2, we only need to check Condition (B*) (which is same as Condition (B) but with $a = \tilde{a}_{K,m}$). We can do this because when $a = \tilde{a}_{K,m}$, our results on NIEP guarantee that Condition (A) is automatically satisfied with an explicit orthogonal matrix $Q$. Moreover, by the discussion in Section 1.1, our NMF results are sharp in the sense that $\tilde{a}_{K,m}$ is nearly equal to the optimal $a$ in (4) (i.e., $a^*_{K,m}$), so Condition (B*) covers a class of $\Omega$ that is nearly as broad as possible.

In Section 1.1, we mentioned that the NMF problem is partially motivated by the problem of network modeling. In fact, it was pointed out by Jin (2022) that if the NMF problem (2) is solvable, then we

can rewrite a rank-$K$ network model as a DCMM model. Our results suggest that the NMF problem (2) is solvable provided with a mild condition. Therefore, a DCMM model is nearly as broad as a rank-$K$ model. Note that in comparison, a rank-$K$ model is broader, but the DCMM is more useful and interpretable and it is more appealing in practice. Combining these points, we recommend the DCMM as a good model for social networks. See (Jin et al., 2022+; Zhang et al., 2020; Jin & Ke, 2021) for more discussion on DCMM.

## 5 DISCUSSION

NIEP is one of the most challenging problems in research on non-negative matrices. We obtain sharp results for a special NIEP, and apply the results to NMF and network modeling. NIEP, NMF, and analysis of networks are important areas in machine learning and applied mathematics, with applications in image processing, Natural Language Processing, and cancer study Donoho & Stodden (2003); Ke & Wang (2022). We make an interesting connection of the three areas by studying a special NIEP motivated by NMF and network modeling. On one hand, our study may open the door for a new line of research on NIEP. On the other hand, we gain valuable insight on NMF and what are the most suitable network models. These are important for a suitable model is the starting point for methods and theory. Our study may help researchers identify the right network models and so can channel their strengths to the right direction. Our ideas are extendable to NMF and modeling text analysis Ke & Wang (2022); Yuan et al. (2021) and tensor analysis Jin et al. (2021) and analysis of financial networks.

Our study may also motivates a new line of research in optimization theory. By the construction approach we introduced, for some of the $(K, m)$, it remains unclear whether Problem (1) is solvable or not. To tackle these cases, we may use an optimization approach in Section 3, but the approach face some major challenges. To overcome these challenges, we need new ideas. We hope our efforts many spark some new research along this line.

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
