# SUPPLEMENT OF "SHARP RESULTS FOR NIEP AND NMF"

In this supplement file, we first present the proofs of our results in Sections A-H. Next, in Section I, we provide details about estimating $(\Theta, \Pi, P)$ in Example 1 of Section 1.1, and in Section J, we discuss how to numerically check Problem (1) is solvable for a given $(K, m)$. We also present two algorithms (as mentioned in Section 2.2). Last, in Section K, we present the numerical results for the three cases (i.e., $(K, m) = (11, 7), (15, 9), (19, 11)$) where we find that the Problem 1 is solvable using the optimization approach in Section 3 (see also Table 1). Section L explains the connection between the NMF problem (2) and social network modeling.

## A  PRELIMINARY

The following definitions appeared the main file, but for their importance, we restate them. We also present some frequently used lemmas.

**Definition A.1** *Fix $K \geq 3$. Let $c_{K,0} \in \mathbb{R}^K$ be the vector $\sqrt{1/K}(1, 1, \ldots, 1)'$, and when $K$ is even, let $s_{K,0} \in \mathbb{R}^K$ be the vector $\sqrt{2/K}(1, -1, 1, -1, \ldots, 1, -1)'$. Moreover, for any $K \geq 3$ and $1 \leq j \leq (K-2)/2$, let*

$$c_{K,j} = \sqrt{2/K}\Big(1, \cos(\frac{2j\pi}{K}), \cos(\frac{2 \cdot 2j\pi}{K}), \cos(\frac{3 \cdot 2j\pi}{K}), \ldots, \cos(\frac{(K-1) \cdot 2j\pi}{K})\Big)',$$

$$s_{K,j} = \sqrt{2/K}\Big(0, \sin(\frac{2j\pi}{K}), \sin(\frac{2 \cdot 2j\pi}{K}), \sin(\frac{3 \cdot 2j\pi}{K}), \ldots, \sin(\frac{(K-1) \cdot 2j\pi}{K})\Big)'.$$

For any $K \geq 2$, let $F = [c_{K,0}, c_{K,1}, s_{K,1}, \ldots, c_{K,(K-1)/2}, s_{K,(K-1)/2}]$ when $K$ is odd and $F = [c_{K,0}, s_{K,0}, c_{K,1}, s_{K,1}, \ldots, c_{K,(K-2)/2}, s_{K,(K-2)/2}]$ when $K$ is even. It is seen that

$$F \text{ is a } K \times K \text{ orthogonal matrix} \tag{A.1}$$

**Definition A.2** *Fix $K \geq 2$. For any $1 \leq j \leq K/2$, let $h_{K,j} \in \mathbb{R}^K$ be the vector where the $(2j-1)$-th row is $1/\sqrt{2}$, the $2j$-th row is $-1/\sqrt{2}$, and all other rows are 0.*

Consider the matrix
$$H = [h_{K,1}, h_{K,2}, \ldots, h_{K,m}] \in \mathbb{R}^{K,m}.$$
Note that for any $K \geq 2$ and $m \leq K/2$, each column of $H$ has a unit-$\ell^2$-norm, and

$$\text{all columns of } H \text{ are orthogonal to each other.} \tag{A.2}$$

Also, by direct calculations,

$$HH' = \sum_{j=1}^{m} h_{K,j} h'_{K,j} = \begin{bmatrix} D_1 & & \\ & \ddots & \\ & & D_{K/2} \end{bmatrix},$$

where

$$D_1 = \ldots = D_m = \begin{bmatrix} 1 & -1 \\ -1 & 1 \end{bmatrix}, \qquad D_{m+1} = \ldots D_{K/2} = 0.$$

Therefore, for any $K \geq 2$ and $m \leq K/2$,

$$I_K - HH' \text{ is non-negative (recall that } H \in \mathbb{R}^{K,m}). \tag{A.3}$$

Especially, when $K$ is even and $m = K/2$,

$$I_K - HH' \text{ is a traceless non-negative matrix.} \tag{A.4}$$

The following lemmas are frequently used in this supplement and are proved below.

**Lemma A.1** *Suppose $QJ_{K,m}Q'$ is a non-negative matrix for some $K \times K$ orthogonal matrix $Q$.*

- *If the first column of $Q$ is $s_{K,0}$, then the matrix $QJ_{K,m}Q'$ is doubly stochastic.*

- *If the matrix is doubly stochastic and $a_{K,m} > 0$, then the first coumn of $Q$ is $\pm c_{K,0}$.*

**Remark** (*Why we can always assume the first column of $Q$ is $c_{K,0}$*). Suppose there is a $K \times K$ orthogonal matrix $Q$ such that $QJ_{K,m}Q'$ is doubly stochastic. We have two cases: $a_{K,m} > 0$ and $a_{K,m} = 0$. In the first case, by Lemma A.1, the first column of $Q$ is $\pm c_{K,0}$, so we can always assume that it is $c_{K,0}$, without loss of generality. In the second case,

$$J_{K,m} = \text{diag}(1, \ldots, 1, -1, \ldots, -1).$$

Therefore, if we partition

$$Q = [Q_1, Q_2], \qquad Q_1 \in \mathbb{R}^{K,K-m}, Q_2 \in \mathbb{R}^{K,m},$$

then

$$M = QJ_{K,m}Q' = Q_1 Q_1' - Q_2 Q_2',$$

and each column of $Q_1$ is an eigenvctor of $M$ corresponding to the eigenvalue of 1. Since $M$ is doubly stochastic, $c_{K,0}$ is also an eigenvector of $M$ where the corresponding eigenvalue is 1. By basic algebra, there is a $(K - m) \times (K - m)$ rotation matrix $U$ such that

$$c_{K,0} \text{ is the first column of the matrix } \widetilde{Q}_1 =: Q_1 U \in \mathbb{R}^{K,K-m}.$$

Let $\widetilde{Q} = [\widetilde{Q}_1, Q_2]$. It is seen that $\widetilde{Q}$ is a $K \times K$ orthogonal matrix and

$$\widetilde{Q} J_{K,m} \widetilde{Q}' = Q J_{K,m} Q'.$$

Therefore, in this case, without loss of generality, we can always assume that the first column of $Q_1$ is $c_{K,0}$.

**Lemma A.2** *Fix integers $(K, N)$ such that $1 \le N \le K - 1$ and consider a matrix $M = a\mathbf{1}\mathbf{1}_K - 2HH'$ for a scalar $a > 0$ and matrix $H \in \mathbb{R}^{K,N}$. If $H'H = I_N$ and all diagonal entries of $HH'$ are equal to $a/2$, then $M$ is a traceless non-negative matrix.*

## A.1 PROOF OF LEMMA A.1

Consider the first claim. Note that when $Q$ is orthogonal and the first column is $s_{K,0}$, $QJ_{K,m}Q'\mathbf{1}_K = (1 + a_{K,m})s_{K,0}s_{K,0}'\mathbf{1}_K = [(1 + a_{K,m})/K]\mathbf{1}_K$. This proves the claim. Consider the second claim. Denote for short $M = QJ_{K,m}Q'$ and write $Q = [q_1, q_2, \ldots, q_K]$. It is seen that one eigenvalue of $M$ is $(1 + a_{K,m})$ (which is the Perron eigenvalue Horn & Johnson (2013)), $(K - m - 1)$ eigenvalues of $M$ are 1, and $m$ eigenvalues of $M$ are $-1$. Especially, the Perron eigenvalue is larger than 1, with a multiplicity of 1. Therefore, the corresponding eigen-space is one-dimensional. Note also the Perron eigenvalue equals to the spectral norm of $M$, denoted by $\rho(M)$.

Now, first, it is seen that

$$\rho(M) = 1 + a_{K,m} \qquad \text{and} \qquad Mq_1 = (1 + a_{K,m})q_1 = \rho(M)q_1.$$

Second, by Perron's theorem (Horn & Johnson, 2013, Page 503), there is nonzero and non-negative eigenvector $\xi$ such that

$$M\xi = \rho(M)\xi = \rho(M)\xi, \tag{A.5}$$

Last, since $M$ is doubly stochastic, there is a number $a > 0$ such that

$$\mathbf{1}_K = a\mathbf{1}_K, \qquad \text{and so } Mc_{K,0} = ac_{K,0}.$$

Combining these,

$$\xi' M c_{K,0} = \rho(M)\xi' c_{K,0} = a\xi'\mathbf{1}. \tag{A.6}$$

Since $\xi' c_{K,0} > 0$, we must have $a = \rho(M)$, and so

$$M c_{K,0} = \rho(M) c_{K,0}. \tag{A.7}$$

Combining (A.5)-(A.7) and recalling that the eigenspace of corresponding to $\rho(M)$ is one-dimensional,

$$q_1 = \pm c_{K,0}.$$

This proves the claim and completes the proof of Lemma A.1.

### A.2 Proof of Lemma A.2

It is seen that all diagonal entries of $M$ are 0. Moreover, since all diagonal entries of $HH'$ are $a$, an off-diagonal entry will not exceed $a$, by Cauchy-Schwartz inequality. Therefore, all off-diagonal entries of $M$ are non-negative. This proves Lemma A.2.

## B Proof of Theorems 2-4

In this section, we prove Theorem 2, Theorem 3, and Theorem 4.

### B.1 Proof of Theorem 2

When $K \geq 5$ and $m = K - 2$, $m > K/2$ and $J_{K,m} = \text{diag}(1 + a_{K,m}, 1, -1, \ldots, -1)$. Also, by definition,

$$1 + a_{K,m} = (2m - K) + 1 = K - 3 > 1.$$

Suppose that there is a $K \times K$ orthogonal matrix $Q$ such that

$$M \equiv Q J_{K,m} Q'$$

is doubly stochastic. Write

$$Q = [q_1, q_2, \ldots, q_K].$$

By Lemma A.1, $q_1 = \pm c_{K,0}$. Without loss of generality, we assume

$$q_1 = c_{K,0}.$$

Now, write

$$Q J_{K,m} Q' = Q\text{diag}(K - 3, 1, -1, \ldots, -1)Q' = (K - 2)q_1 q_1' + 2q_2 q_2' - I_K.$$

It is seen that $\text{trace}(M) = \text{trace}(Q J_{K,m} Q') = \text{trace}(J_{K,m}) = 0$. Since $M$ is non-negative, all diagonal entries of $M$ are 0. By direct calculations, this implies that all entries of $q_2$ are $\pm 1/\sqrt{K}$. Suppose $N$ entries of $q_2$ are $-1/\sqrt{K}$ and $(K - N)$ entries are $1/\sqrt{K}$. Recall that $q_1 = c_{K,0}$ and all entries of $q_1$ are $1/\sqrt{K}$. When $K$ is odd, we can not have

$$q_1' q_2 = (1/K)(K - 2N)$$

When $K$ is odd, the RHS is nonzero, and so a contradiction. The contradiction proves the claim.

### B.2 Proof of Theorem 3

The theorem contains (a)-(c). Note that for any $(K, m)$ with $K \leq 4$ and $1 \leq m \leq K - 1$, we either have (1) $m \leq K/2$ or (2) $m > K/2$ and $m = K - 1$. The claim for Case (1) is a direct result of Theorem 8. Therefore, once (a)-(b) are proved, then (c) follows from For this reason, we only prove (a)-(b).

Consider (a). In this case, $m = (K - 1)$ and $m \geq 1$ (and so $K \geq 2$). By definition,

$$a_{K,m} = 2m - K = K - 2 \geq 0. \tag{B.8}$$

Let

$$Q = [c_{K,0}, Q_0],$$

where $Q_0$ is any $K \times (K-1)$ matrix satisfying $Q_0'Q_0 = I_{K-1}$. By direct calculations,

$$QJ_{K,m}Q' = Q\mathrm{diag}(2 + a_{K,m}, 0, \ldots, 0)Q' - I_K = [(2 + a_{K,m})/K]\mathbf{1}\mathbf{1}' - I_K.$$

Combining this with (B.8),

$$QJ_{K,m}Q' = \mathbf{1}\mathbf{1}' - I_K,$$

which is a traceless non-negative matrix. Since the first column of $Q$ is $s_{K,0}$, by Lemma A.1, $QJ_{K,m}Q'$ is doubly stochastic. This proves the claim.

Consider (b). Note that in this case, $K$ is even. The case $K = 2$ is trivial, so we assume $K \geq 4$. By definition,

$$a_{K,m} = 2m - K = K - 4 \geq 0.$$

Similarly, let

$$Q = [c_{K,0}, s_{K,0}, Q_0],$$

where $Q_0$ is any $K \times (K-2)$ matrix satisfying $Q_0'Q_0 = I_{K-2}$. Similarly, in this case,

$$QJ_{K,m}Q' = (a_{K,m} + 2)c_{K,0}c_{K,0}' + 2s_{K,0}s_{K,0}' - I_K = ((K-2)/K)\mathbf{1}\mathbf{1}' + 2s_{K,0}s_{K,0}' - I_K.$$

By definitions, it is seen that (a) all diagonal entries of the matrix $QJ_{K,m}Q'$ are $((K-2)/K) + (2/K) - 1 = 0$, and (b) all off-diagonal entries $\geq ((K-2)/K) - 2/K = (K-4)/K \geq 0$. Therefore, the matrix is non-negative. Since the first column of $Q$ is $c_{K,0}$, the matrix is doubly stochastic by Lemma A.1. This proves the claim and completes the proof of Theorem 3.

## B.3 Proof of Theorem 4

In this case, $(K, m) = (7, 4)$, $m \geq K/2$, $2m - K = 1$, and $J_{K,m} = \mathrm{diag}(2, 1, 1, -1, \ldots, -1)$. The goal is to show Problem 1 is not solvable in this case.

If Problem (1) is solvable in this case, then there is a $7 \times 7$ orthogonal matrix $Q$ such that

$$Q = [c_{K,0}, Q_1, G]$$

such that

$$QJ_{K,m} = (3/7)\mathbf{1}_7\mathbf{1}_7 + 2Q_1Q_1' - I_7$$

is non-negative. Since $J_{K,m}$ is traceless, $QJ_{K,m}Q'$ is a traceless non-negative matrix. Therefore,

- all diagonal entries of $Q_1Q_1'$ are $2/7$,
- all off-diagonal entries of $Q_1Q_1'$ are no smaller than $-3/14$.

For $1 \leq i \leq 7$, let row $i$ of $Q_1$ be

$$(\cos(\theta_i), \sin(\theta_i)), \qquad 0 \leq \theta_1 \leq \theta_2 \leq \ldots \leq \theta_7 < 2\pi.$$

Since $Q$ is an orthogonal matrix, by elementary algebra and triogometrics, we must have

- (R1). $\sum_{i=1}^7 \cos(\theta_i) = 0$.
- (R2). $\sum_{i=1}^7 \sin(\theta_i) = 0$.
- (R3). $\sum_{i=1}^n \cos(2\theta_i) = 0$.
- (R4). $\sum_{i=1}^n \sin(2\theta_i) = 0$.
- (R5). $\cos(\theta_i - \theta_j) \geq -3/4$.

For each $\theta_i$, there are two neighbors, $\theta_{i-1}$ and $\theta_{i+1}$ (the neighbors of $\theta_7$ are $\theta_6$ and $\theta_1$). Define the maximal neighboring distance for point $i$ by

$$d_i = \max\{|\theta_i - \theta_{i-1}|, |\theta_i - \theta_{i+1}|\}, \tag{B.9}$$

Note that if (R1)-(R5) hold for $\{\theta_1, \theta_2, \ldots, \theta_7\}$, then they also hold for

$$\{\theta_1 + \Delta, \theta_2 + \Delta, \ldots, \theta_7 + \Delta\} \qquad \text{(which is a rotation on the circle).} \qquad \text{(B.10)}$$

We call this *symmetry in rotation*. Therefore, without loss of generality, we assume

$$d_1 = \min_{1 \le i \le 7}\{d_i\}, \qquad \theta_1 = 0, \qquad \theta_1 \le \theta_2 \le \theta_3 \le \ldots \le \theta_7 < 2\pi. \qquad \text{(B.11)}$$

We now show

either that $\sum_{i=1}^{7} \cos(\theta_i) = 0$, or that $\sum_{i=1}^{7} \cos(\theta_i) = 0$ but $\sum_{i=1}^{n} \sin(\theta_i) \ne 0$. $\qquad$ (B.12)

Once this is proved, some of (R1)-(R5) are not satisfied. The contradiction proves the claim.

We now show (B.12). Let $\alpha = \cos^{-1}(3/4) \approx 41.4°$ and $\beta = (1-\alpha)/2 \approx 69.3°$. By (R5) and that $\theta_1 = 0$,

$$\cos(\theta_i) \ge -3/4,$$

so all $\theta_1, \theta_2, \ldots, \theta_7$ belongs to the region

$$\Omega = \{0 \le \theta \le \pi - \alpha\} \cup \{\pi + \alpha \le \theta \le 2\pi\}.$$

We now divides $\Omega$ into 4 regions, (I)-(V) as follows.

- (I). $\{0 \le \theta \le \beta\}$.
- (II). $\{\beta < \theta \le \pi - \alpha\}$.
- (III). $\{\pi + \alpha \le \theta < 2\pi - \beta\}$.
- (IV). $\{2\pi - \beta \le \theta < 2\pi\}$.

Suppose there are $m$ different $\theta_i$ are 0. It follows that when $m \ge 3$,

$$\sum_{i=1}^{7} \cos(\theta_i) \ge m - (3/4)(7-m) \ge 0,$$

with equality only when $m = 3$ and all remaining 4 other $\theta_i$ satisfying $\cos(\theta_i) = -3/4$. In such a special case, $\sum_{i=1}^{6} \cos^2(\theta_i) \ne \sum_{i=1}^{7} \sin^2(\theta_i)$, so (R3) is violated. Therefore, the result follows when $m \ge 3$. If $m = 2$, then we treat $\theta_1 = \theta_2 = 0$. For this reason, we can assume

no more than $\theta_i$ are 0 and when there are two $\theta_i$ are 0, $\theta_1 = \theta_2 = 0$. $\qquad$ (B.13)

If (R1)-(R5) hold, then

At least one points in $\{\theta_2, \ldots, \theta_7\}$ fall in (II) and at least one of them fall in (III). (B.14)

To show this, without loss of generality, assume $(III)$ does not contains any of the 6 points. Therefore, up to a rotation on the circle and relabeling of the 7 points, we have

$$0 = \theta_1 \le \theta_2 \le \ldots \le \theta_7 \le \pi + (\beta - \alpha) \approx 208°,$$

where the RHS is smaller than $\pi + \alpha \approx 221.4°$. By (R5), for any $i < j$,

$$\cos(\theta_j - \theta_i) \le -3/4,$$

so we either have

$$0 \le \theta_j - \theta_i < \pi - \alpha, \qquad \text{or} \qquad \pi + \alpha \le \theta_j - \theta_i < 2\pi.$$

Combining these, we must have

$$0 \le \theta_1 \le \theta_2 \le \ldots \le \theta_7 \le \pi - \alpha.$$

By (B.10), (R1)-(R5) holds for such $\{\theta_1, \theta_2, \ldots, \theta_7\}$, but this contradicts with $(R2)$ as all $\sin(\theta_i) \ge 0$. This proves (B.14).

Moreover, by (B.9)-(B.11),

- If Region (I) does not contain any of the 6 points $\{\theta_2, \ldots, \theta_7\}$ and Region (IV) does not contains any of these point either, then Region (II) has at most two points. Otherwise, if Region (II) has three or more points, then we can find one among them (say, $i$) such that $d_i < d_1$, where $d_i$ is maximum neighboring distance defined in (B.9). Similarly, Region (III) has at most two points. However, these say Region (I)-(IV) contain at most 4 points of $\{\theta_2, \ldots, \theta_7\}$, and so a contradiction. Therefore, the total points in Region (I) and Region (IV) is nonzero.

- Suppose Region (I) contains no point of $\{\theta_2, \ldots, \theta_7\}$ and Region (IV) contains exactly one of them. In this case, Region II contains no more than two points of them, so Region (IV) has at least three points of them. This again contradicts with (B.9)-(B.11), by similar reasons as above. Suppose Region (I) has none of these 6 points and Region (IV) contains two or more of them. This again contradicts with (B.9)-(B.11), for one point in Region (IV) may have a smaller maximum neighboring distance then $\theta_1$.

Combining the above as well as (B.12), in each of the four regions, (I)-(IV),

$$\text{we have at least one point } \{\theta_2, \ldots, \theta_7\}, \text{ but no more than two of them.} \qquad \text{(B.15)}$$

We now study

$$\sum_{i=1}^{7} \cos(\theta_i) = 1 + (A) + (B),$$

where

$$(A) = \sum_{\{i : \theta_i \in (IV)\}} \cos(\theta_i) + \sum_{\{i : \theta_i \in (II)\}} \theta_i,$$

and

$$(B) = \sum_{\{i : \theta_i \in (IV)\}} \cos(\theta_i) + \sum_{\{i : \theta_i \in (II)\}} \theta_i.$$

We now analyze (A) first. Suppose (IV) contains $s$ points of $\{\theta_2, \ldots, \theta_7\}$ and (II) contains $t$ of them. By the above arguments,

$$1 \leq s, t \leq 2.$$

Since for any $\theta_i, \theta_j$ in (II) or (IV),

$$|\theta_i - \theta_j| \leq \pi + \beta - \alpha < 2\pi - \alpha,$$

we must have

$$|\theta_i - \theta_j| \leq \pi - \alpha \equiv 2\beta.$$

It is seen for an $x$ such that $-\beta < x < 0$,

$$(A) \geq s\cos(x) + t\cos(x + 2\beta) \equiv S(x).$$

Using $\cos(2\beta) = -3/4$ and $\sin(x) = \sqrt{7}/4$,

$$S(x) = (s - 3t/4)\cos(x) - (\sqrt{7}t/4)\sin(x), \qquad S'(x) = -(s - 3t/4)\sin(x) - (\sqrt{7}t/4)\cos(x).$$

We have three cases.

- (1). $s = 1, t = 2$. In this case, $(s - 3t/4) \leq 0$. In this case, $S(X)$ is monotonely decreasing, with minimum (achieved at $4x = 0$) of

$$s - 3t/4 = -1/2;$$

note that (IV) does not contain the point 0 or $2\pi$, so the minimum is not achievable.

- (2). $s = t$. In this case, $s - 3t/4 > 0$,

$$S'(x) = -(3/4)[\sin(x) + \sqrt{7}\cos(x)].$$

In this case, the minimum is no smaller than

$$s/4.$$

- (3). $s = 2, t = 1$. In this case,

$$S(x) = (5/4)\cos(x) - (\sqrt{7}/4)\sin(x), \qquad S'(x) = -(5/4)\sin(x) - (\sqrt{7}/4)\cos(x).$$

The minimum is achieved when $S'(x) = 0$, with the value of

$$1/4.$$

Therefore, the only case where $(A)$ is negative is when

$$(s, t) = (1, 2) \qquad \text{that is, (IV) contains one point of } \{\theta_2, \ldots, \theta_7\}, \text{ and (II) contains 2 of them.}$$

Also, the minimum strictly larger than $-1/2$, achieved only when the point in $(IV)$ is (approaching) 0 and the two points in (II) are $\pi - \alpha$. Similarly, $(B)$ can only be negative when (I) contains one point of $\{\theta_2, \ldots, \theta_7\}$, and (III) contains 2 of them, where the minimum is $-1/2$ achieved only when the point in $(I)$ is 0 and the two points in (III) are $\pi - \alpha$. Combining these,

$$\sum_{i=1}^{7} \cos(\theta_i) > 0.$$

This says (R1)-(R5) can not hold simultaneously. The contradiction proves the claim.

## C  PROOF OF THEOREMS 5-6

In this section, we prove Theorem 5 and Theorem 6.

### C.1  PROOF OF THEOREM 5

The goal is to show when $K \geq 2$ and $m \leq K/2 \leq K - 1$, Problem 1 is solvable. The case of $K = 2$ is trivial, so we assume $K \geq 3$. Let

$$Q_0 = [c_{K,0}, h_{K,1}, \ldots, h_{K,m}] \in \mathbb{R}^{K,m+1}.$$

Note that since $m \leq K/2$ and $K \geq 3$, $m + 2 \leq K$. It is seen that

$$Q_0' Q_0 = I_{m+1}, \qquad \text{where } m + 1 \leq K - 1.$$

Therefore, there is a matrix $G \in \mathbb{R}^{K, K-(m+1)}$ such that the matrix

$$Q = [c_{K,0}, G, \ldots, h_{K,1}, \ldots, h_{K,m}]$$

is a $K \times K$ orthogonal matrix. Since $m \leq K/2$, $a_{K,m} = 0$ by definition. Therefore, in this case,

$$J_{K,m} = (1, \ldots, 1, -1, \ldots, -1),$$

where $m$ is the number of $-1$'s. It follows that

$$Q J_{K,m} Q' = 2 [I_K - \sum_{j=1}^{m} h_{K,j} h_{K,j}'].$$

By direct calculations,

$$\sum_{j=1}^{m} h_{K,j} h_{K,j}' = \begin{bmatrix} D_1 & & & & & \\ & \ddots & & & & \\ & & D_m & & & \\ & & & 0 & & \\ & & & & \ddots & \\ & & & & & 0 \end{bmatrix},$$

where

$$D_1 = \ldots = D_m = \begin{bmatrix} 1 & -1 \\ -1 & 1 \end{bmatrix}.$$

Therefore, $Q J_{K,m} Q'$ is a $K \times K$ symmetric non-negative matrix. Moreover, since the first column of $Q$ is $c_{K,0}$, so by Lemma A.1, $Q J_{K,m} Q'$ is doubly stochastic. This proves the claim and completes the proof of Theorem 3.

## C.2  PROOF OF THEOREM 6

In this case, $K$ is dyadic so there is an integer $s \geq 1$ so that $K = 2^s$. Let $H_0$ be the Haar basis matrix. For example, when $K = 4$,

$$H_0 = \begin{bmatrix} 1 & 1 & 1 & 0 \\ 1 & 1 & -1 & 0 \\ 1 & -1 & 0 & 1 \\ 1 & -1 & 0 & -1 \end{bmatrix}.$$

and when $K = 8$,

$$H_0 = \begin{bmatrix} 1 & 1 & 1 & 0 & 1 & 0 & 0 & 0 \\ 1 & 1 & 1 & 0 & -1 & 0 & 0 & 0 \\ 1 & 1 & -1 & 0 & 0 & 1 & 0 & 0 \\ 1 & 1 & -1 & 0 & 0 & -1 & 0 & 0 \\ 1 & -1 & 0 & 1 & 0 & 0 & 1 & 0 \\ 1 & -1 & 0 & 1 & 0 & 0 & -1 & 0 \\ 1 & -1 & 0 & -1 & 0 & 0 & 0 & 1 \\ 1 & -1 & 0 & -1 & 0 & 0 & 0 & -1 \end{bmatrix}.$$

In general, we can write

$$H_0 = [h_1, h_2, \ldots, h_K]$$

We remove the first column and the last $(K/2)$ columns of $H_0$, and partition the remaining $(K/2) - 1$ columns into $(s+1)$ blocks as follows:

$$H_0^* = [h_2, h_3, \ldots, h_K] = [U_0, U_1, U_2, \ldots, U_{s-2}],$$

where $U_j$ are blocks of columns of $H$ as follows:

- Block 0, $U_0$, contains $2^0$ vector which is $h_1$.
- Block 1, $U_1$, contains $2^1$ vectors $h_3$ and $h_4$.
- Block 2, $U_2$, contains $2^2$ vectors $h_5, \ldots, h_8$.
- The last block $U_{s-1}$ contains $2^{s-1}$ vectors $h_{(K/4)+1}, \ldots, h_{K/2}$.

By basic number theory, for any number $1 \leq m \leq K$, there is a unique way where we can write

$$m - K/2 = \sum_{j=0}^{s-2} a_j 2^j, \qquad a_j \in \{0, 1\}. \tag{C.16}$$

We construct a matrix $Q_1$ as follows. We start with

$$H^* = [U_0, U_2, \ldots, U_{s-2}] \in \mathbb{R}^{K, K/2 - 1}.$$

Next, for $0 \leq j \leq s - 1$, remove block $U_j$ if and only if $a_j = 0$ in (C.16). Denote the resultant matrix by $Q_1$. We have the following observations.

- $Q_1$ has exactly $\sum_{j=0}^{s-1} a_j 2^j = m$ columns.
- For each block $U_j$, $0 \leq j \leq s - 2$, either all columns of $U_j$ shows up in $Q_1$ or none of its columns shows up in $Q_1$ (no column shows up more than once).
- For each $0 \leq j \leq s - 1$, $U_j' U_j = I_{2^j}$ and all diagonal entries of $U_j U_j'$ are the same.

It follows that

$$Q_1' Q_1 = I_{m-K/2}, \qquad \text{and} \qquad \text{all diagonal entries of } Q_1 Q_1' \text{ are the same.} \tag{C.17}$$

Next, let

$$Q_2 = [h_{(K/2)+1}, h_{(K/2)+2}, \ldots, h_K]$$

and

$$Q_0 = [Q_1, Q_2].$$

It follows that

$$Q_0'Q_0 = I_m.$$

Therefore, there is a matrix $G \in \mathbb{R}^{K, K-m-1}$ such that

$$Q = [c_{K,0}, G, Q_1, Q_2]$$

is a $K \times K$ orthogonal matrix. Recall that $J_{K,m} = \text{diag}(1 + a_{K,m}, 1, 1, \ldots, -1, \ldots, -1)$ where we have exactly $m$ of $-1$'s. It follows that

$$Q J_{K,m} Q' = (a_{K,m}/K)\mathbf{1}\mathbf{1}' + Q_1 Q_1' - Q_2 Q_2' - Q_3 Q_3' = (a_{K,m}/K)\mathbf{1}\mathbf{1}' + I_K - 2Q_2 Q_2' - Q_3 Q_3' = (I) + (II),$$

where

$$(I) = (a_{K,m}/K)\mathbf{1}\mathbf{1}' - 2Q_1 Q_1,$$

and

$$(II) = I_K - 2Q_2 Q_2'.$$

Now, first, by (A.3),

$$\text{(II) is a traceless non-negative matrix.}$$

Second, since $m > K/2$, $a_{K,m} = (2m - K)$, and

$$\text{trace}(Q J_{K,m} Q') = \text{trace}(J_{K,m}) = (2m - K + 1) + (K - m - 1) - m = 0.$$

Combining these,

$$\text{trace}((I)) = 0.$$

By (C.17), all diagonal entries of $2Q_1 Q_1'$ are the same; denote the common value by $a$. It follows that

$$0 = \text{trace}((I)) = a_{K,m} - Ka,$$

and so $a = a_{K,m}/K$. By Cauchy-Schwartz inequality, all off-diagonals of $2Q_2 Q_2'$ are no greater than $a$. Therefore, all diagonal entries of $(I)$ are no smaller than

$$(a_{K,m}/K) - a \geq 0,$$

and so $(I)$ is non-negative.

Combining these, $Q J_{K,m} Q'$ is non-negative. Since the first column of $Q$ os $s_{K,0}$, by Lemma A.1, $Q J_{K,m} Q'$ is doubly stochastic. This proves the claim.

## D    Proof of Lemma 7 and Theorems 8-9

In this section, we prove Lemma 7 and Theorems 8-9.

### D.1    Proof of Lemma 7

We prove these for the case (a)-(d) separately. Consider (a). Let $n = (K - m - 1)/2$. In this case, $n$ is an integer. Also, by (D.18), $n \geq 1$. Let

$$Q_0 = [c_{K,0}, Q_1], \qquad \text{where} \qquad Q_1 = [c_{K,1}, s_{K,1}, \ldots, c_{K,n}, s_{K,n}].$$

By (A.1), $Q_0'Q_0 = I_{2n+1}$, where $(2n + 1) = (K - m) < K$. Therefore, there is a matrix $G \in \mathbb{R}^{K,m}$ such that the matrix

$$G = [Q_0, G] = [c_{K,0}, Q_1, G]$$

is a $K \times K$ orthogonal matrix. Moreover, by (A.1), each diagonal entry of $Q_1 Q_1'$ is

$$(2/K)n = (K - m - 1)/K,$$

so (D.20) holds, and the claim follows.

Consider (b). Let $n = (K - m - 2)/2$. Note that in the current case, $K$ is even, $n$ is an integer and $n \geq 0$. Let

$$Q_0 = [c_{K,0}, Q_1], \qquad \text{where} \qquad Q_1 = [s_{K,0}, c_{K,1}, s_{K,1}, \ldots, c_{K,n}, s_{K,n}].$$

By (A.1), $Q_0'Q_0 \in I_{2n+2}$. Since $2n + 2 = K - m < K$, there is a matrix $G \in \mathbb{R}^{K,m}$ such that the matrix

$$Q = [c_{K,0}, Q_1, G]$$

is an orthogonal matrix. Moreover, by construction, all diagonal entries of $Q_1 Q_1'$ are

$$(1/K) + (2n)/K = (2n + 1)/K = (K - m - 1)/K,$$

so (D.20) holds. The claim follows by using Lemma D.1.

Consider case (c). This case is very similar to that of case (b) so we omit the proof.

Consider (d). Let $K = 2N + 1$ and $n = (K - m - 2)/2$. Since (a) both $K$ and $K - m - 1$ are odd, and (b) $m \neq K - 2$, we must have $m \leq K - 4$ and so $n \geq 1$. By the construction, it is seen that all columns of $Q_1$ are orthogonal to each other. Therefore, It is sufficient to show

- (d1) all columns of $Q_1$ are orthogonal to the vector $\mathbf{1}_K$,
- (d2) the $\ell^2$-norm of each column of $Q_1$ are 1,
- (d3) the square $\ell^2$-norm of each row of $Q_0$ is $(2n + 1)/K$,

Consider (d1). By constructions and (A.1), $\mathbf{1}_{(K+1)/2}$ is orthogonal to all columns of $B_1$, and $\mathbf{1}_{(K-1)/2}$ is orthogonal to all columns of $B_2$. Therefore, for any $2 \leq j \leq 2n + 1$, $\mathbf{1}_K$ is orthogonal to column $j$ of $Q_0$. At the same time, the sum of all entries of the first column of $Q_0$ is

$$c_0(K + 1) - d_0(K - 1) = \sqrt{(K^2 - 1)/(2K)} - \sqrt{(K^2 - 1)/(2K)} = 0,$$

so $\mathbf{1}_K$ is also orthogonal to the first column of $Q_0$. This proves (d1).

Consider (d2). For the first column of $Q_0$, the square $\ell^2$-norm is seen to be

$$\left(\frac{K + 1}{2}\right)c_0^2 + \left(\frac{K - 1}{2}\right)d_0^2 = (K - 1)/(2K) + (K + 1)/(2K) = 1.$$

For any $1 \leq j \leq n$, by (A.1), for either $N = (K + 1)/2$ or $N = (K - 1)/2$, the square $\ell^2$-norm of $c_{N,j}$ and $s_{N,j}$ are 1. Therefore, fixing $2 \leq j \leq 2n + 1$ and considering column $j$ of $Q_0$, the square $\ell^2$-norm is

$$c^2 + d^2 = \frac{1}{K}\left(\frac{K + 1}{2} + \frac{1}{2n}\right) + \frac{1}{K}\left(\frac{K - 1}{2} - \frac{1}{2n}\right) = 1.$$

This proved (d2).

Consider (d3). Note that by the construction, the square $\ell^2$-norm of each row of $B_1$ is $2n/((K + 1)/2) = 4n/(K + 1)$, and that for each row of $B_2$ is $4n/(K - 1)$. Therefore, he square $\ell^2$-norm of the $j$-th row of $Q_0$ is

$$\begin{cases} c_0^2 + c^2\frac{4n}{K+1}, & 1 \leq j \leq (K + 1)/2, \\ d_0^2 + d^2\frac{4n}{K-1}, & (K + 1)/2 + 1 \leq j \leq K. \end{cases}$$

Now,

$$c_0^2 + c^2\frac{4n}{K + 1} = \frac{(K - 1)}{K(K + 1)} + \frac{1}{K}\left(\frac{K + 1}{2} + \frac{1}{2n}\right) \times \frac{4n}{K + 1} = \frac{2n + 1}{K},$$

and

$$d_0^2 + d^2\frac{4n}{K - 1} = \frac{K + 1}{K(K - 1)} + \frac{1}{K}\left(\frac{K - 1}{2} - \frac{1}{2n}\right) \times \frac{4n}{K - 1} = \frac{2n + 1}{K}.$$

This verifies (d3) and completes the proof of Lemma 7.

### D.2  PROOF OF THEOREM 8

The goal is to show Problem (1) is solvable when $m + 1 \geq 3K/4$ and $m \neq K - 2$ when $K$ is odd. Since the case of $K \leq 4$ and the case of $K = m - 1$ are proved in Theorem 2, we assume

$$(m + 1) \geq 3K/4, \qquad K \geq 5, \qquad \text{and } K \neq m - 2 \text{ when } K \text{ is odd.} \qquad (D.18)$$

Note that in the current case, $m \geq K/2$, so by definition,

$$a_{K,m} = (2m - K).$$

The following lemma is proved below.

**Lemma D.1** *Fix $(K, m)$ as in Theorem 8 and consider a $K \times K$ orthogonal matrix $Q$ with the form of*

$$Q = [c_{K,0}, Q_1, G], \qquad where \ Q_1 \in \mathbb{R}^{K,K-m-1} \ and \ G \in \mathbb{R}^{K,m}. \qquad (D.19)$$

*If all diagonal entries of $Q_1 Q_1'$ equal to $(K - m - 1)/K$, then*

$$Q J_{K,m} Q'$$

*is a non-negative doubly stochastic matrix.*

**Proof of Lemma D.1**. Recall that $J_{K,m} = \text{diag}(1 + a_{K,m}, 1, \ldots, 1, -1, \ldots, -1)$ and $a_{K,m} = (2m - K)$. It follows that

$$M \equiv Q J_{K,m} Q' = (2 + a_{K,m}) c_{K,0} c_{K,0}' + 2 Q_1 Q_1' - I_K.$$

First, it is seen that each diagonal entry of $M$ is

$$(2 + a_{K,m})/K + 2(K - m - 1)/K - 1 = (1/K)[(2 + 2m - K) + 2(K - m - 1)] - 1 = 0.$$

Second, since all diagonal entries of $Q_1 Q_1'$ are $(K - m - 1)/K$, by Cauchy-Schwartz inequality, any off-diagonal entry of $Q_1 Q_1'$

$$\geq -(K - m - 1)/K.$$

Therefore, any off-diagonal entry of $M$ is

$$\geq (2 + a_{K,m})/K - 2(K - m - 1)/K = (1/K)[(2 + 2m - K) - 2(K - m - 1)] = (1/K)[4(m + 1) - 3K],$$

and the claim follows by $(m + 1) \geq (3K)/4$.

We now prove Theorem 8. By Lemma 7 and the way $Q_1$ is constructed, all columns of the matrix $[c_{K,0}, Q_1]$ have unit $\ell^2$-norm and are orthogonal to each other. Therefore, there is a matrix $G \in \mathbb{R}^{K,m}$ such that

$$Q = [c_{K,0}, Q_1, G]$$

is a $K \times K$ orthogonal matrix. Also, by Lemma 7,

$$\text{all diagonal entries of } Q_1 Q_1' \text{ are } (K - m - 1)/\text{K}. \qquad (D.20)$$

By Cauchy-Schwartz inequailty,

$$\text{all off-diagonal entries of } Q_1 Q_1' \text{ are no smaller than } -(K - m - 1)/\text{K}. \qquad (D.21)$$

Without loss of generality, assume $K \geq 4$. When $(m + 1) \geq 3K/4$, $m \geq K/2$ and so $a_{K,m} = (2m - K)$. By (6),

$$M = Q J_{K,m} Q' = (2 + a_{K,m}) c_{K,0} c_{K,0} + 2 Q_1 Q_1' - I_K = [(2m - K + 2)/K] \mathbf{1}_K \mathbf{1}_K' + 2 Q_1 Q_1' - I_K.$$

Combining this with (D.20)-(D.21), first, all diagonal entries of $M$ are

$$[(2m - K + 2)/K] + 2(K - m - 1)/K - 1 = 0,$$

and second, all off-diagonal entries of $M$ are

$$\geq [(2m - K + 2)/K] - 2(K - m - 1)/K = (4(m + 1) - 3K)/K,$$

where the RHS is non-negative as $(m + 1) \geq (3K/4)$. Therefore, $M$ is doubly stochastic. This proves Theorem 8.

### D.3 Proof of Theorem 9

In this case, $K$ is odd and $m \neq (K - 2)$. By Theorems 2 and 8, Problem (1) is solvable when $K \leq 4$, when $m \leq K/2$, and when $(m + 1) \geq 3K/4$ but $m \neq K - 2$. For these reasons, we assume

$$\max\{a_K, b_K\} K \leq m < (3K/4) - 1, \qquad m \neq (K - 2), \qquad K \geq 5. \qquad (D.22)$$

Especially, in this case, since $\max\{a_K, b_K\} > 1/2$, so in this case,

$$m > K/2, \qquad \text{and so by definition,} \qquad a_{K,m} = 2m - K.$$

Let $n = (K - m - 1)/2$. It is seen $n > 0$. Since $K$ is odd and $m$ is even, $n$ is an integer and $n \geq 1$. Let

$$Q_0 = [c_{K,0}, Q_1], \qquad \text{where} \qquad Q_1 = [c_{K,1}, s_{K,1}, \ldots, c_{K,n}, s_{K,n}].$$

It is seen that $Q_0' Q_0 = I_{2n+1}$, where $(2n + 1) = K - m$, so there is a matrix $G \in \mathbb{R}^{K,m}$ such that

$$Q = [c_{K,0}, Q_1, G]$$

is a $K \times K$ orthogonal matrix. It follows

$$M \equiv Q J_{K,m} Q' = (a_{K,m} + 2) c_{K,0} c_{K,0} + 2 Q_1 Q_1' - I_K.$$

By the construction, all diagonal entries of $Q_1 Q_1'$ are $2n/K$, so all diagonal entries of $M$ are

$$(2 + a_{K,m})/K + 4n/K - 1 = (1/K)[(2 + 2m - K) + 2(K - m - 1)] - 1 = 0.$$

At the same time, by basic trigonometrics and

$$\cos(x) + \ldots + \cos(nx) = -(1/2) + \frac{\sin((n + 1/2)x)}{2 \sin(x/2)},$$

we have that for any $i \leq j$,

$$M(i,j) = (2 + a_{K,m})/K + (4/K) \sum_{k=1}^{n} [\cos(\frac{2\pi(i-1)k}{K}) \cos(\frac{2\pi(j-1)k}{K}) + \sin(\frac{2\pi ik}{K}) \sin(\frac{2\pi jk}{K})]$$

$$= (1/K)[(2m - K + 2) + 4 \sum_{k=1}^{n} \cos(\frac{2\pi(i-j)k}{K})]$$

$$= (2/K)[(m - K/2) + \frac{\sin((K - m)|i - j|\pi/K)}{\sin(|i - j|\pi/K)}]$$

$$\equiv (2/K) g_{|i-j|},$$

where for $1 \leq k \leq K - 1$,

$$g_k = (m - K/2) + \frac{\sin((K - m)k\pi/K)}{\sin(k\pi/K)}.$$

Note here $1 \leq |i - j| \leq K - 1$, and for all $1 \leq k, \ell \leq K$ with $k + \ell = K$,

$$g_k = g_\ell.$$

Therefore, to show that $M$ is non-negative, it is sufficient to show that when (D.22) holds,

$$g_k \geq 0, \qquad \text{for any } 1 \leq k \leq (K - 1)/2. \tag{D.23}$$

We now show (D.23). Recall that in our range of interest, $K/2 < m < (3K/4) - 1$. We have the following observations.

- (a) For all $1 \leq k \leq (K - 1)/2$, $\sin(k\pi/K) > 0$.
- When $1 \leq k \leq 2$, $0 < (K - m)k/K < 1$, so $\sin((K - m)k\pi/K) > 0$. Therefore,

$$g_k = (m - K/2) + \frac{\sin((K - m)k\pi/K)}{\sin(k\pi/K)} \geq 0.$$

- (b) When $k = 3$, since $\sin(3\pi - z) = \sin(z)$ for any $z$,

$$g_k m - K/2 - \frac{\sin(3(K - m)\pi/K)}{\sin(2\pi/K)} = (m - K/2) - \frac{\sin(3m\pi/K)}{\sin(3\pi/K)}$$

$$= K[(m/K) - 1/2 + \frac{\sin(3\pi m/K)}{\sin(3\pi/K)}].$$

Consider the function $x - 1/2 + \sin(3\pi x)/\sin(3\pi/K)$ in $1/2 < x < 3/4$. It is seen that the function strictly increasing in $1/2 < x < 3/4$ and $b_K \in (1/2, 3/4)$ is the unique solution of the equation $x - 1/2 + \sin(3\pi x)/\sin(3\pi/K) = 0$. Therefore, $g_k \geq 0$ when $k = 3$.

- (c) When $4 \leq k \leq (K-1)/2$, by $m/K \geq a_K = 1/2 + \frac{1/K}{\sin(4\pi/K)}$,

$$g_k = (m - K/2) - \frac{1}{\sin(k\pi/K)} \geq (m - K/2) - \frac{1}{\sin(4\pi/K)}$$
$$= K[m/K - 1/2 - \frac{1/K}{\sin(4\pi/K)}] \geq 0.$$

Combining these, $M$ is non-negative. Since the first column of $Q$ is $c_{K,0}$, $M$ is doubly stochastic by Lemma A.1. This proves Theorem 9.

## E   PROOF OF THEOREMS 10-11

In this section, we prove Theorems 10-11.

### E.1   PROOF OF THEOREM 10

In this case, we assume $4 \mid K$. By Theorem 3, Problem (1) is solvable

when $m \leq K/2$ and when $m \geq K - 2$ (note that $4|K$ so it is even in this case).

Therefore, we assume

$$K/2 + 1 \leq m \leq K - 3. \tag{E.24}$$

In this case, $a_{K,m} = 2m - K > 0$ and

$$J_{K,m} = \text{diag}(1 + 2m - K, 1, \ldots, 1, -1, \ldots, -1).$$

We consider two cases:

- Case 1. $m$ is even.
- Case 2. $m$ is odd.

Consider Case 1. Let

$$n = (m - K/2)/2.$$

By (E.24),

$$n \geq 1 \text{ and } n \text{ is an integer.}$$

Let

$$Q_0 = [c_{K,0}, Q_1, Q_2] \in \mathbb{R}^{K,m+1},$$

where

$$Q_2 = [h_{K,1}, h_{K,2}, \ldots, h_{K,K/2}] \in \mathbb{R}^{K,K/2},$$

and $Q_1 \in \mathbb{R}^{K,m-(K/2)}$ is the matrix where (a) the first two rows $Q_1$ equal to each other, the next two rows of $Q_1$ equal to each other, and so on and so forth, and (b) row $1, 3, \ldots, K-1$ of $Q_1$ equal to row $1, 2, \ldots, K/2$ of

$$M = \sqrt{(1/2)} \cdot [c_{K/2,1}, s_{K/2,1}, \ldots, c_{K,n}, s_{K,n}]. \tag{E.25}$$

respectively. Here, note that $2n = m - K/2 \leq K/2$, and that each diagonal entry of $MM'$ is

$$(1/2)\frac{2}{K/2}n = 2n/K = (m - K/2)/K.$$

By (A.1)-(A.2) and basic algebra,

- $c_{K,0}$ is orthogonal to any columns of $Q_1$ and any columns of $Q_2$.
- $Q_1 \in \mathbb{R}^{K,2n}$ and $Q_1'Q_1 = 2M'M = I_{2n}$.
- $Q_2 \in \mathbb{R}^{K,K/2}$ and $Q_2'Q_2 = I_{K/2}$.
- Fixing a column of $Q_1$ and a column of $Q_2$, they are orthogonal to each other.

Combining these,
$$Q_0'Q_0 = I_{m+1}.$$
Therefore, there is $G \in \mathbb{R}^{K,K-m-1}$ such that the matrix
$$Q = [Q_0, G] = [c_{K,0}, c_{K,1}, s_{K,1}, \ldots, c_{K,n}, s_{K,n}, h_{K,1}, h_{K,2}, \ldots, h_{K,K/2}, G] \qquad \text{(E.26)}$$
is a $K \times K$ orthogonal matrix.

Next, recall that
$$J_{K,m} = \text{diag}(1 + a_{K,m}, 1, \ldots, 1, -1, \ldots, -1).$$
It follows that
$$Q J_{K,m} Q' = a_{K,m} s_{K,0} s_{K,0}' + I_K - 2Q_1 Q_1' - 2Q_2 Q_2' = (I) + (II), \qquad \text{(E.27)}$$
where
$$(I) = (a_{K,m} s_{K,0} s_{K,0}' - 2Q_1 Q_1'.$$
and
$$(II) = I_K - 2Q_2 Q_2'.$$

Consider (I). Note that by construction,

- $a_{K,m} s_{K,0} s_{K,0} = (a_{K,m}/K) \mathbf{1} \mathbf{1}_K'.$
- $Q_1' Q_1 = I_{(K/2-m)}$ and every diagonal entry of $Q_1 Q_1'$ is $2n/K$.

Recall that $a_{K,m} = 2m - K$ and $n = (m - K/2)/2$. It follows that
$$a_{K,m}/K = (2m - K)/K, \qquad 2n/K = (m - K/K)/K = (1/2)(2m - K)/K.$$
Applying Lemma A.2 with $a = (2m - K)/K$ and $H = Q_1$, it follows that
$$(I) \text{ is a non-negative matrix.} \qquad \text{(E.28)}$$
At the same time, by (A.3),
$$(II) \text{ is a traceless non-negative matrix.} \qquad \text{(E.29)}$$
Inserting (E.28)-(E.29) into (E.27), $Q J_{K,m} Q'$ is a traceless non-negative matrix. Finally, since the first column of $Q$ is $s_{K,0}$, the matrix $Q J_{K,m} Q'$ is doubly stochastic matrix by Lemma A.1. This proves Case 1.

Consider Case 2. In this case, $m$ is odd and $m \geq (K/2) + 1$. Let
$$n = (m - K/2 - 1)/2.$$
It is seen $n \geq 0$. Let
$$Q_0 = [c_{K,0}, Q_1, Q_2] \in \mathbb{R}^{K,m+1},$$
where $Q_2 \in \mathbb{R}^{K,K/2}$ is as the same as in Case 1, and
$$Q_1 = [s_{K,0}, Q_1^*] \in \mathbb{R}^{K,m-(K/2)}$$
for a matrix $Q_1^* \in \mathbb{R}^{K,2n}$; note that $2n = m - (K/2) - 1$. We construct $Q_1^*$ in a way such that (a) the first two rows equal to each other, the next two rows equal to each other, and so on and so forth, and (b) row $1, 3, \ldots, K - 1$ of $Q_1^*$ equal to row $1, 2, \ldots, K/2$ of
$$M = \sqrt{(1/2)} \cdot [c_{K/2,1}, s_{K/2,1}, \ldots, c_{K,n}, s_{K,n}]. \qquad \text{(E.30)}$$
By similar arguments, it is seen that
$$Q_0' Q_0 = I_{m+1},$$
so there is a matrix $G \in \mathbb{R}^{K,K-m-1}$ so that
$$Q = [c_{K,0}, G, Q_1, Q_2] \qquad \text{(E.31)}$$
is a $K \times K$ orthogonal matrix. Now, we similarly write
$$Q J_{K,m} Q' = Q \text{diag}((1 + a_{K,m}), 1, \ldots, 1, -1, \ldots, -1) = (I) + (II),$$
where
$$(I) = a_{K,m} s_{K,0} s_{K,0}' - 2Q_1 Q_1'.$$
and
$$(II) = I_K - 2Q_2 Q_2', \qquad \text{which is non-negative; see (A.3).}$$
By the construction,

- $a_{K,m}s_{K,0}s'_{K,0} = (a_{K,m}/K)\mathbf{1}_K\mathbf{1}'_K$, where $a_{K,m}/K = (2m-K)/K$.
- $Q'_1 Q_1 = I_N$ with $N = m - K/2$.
- Every diagonal entry of $MM'$ is $(1/2)(2n)/(K/2) = 2n/K$, so every diagonal entry of $Q'_2 Q_2$ is $(2n+1)/K$, where $(2n+1)/K = [(m-K/2-1)+1)]/K = (1/2)(2m-K)/K$.

Applying Lemma A.2 with $a = (2m-K)/K$ and $H = Q_1$, the matrix (I) is also non-negative. Combining these gives the claim of Case 2 and completes the proof of Theorem 10.

### E.2 Proof of Theorem 11

We now prove Theorem 11. In this case, $K$ is even but $4 \nmid K$, and $m \neq (K/2) + 1$. By Theorems 2 and Theorem 5, Problem (1) is solvable when $m \leq K/2$ and $m \geq K - 2$. Combining these, we only need to consider the case where

$$(K/2) + 2 \leq m \leq K - 3. \tag{E.32}$$

We consider two cases.

- Case 1. $m$ is odd.
- Case 2. $m$ is even. In this case, since $K$ is even, it follows from (E.32) that $m \leq K - 4$.

Consider Case 1. Let $n = (m - (K/2))/2$. Note that in our case, $K/2$ is odd, so $(m - K/2)$ is even. Since $m > K/2$, $n \geq 1$. Also, since $m \leq K - 3$, $n \leq (K-6)/4$, and $2n \leq (K/2) - 3$. Let

$$M = (1/\sqrt{2})[c_{K/2,1}, s_{K/2,1}, \ldots, c_{K/2,n}, s_{K/2,n}].$$

Similarly as in the proof of Theorem 10, we construct a matrix $Q_1 \in \mathbb{R}^{K,2n}$ in a way such that (a) the first two rows $Q_1$ equal to each other, the next two rows of $Q_1$ equal to each other, and so on and so forth, and (b) row $1, 3, \ldots, K - 1$ of $Q_1$ equal to row $1, 2, \ldots, K/2$ of $M$, respectively. Also, similarly, we let

$$Q_2 = [h_{K,1}, h_{K,2}, \ldots, h_{K,K/2}],$$

and

$$Q_0 = [c_{K,0}, Q_1, Q_2] \in \mathbb{R}^{m+1}.$$

It is seen $Q'_0 Q_0 = I_{m+1}$. Since $m + 1 < K$, there is a matrix $G \in \mathbb{R}^{K,K-m-1}$ such that

$$Q = [c_{K,0}, G, Q_1, Q_2]$$

is an orthogonal matrix. Similarly, we write

$$Q J_{K,m} Q' = a_{K,m}s_{K,0}s'_{K,0} + I_K - 2Q_1 Q'_1 - 2Q_2 Q'_2 = (I) + (II), \tag{E.33}$$

where

$$(I) = (a_{K,m}s_{K,0}s'_{K,0} - 2Q_1 Q'_1.$$

and

$$(II) = I_K - 2Q_2 Q'_2, \qquad \text{which is non-negative; see (A.3).}$$

Moreover, by similar argument as before,

- $a_{K,m}s_{K,0}s_{K,0} = ((2m-K)/K)\mathbf{1}_K\mathbf{1}'_K$ (by definition and that $m > K/2$),
- $Q'_1 Q_1 = I_N$ with $N = m - (K/2)$,
- For every row of $Q_1$, the square $\ell^2$-norm is $2n/K$, where $(2n)/K = (m - K/2)/K = (1/2)(2m-K)/K$.

Applying Lemma A.2 with $a = (2m-K)/K$ and $H = Q_1$, it follows that $(I)$ is a non-negative matrix. This proves the claim in Case 1.

Consider Case 2. In this case, $m$ is even. By (E.32) and that $K/2$ is odd,

$$m - (K/2) \geq 3.$$

Let

$$K = 2N = 4s + 2, \qquad n = (m - (K/2) - 1)/2, \qquad \text{so } n \text{ is an integer and } n \geq 1.$$

Note that

$$N = 2s + 1 \qquad \text{which is an odd number.}$$

Since $m \leq \frac{3K}{4}$, and $m$ is even, we must have

$$m \leq \begin{cases} \frac{3K-6}{4}, & \text{if } (3K-2)/4 \text{ is odd,} \\ \frac{3K-2}{4}, & \text{if } (3K-2)/4 \text{ is even,} \end{cases}$$

Since $(3K - 2)/4 = 3s + 1$, this is equivalent to

$$m \leq \begin{cases} 3s, & \text{if } s \text{ is even,} \\ 3s + 1, & \text{if } s \text{ is odd.} \end{cases}$$

It follows

$$2n \leq (m - K/2 - 1) \leq \begin{cases} s - 2, & \text{if } s \text{ is even,} \\ s - 1, & \text{if } s \text{ is odd,} \end{cases}$$

Recall that $N = 2s + 1$. This is equivalent to

$$2n \leq \begin{cases} (N-5)/2 = [(N-1)/2] - 2, & \text{if } (N-1)/2 \text{ is even,} \\ (N-3)/2 = [(N-1)/2] - 1, & \text{if } (N-1)/2 \text{ is odd.} \end{cases} \tag{E.34}$$

Let

$$M_0^* = [q_0, q_1, q_2, \ldots, q_{2n}] \in \mathbb{R}^{N,2n+1},$$

where for $1 \leq j \leq n$,

$$q_0 = \begin{bmatrix} \mathbf{1}_{(N+1)/2} \\ -\mathbf{1}_{(N-1)/2} \end{bmatrix}, \qquad q_{2j-1} = \begin{bmatrix} c_{(N+1)/2,j} \\ c_{(N-1)/2,j} \end{bmatrix}, \qquad q_{2j} = \begin{bmatrix} s_{(N+1)/2,j} \\ s_{(N-1)/2,j} \end{bmatrix}.$$

We partition $M^*$ as

$$M_0^* = \begin{bmatrix} \mathbf{1}_{(N+1)/2} & B_1 \\ -\mathbf{1}_{(N-1)/2} & B_2 \end{bmatrix}.$$

Note that especially, by (A.1) and (E.34),

$$B_1' B_1 = B_2' B_2 = I_{2n}.$$

For

$$c_0 = \sqrt{\frac{N-1}{N(N+1)}}, \qquad d_0 = \sqrt{\frac{N+1}{N(N-1)}},$$

and

$$c = \sqrt{\frac{1}{N}\left(\frac{N+1}{2} + \frac{1}{2n}\right)}, \qquad d = \sqrt{\frac{1}{N}\left(\frac{N-1}{2} - \frac{1}{2n}\right)},$$

we let

$$M_0 = \begin{bmatrix} a\mathbf{1}_{(N+1)/2} & cB_1 \\ -b\mathbf{1}_{(N-1)/2} & dB_2 \end{bmatrix}, \qquad \text{and} \qquad M = (1/\sqrt{2})M. \tag{E.35}$$

Next, we construct (since $(2n + 2 + K/2) = (m - K/2 - 1) + 2 + K/2 = m + 1$)

$$Q_0 = [c_{K,0}, Q_1, Q_2] \in \mathbb{R}^{K,m+1},$$

where $Q_2$ is the same as in Case 1, and $Q_1$ is constructed so that (a) the first two rows $Q_1$ equal to each other, the next two rows of $Q_1$ equal to each other, and so on and so forth, and (b) row $1, 3, \ldots, K - 1$ of $Q_1$ equal to row $1, 2, \ldots, K/2$ of $M$, respectively. Note that since $Q_1$ has $(2n + 1)$ columns with $2n + 1 = (m - K/2)$, $Q_0$ has

$$1 + (m - K/2) + K/2 = (m + 1)$$

columns, so

$$Q_0 = \mathbb{R}^{K,m+1}.$$

By Lemma 7,
$$[c_{N,0}, M_0]'[c_{N,0}, M_0] = I_{2n+2}.$$
Combining this with our construction,
$$Q_0'Q_0 = I_{m+1}.$$
Since in our range of interest, $(m + 1) < K$, so there is a matrix $G \in \mathbb{R}^{K,K-m-1}$ so that
$$Q = [c_{K,0}, G, Q_1, Q_2]$$
is a $K \times K$ orthogonal matrix. Note that

- $a_{K,m}s_{K_0}s_{K,0} = (a_{K,m}/K)\mathbf{1}_K\mathbf{1}_K'$,
- For each row of $Q_1$, the square $\ell^2$-norm is $(2n + 1)/K$, where
$$(2n + 1)/K = (m - K/2) = (1/2)(2m - K).$$

Applying Lemma A.2 with $a = (2m - K)/K$ and $H = Q_1$, the matrix
$$a_{K,m}s_{K_0}s_{K,0} - 2Q_1Q_1'$$
is non-negative. The remaining part of the proof is very similar to that in Case 1 so is omitted. This proves that $QJ_{K,m}Q'$ is doubly stochastic in Case 2.

## F   PROOF OF THEOREM 12

In this setting, we assume
$$m_1/m_2 = K_1/K_2, \qquad K_1/2 < m_1 \le K_1 - 1, \qquad K_2/2 < m_2 \le K_2 - 1.$$
Given
$$A = \begin{bmatrix} A_1 & \rho u_1 v_1' \\ \rho v_1 u_1' & A_2 \end{bmatrix}, \qquad \text{where } \rho = \sqrt{(2m_1 - K_1)(2m_2 - K_2)}, \tag{F.36}$$
where $A_1 \in \mathbb{R}^{K_1,K_1}$ is a traceless doubly stochastic matrix with spectrum $\sigma(A_1) = \{(2m_1 - K_1 + 1, 1, \dots, 1, -1, \dots, -1\}$ (where exactly $m_1$ of them are $-1$), $A_2 \in \mathbb{R}^{K_2,K_2}$ is traceless doubly stochastic matrix with spectrum $\sigma(A_2) = \{(2m_2 - K_2 + 1, 1, \dots, 1, -1, \dots, -1\}$ (where exactly $m_2$ of them are $-1$), and $u_1 = (K_1)^{-1/2}\mathbf{1}_{K_1}$ and $u_1 = (K_2)^{-1/2}\mathbf{1}_{K_2}$ are the Perron eigenvector of $A_1$ and $A_2$, respectively (the Perron roots of $A_1$ and $A_2$ are $(2m_1 - K_1 + 1)$ and $(2m_2 - K_2 + 1)$, respectively. Let
$$K = K_1 + K_2, \qquad m = m_1 + m_2.$$
All we need to show are

- (a) The spectrum of $A$ are $(1 + (2m - K), 1, \dots, 1, -1, \dots, -1)$, where we have exactly $m$ of $-1$'s.
- (b) $A$ is doubly stochastic.

Consider (a). By Fiedler (1974), let $\gamma_1$ and $\gamma_2$ (assuming $\gamma_1 > \gamma_2$) be the eigenvalues of the $2 \times 2$ matrix
$$C = \begin{bmatrix} (2m_1 - K_1 + 1) & \rho \\ \rho & (2m_2 - K_2 + 1) \end{bmatrix}, \qquad \rho = \sqrt{(2m_1 - K_1)(2m_2 - K_2)},$$
then the spectrum of $A$ is
$$\sigma(A) = (\gamma_1, \gamma_2, 1, \dots, 1, -1, \dots, -1),$$
where we have exactly $m$ of $-1$'s. By basic algebra,
$$\gamma_2 = 1, \qquad \gamma_2 = (2m_1 - K_1 + 1) + (2m_2 - K_2 + 1) - 1 = (2m - K + 1).$$
Therefore, the spectrum of $A$ is
$$\sigma(A) = (2m - K + 1, 1, \dots, 1, -1, \dots, -1),$$

where we have exactly $m$ of $-1$'s. This proves (a).

Consider (b). Note that the Perron root of $A_1$ is $(2m_1 - K_1 + 1) > 1$, with a multiplicity of 1. By Lemma A.1, the Perron eigenvector $u_1$ of $A_1$ is

$$u_1 = (1/\sqrt{K_1})(1, 1, \ldots, 1)'.$$

Similarly, the Perron eigenvector $u_2$ of $A_2$ is

$$u_2 = (1/\sqrt{K_2})(1, 1, \ldots, 1)'.$$

Therefore, $A$ is non-negative and

$$A_1 \mathbf{1}_{K_1} = (2m_1 - K_1 + 1)\mathbf{1}_{K_1}, \qquad A_1 \mathbf{1}_{K_1} = (2m_1 - K_1 + 1)\mathbf{1}_{K_1}.$$

Also, note that

$$\rho = \sqrt{(2m_1 - K_1)(2m_2 - K_2)}, \qquad \rho u_1 v_1' \mathbf{1}_{K_2} = \rho\sqrt{K_2/K_1}\mathbf{1}_{K_1}, \qquad \rho v_1 u_1' \mathbf{1}_{K_1} = \rho\sqrt{K_1/K_2}\mathbf{1}_{K_2}.$$

Combining these, we have

$$A\mathbf{1}_K = \left[ \begin{array}{c} A_1 \mathbf{1}_{K_1} + \rho u_1 v_1' \mathbf{1}_{K_2} \\ \rho v_1 u_1' \mathbf{1}_{K_1} + A_1 \mathbf{1}_{K_2} \end{array} \right] = \left[ \begin{array}{c} x_1 \mathbf{1}_{K_1} \\ x_2 \mathbf{1}_{K_2} \end{array} \right],$$

where

$$x_1 = (2m_1 - K_1 + 1) + \sqrt{2m_1 - K_1)(2m_2 - K_2)}\sqrt{K_2/K_1},$$

and

$$x_2 = (2m_2 - K_2 + 1) + \sqrt{(2m_1 - K_1)(2m_2 - K_2)}\sqrt{K_1/K_2}.$$

By basic algebra,

$$\frac{x_2 - x_1}{\sqrt{(2m_1 - K_1)(2m_2 - K_2)}} = \sqrt{\frac{2m_2 - K_2}{2m_1 - K_1}} - \sqrt{\frac{2m_1 - K_1}{2m_2 - K_2}} + \sqrt{K_1/K_2} - \sqrt{K_2/K_1}. \tag{F.37}$$

By $m_1/m_2 = K_1/K_2$, there is a number $a > 0$ such that

$$a = m_1/m_2 = K_1/K_2.$$

It follows that the RHS of (F.37) is

$$\sqrt{1/a} - \sqrt{a} + \sqrt{a} - \sqrt{1/a} = 0.$$

Therefore, $x_1 = x_2$ and $A$ is doubly stochastic. This proves the claim.

## G    Proof of Lemma 13

In this case, $m > K/2$, so $a_{K,m} = \max\{0, 2m - K\} = 2m - K$ and

$$Q = [c_{K,0}, Q_1, Q_2], \qquad J_{K,m} = \text{diag}((1 + 2m - K), 1, \ldots, 1, -1, \ldots, -1).$$

It follows

$$QJ_{K,m}Q' = (2 + 2m - K)c_{K,0}c_{K,0}' + 2Q_1Q_1' - I_K. \tag{G.38}$$

Let $M = Q_1Q_1'$. Note that for any such $Q$,

$$\text{trace}(QJ_{K,m}Q') = \text{trace}(J_{K,m}) = 0.$$

Therefore, if $QJ_{K,m}Q'$ is doubly stochastic, then all diagonal entries of $QJ_{K,m}Q'$ are 0. As a result ($\delta_{ij} = 1$ if $i = j$ and 0 otherwise),

$$\text{rank}(M) = K - m - 1, \tag{G.39}$$

$$\lambda_{\max}(M) \leq 1, \quad \mathbf{1}_K' M \mathbf{1}_K = 0, \tag{G.40}$$

$$M_{ii} = (K - m - 1)/K, \quad M_{ij} \geq -(2m - K + 2)/(2K), \quad 1 \leq i \neq j \leq K. \tag{G.41}$$

This proves one direction of the lemma.

At the same time, suppose there is a matrix $M \in \mathbb{R}^{K,K}$ satisfying (G.39)-(G.41). By $\text{rank}(M) = (K - m - 1)$ and $\text{trace}(M) = (K - m - 1) \geq (K - m - 1)\lambda_{\max}(M)$, $M$ has exactly $(K - m - 1)$ nonzero eigenvalues that are all 1. Therefore, $M$ is a projection matrix and there is a $Q_1 \in \mathbb{R}^{K,K-m-1}$ such that $M = Q_1Q_1'$ and $Q_1'Q_1 = I_{K-m-1}$. Combining this with $\mathbf{1}_K' M \mathbf{1}_K = 0$, $Q_1 \mathbf{1}_K = 0$. Therefore, there is a matrix $Q_2 \in \mathbb{R}^{K,m}$ such that $Q = [c_{K,0}, Q_1, Q_2]$ is a $K \times K$ orthogonal matrix. Combining (G.41) with (G.38), it is seen that

$$QJ_{K,m}Q'$$

is non-negative. Since the first column of $Q$ is $c_{K,0}$, $QJ_{K,m}Q'$ is doubly stochastic. This proves the claim in the another direction and completes the proofs.

## H    Proof of Theorem 14

First, note that $a^*_{K,m} \geq a_{K,m}$. Otherwise, if $a^*_{K,m} < a_{K,m}$, the by definition, $a^*_{K,m} = 0$ and $a_{K,m} = \max\{0, 2m - K\}$, so we must have $m > K/2$ and $a^*_{K,m} < (2m - K)$. Therefore, at $a = a^*_{K,m}$, for any orthogonal matrix $Q \in \mathbb{R}^{K,K}$, $\mathrm{trace}(QJ_{K,m,a}Q') = \mathrm{trace}(J_{K,m,a}Q') < 0$ and so $QJ_{K,m,a}Q'$ can not be a non-negative matrix. This proves $a^*_{K,m} \geq a_{K,m}$.

It remains to show $a^*_{K,m} \leq \tilde{a}_{K,m}$ and we can write $\Omega = \Theta \Pi P \Pi' \Theta$ for $(\Theta, \Pi, P)$ as in the DCMM model. By Jin (2022) and Lemma A.1, all we need to show is when $a \geq \tilde{a}_{K,m}$, there is a $K \times K$ orthogonal matrix $Q \in \mathbb{R}^{K,K}$ such that

$$\text{the first column of } Q \text{ is } c_{K,0} \equiv (1/\sqrt{K})\mathbf{1}_K \text{ and } QJ_{K,m,a}Q' \text{ is non-negative.} \qquad \text{(H.42)}$$

The claim for the Case (S1)-(S9) follows directly from our results in Section 2, so we only need to show the claim for Case (N1), Case (N2), Case (U1), and Case (U2). The case (U2) is relatively long, so we further split. Throughout this section, let $N$ be the largest integer such that

$$4N \leq K.$$

Note that in Case (U2), we must have

$$K \text{ is odd, } K \geq 5, \text{ and } K/2 < m < (3K/4) - 1,$$

so we further divide (U2) as two sub-cases,

- (U2a): $K = 4N + 1$, $N \geq 2$, $K/2 < m < (3K/4) - 1$ and $m < K - 2$.
- (U2b): $K = 4N + 3$, $N \geq 1$, $K/2 < m < (3K/4) - 1$ and $m < K - 2$.

In (U2a), we assume $N \geq 2$. The reason is that when $N = 1$, $K = 5$, where Problem (1) is solvable for $(K, m) = (5, 1), (5, 2), (5, 4)$ and is not solvable when $(K, m) = (5, 3)$, but the last case is covered in Case (N). Therefore, we assume $N \geq 2$ in (U2a). We consider the four cases: (N1), (N2), (U1), (U2a), and (U2b) separately in the sections below.

### H.1    Proof for Theorem 14 for the Case (N1)

In this case,

$$K \text{ is odd, } K \geq 5, \ m = K - 2, \text{ and } a \geq \tilde{a}_{K,m} = K - 4K/(K+1), \qquad \text{(H.43)}$$

and the goal is to construct a matrix $Q$ such that (H.42) holds. Let

$$Q_0 = [c_{K,0}, q],$$

where

$$q = \begin{bmatrix} c\mathbf{1}_{(K+1)/2} \\ d\mathbf{1}_{(K-1)/2} \end{bmatrix}, \qquad c = \sqrt{\frac{(K-1)}{K(K+1)}}, \qquad d = -\sqrt{\frac{K+1}{K(K-1)}}.$$

It is seen that $Q_0'Q_0 = I_2$, so there is a matrix $G \in \mathbb{R}^{K,K-2}$ such that

$$Q = [c_{K,0}, q, G]$$

is a $K \times K$ orthogonal matrix. For any $a \geq \tilde{a}_{K,m}$, write

$$QJ_{K,m,a}Q' = (a+2)c_{K,0}c_{K,0}' + 2qq' - I_K = (I) + (II).$$

Note that all diagonal entries of the LHS are no smaller than

$$(a+2)/K + 2c^2 - 1 \geq (\tilde{a}_{K,m} + 2)/K + \frac{2(K-1)}{K(K+1)} - 1 \geq 0,$$

and all off-diagonals of the LHS are no smaller than

$$(a+2)/K - 2/K \geq (\tilde{a}_{K,m} + 2 - 2)/K \geq \tilde{a}_{K,m}/K \geq 0.$$

Note that the first column of $Q$ is $c_{K,0}$. Combining this gives the claim.

## H.2 Proof for Theorem 14 for the Case (N2)

In this case, $(K, m) = (7, 4)$ and $a_{K,m} = 1$. Let $a_0 = 4\cos(\pi/7) - 2$ and

$$J_0 \equiv J_{K,m,a_0} = \text{diag}(1 + a_0, 1, 1, -1, -1, -1).$$

By similar argument as above, it is sufficient to show that there is a $7 \times 7$ orthogonal matrix of the form

$$Q = [c_{K,0}, Q_1, Q_2], \qquad \text{where } Q_1 \in \mathbb{R}^{7,2}, Q_2 \in \mathbb{R}^{7,4},$$

such that

$$QJ_0Q'$$

is entry-wise non-negative.

To show the claim, let

$$Q_1 = [c_{K,1}, s_{K,1}].$$

It follows

$$QJ_0Q' = (2 + a_0)c_{K_0}c'_{K,0} + 2Q_1Q'_1 - I_K \equiv M.$$

Now, first, since $a_0 \approx 1.6 > 1$, all diagonal entries of $M$ on the RHS are

$$(2 + a_0)/7 + (4/7) - 1 = (2 + a_0 - 3)/7 \ge 0.$$

Second, by definition, for any $1 \le i \ne j \le 7$,

$$M(i, j) = (2 + a_0)/7 + (4/7)\cos(2|i - j|\pi/K) = (1/7)[2 + a_0 + 4\cos(2|i - j|\pi/K)],$$

where the minimum is achieved when $|i - j| = 3$ or $|i - j| = 4$, with the same value of

$$(1/7)[2 + a_0 + 4\cos(\pi/7)] = 0.$$

This proves the claim.

## H.3 Proof of Theorem 14 for the case of (U1)

In this case,

$$K = 4N + 2, \quad N \ge 1, \quad m = (K/2) + 1 = 2N + 2, \quad \tilde{a}_{K,m} = \frac{2(K + 2)}{(K - 2)},$$

and the goal is to show (H.42) holds for all $a \ge \tilde{a}_{K,m}$. Let

$$Q_0 = [c_{K,0}, q, Q_2], \qquad Q_2 = [h_{K,1}, \ldots, h_{K,K/2}],$$

where $h_{K,j}$ is as in the first page of the supplement, and

$$q = \begin{bmatrix} c\mathbf{1}_{(K+2)2} \\ d\mathbf{1}_{(K-2)/2} \end{bmatrix}, \qquad c = \sqrt{\frac{(K - 2)}{K(K + 2)}}, \qquad d = -\sqrt{\frac{K + 2}{K(K - 2)}}.$$

It is seen that $Q'_0Q_0 = I_{m+1}$, so there is a matrix $G \in \mathbb{R}^{K, K-m-1}$ such that

$$Q = [c_{K,0}, G, q, Q_2]$$

is a $K \times K$ orthogonal matrix. For any $a > 0$, write

$$QJ_{K,m,a}Q' = (ac_{K,0}c'_{K,0} - 2qq') + (I_K - Q_2Q'_2) = (I) + (II).$$

First, by (A.3), $(II)$ is non-negative. Second, recall that $a \ge \tilde{a}_{K,m} = 2(K + 2)/[(K - 2)]$. It follows that the smallest entry of $(I)$ is no smaller than

$$a/K - 2d^2 \ge \frac{2(K + 2)}{K(K - 2)} - \frac{2(K + 2)}{K(K - 2)} = 0,$$

so $(I)$ is also non-negative. Also, note that the first column of $Q$ is $c_{K,0}$. Combining these gives the claim.

### H.4 Proof of Theorem 14 for the case of (U2a)

In this case,

$$K = 4N + 1, \quad N \geq 2, \quad \frac{K}{2} < m < \frac{3K}{4} - 1, \quad \tilde{a}_{K,m} = \frac{K}{K-1}(2m - K - 1) + \frac{2K}{\sqrt{K-1}}.$$

By elementary algebra, we have

$$2N + 1 \leq m \leq 3N - 1, \qquad (m - 1) \geq (K - 1)/2. \tag{H.44}$$

The goal is to show that (H.42) holds for any $a \geq \tilde{a}_{K,m}$.

Now, when $m$ is odd, (so $m - 1$ is even), let (see the first page of the supplement for definition of $c_{K,i}$)

$$2n = (m - 1) - 2N, \qquad M = [c_{2N,1}, s_{2N,1}, \ldots, c_{2N,n}, s_{2N,n}] \in \mathbb{R}^{2N,2n}.$$

and when $m$ is even, let

$$2n = (m - 2) - 2N, \qquad M = [s_{2N,0}, c_{2N,1}, s_{2N,1}, \ldots, c_{2N,n}, s_{2N,n}] \in \mathbb{R}^{2N,2n+1}.$$

Combining this with (H.44) and recalling $N \geq 2$,

$$n \leq (m - 1)/2 - N \leq (N/2) - 1 \leq N - 2.$$

Therefore, in both cases,

$$c_{2N,n+1} \text{ is well-defined in our range of interest and is orthogonal to all columns of } M. \tag{H.45}$$

Also, in both cases, let $Q_1 \in \mathbb{R}^{K,(m-1)-2N}$ be the matrix such that (a) the first two rows of $Q_1$ equal to each other, the next two rows of $q$ equal to each other, and so on and so forth, and (b) row $1, 3, 5, \ldots, 4N - 1$ of $Q_1$ equals to row $1, 2, \ldots, 2N$ of $M$, respectively.

At the same time, define a vector $q \in \mathbb{R}^{4N}$ so that (a) the first two rows of $q$ equal to each other, the next two rows of $q$ equal to each other, and so on and so forth, and (b) row $1, 3, 5, \ldots, 4N - 1$ of $q$ equals to

$$u = -\frac{\sqrt{2}}{8N} + x_0 c_{2N,n+1} \in \mathbb{R}^{2N}, \qquad \text{where } x_0 > 0 \text{ and } x_0^2 = \tfrac{1}{4}(1 - \tfrac{1}{2N}).$$

Let (see the first page of the supplement for definition of $h_{N,i}$)

$$Q_2 = [h_{4N,1}, h_{4N,2}, \ldots, h_{4N,2N}] \in \mathbb{R}^{4N,2N},$$

Also, let

$$Q_1^* = \begin{bmatrix} Q_1 \\ 0 \end{bmatrix} \in \mathbb{R}^{K,m-1-2N}, \qquad Q_2^* = \begin{bmatrix} Q_2 & q \\ 0 & \sqrt{2}/2 \end{bmatrix} \in \mathbb{R}^{K,2N+1},$$

and

$$Q_0^* = [c_{K,0}, Q_1^*, Q_2^*],$$

We have the following observations.

- $\|q\|^2 = 2\|u\|^2 = 1/2$.
- $\|q\|_\infty \leq 1/\sqrt{2N}$.
- The sum of all rows of $q$ is $-1/\sqrt{2}$, so the last column of $Q_2^*$ is orthogonal to $c_{K,0}$.
- The last column of $Q_2^*$ is orthogonal to all other columns of $Q_2^*$, and is also orthogonal to all columns of $Q_1^*$.
- As a result, $Q_0^{*\top} Q_0^* = I_{m+1}$.

Now, let

$$Q^* = [Q_0^*, G^*],$$

where $G^* \in \mathbb{R}^{K-m-1}$ is any matrix so that $Q^*$ is a $K \times K$ orthogonal matrix. For any $a \geq \tilde{a}_{K,m}$, write

$$Q^* J_{K,m,a}(Q^*)' = (I) + (II) + (III),$$

where

$$(I) = ac_{K,0}c'_{K,0} - a_{K-1,m-1}c_{K-1,0}c'_{K-1,0},$$
$$(II) = (a_{K-1,m-1}c_{K-1,0}c'_{K-1,0} - 2Q_1^*(Q_1^*)'),$$

and

$$(III) = (I_K - 2Q_2^*(Q_2^*)').$$

Note that by (H.44), $m - 1 \geq (K-1)/2$, so

$$a_{K-1,m-1} = 2m - K - 1.$$

Now, first, for all $a \geq \tilde{a}_{K,m}$, all entries of $(I)$ are no smaller than

$$a/K - a_{K-1,m-1}/K - 1) \geq \tilde{a}_{K,m}/K - (2m - K - 1)/(K - 1).$$

Second, by the same argument as in the proof of Theorem 10,

$$(II) \text{ is non-negative.}$$

Second, by construction,

$$(III) = \begin{bmatrix} -2qq' & -\sqrt{2}q' \\ -\sqrt{2}q & 0 \end{bmatrix}.$$

Recall that $\|q\|_\infty \leq 1/\sqrt{2N}$ and $N \geq 2$. It follows that all entries of $(III)$ are no smaller than

$$-1/\sqrt{N} = -2/\sqrt{K-1}.$$

Combining all these, all entries of $QJ_{K,m,a}Q'$ are no smaller than

$$\tilde{a}_{K,m}/K - (2m - K - 1)/(K - 1) - 2/\sqrt{K-1} = 0.$$

Combining these, $QJ_{K,m,a}Q'$ is non-negative for all $a \geq \tilde{a}_{K,m}$, and the claim follows.

## H.5 Proof of Theorem 14 for the case of (U2b)

In this case,

$$K = 4N + 3, \quad N \geq 2, \quad K/2 < m < (3K/4) - 1, \quad m < (K - 2).$$

and

$$\tilde{a}_{K,m} = \frac{K}{K-1}(2m - K - 1) + \frac{2K}{\sqrt{K-1}}.$$

By elementary algebra, we have

$$2N + 2 \leq m \leq 3N + 1, \qquad (m - 1) \geq (K - 1)/2. \tag{H.46}$$

and the goal is to show (H.42) holds for all $a \geq a_{K,m}^*$. We consider three cases:

- (a). $N \geq 2$ and $m \neq (2N + 3)$.
- (b). $(m - 1) = (K - 1)/2 + 1$ (or equivalently $m = 2N + 3$) and $N \geq 2$.

Consider Case (a). In this case, $N \geq 2$ (and so $K \geq 11$). The range of interest for $m$ in this case is

$$(2N + 2) \leq m \leq 3N + 1, \qquad m \neq 2N + 3. \tag{H.47}$$

When $m$ is even, $m - 1 - (K-1)/2 = m - 2N - 2$ is even. In this case, let $n = (m - 2N - 2)/2$ and

$$M = [c_{2N+1,1}, s_{2N+1,1}, \ldots, c_{2N+1,n}, s_{2N+1,n}] \in \mathbb{R}^{2N+1,2n}. \tag{H.48}$$

By (H.46), $m \leq 3N + 1$. Since $m$ is even, we have $m \leq 3N$ in the special case of $N = 2$. Combining this with (H.46) and recalling $N \geq 3$,

$$n \leq m/2 - N - 1 \leq [(3N + 1)/2] - N - 1 \leq (N - 1)/2 \leq N - 2$$

and when $N = 2$,

$$n \leq m/2 - N - 1 \leq 3N/2 - N - 1 = (N - 2)/2 \leq N - 2.$$

Therefore,

$c_{2N+1,n+1}$ is well-defined in our range of interest and is orthogonal to all columns of $M$. (H.49)

Let $u \in \mathbb{R}^{2N+1}$ be the vector

$$u = -\frac{\sqrt{2}}{4(2N+1)} + x_0 \cdot c_{2N+1,n+1}, \qquad x_0 > 0 \text{ and } x_0^2 = \frac{1}{4}(1 - \frac{1}{2(2N+1)}). \qquad (\text{H.50})$$

When $m$ is odd, then $(m - 1 - (K - 1)/2 = m - 2N - 2$ is odd. In this case, let $n = (m - 2N - 3)/2$. By (H.47),

$$n \geq 1.$$

Let

$$M_0^* = [q_0, q_1, q_2, \ldots, q_{2n}] \in \mathbb{R}^{N, 2n+1},$$

where for $1 \leq j \leq n$,

$$q_0 = \begin{bmatrix} \mathbf{1}_{N+1} \\ -\mathbf{1}_N \end{bmatrix}, \qquad q_{2j-1} = \begin{bmatrix} c_{N+1,j} \\ c_{N,j} \end{bmatrix}, \qquad q_{2j} = \begin{bmatrix} s_{N+1,j} \\ s_{N,j} \end{bmatrix}.$$

Moreover, we partition $M^*$ as

$$M_0^* = \begin{bmatrix} \mathbf{1}_{N+1} & B_1 \\ -\mathbf{1}_N & B_2 \end{bmatrix}.$$

For

$$c_0 = \sqrt{\frac{N}{(N+1)(2N+1)}}, \qquad d_0 = \sqrt{\frac{N+1}{N(2N+1)}},$$

and

$$c = \sqrt{\frac{1}{2N+1}(N + 1 + \frac{1}{2n})}, \qquad d = \sqrt{\frac{1}{2N+1}(N - \frac{1}{2n})},$$

we let

$$M = \begin{bmatrix} c_0 \mathbf{1}_{N+1} & cB_1 \\ -d_0 \mathbf{1}_N & dB_2 \end{bmatrix} \in \mathbb{R}^{2N+1, 2n+1}. \qquad (\text{H.51})$$

In this case, by (H.47), $m \leq 3N + 1$. Since $m$ is odd, we have four cases (1) $m \leq 3N$ and $N$ is odd, (2) $m \leq 3N - 1$ and $N$ is even, (3) $m = 3N + 1$ and $N$ is even. In (1), $n = (m - 2N - 3)/2 \leq (N - 3)/2$ and $N$ is odd. In (2). $n = (m - 2N - 3)/2 \leq (N - 4)/2$ and $N$ is even. In (3), $n = (m - 2N - 3)/2 \leq (N - 2)/2$ and $N$ is even.

- In all three cases, $c_{N+1,n+1}$ is well-defined, and is orthogonal to all columns of $[a\mathbf{1}_{N+1}, cB_1]$.
- In the first two cases, $c_{N,n+1}$ is well-defined, and is orthogonal to all columns of $[-b\mathbf{1}_N, dB_2]$. In the last case, $s_{N,0}$ is well-defined, and is orthogonal to all columns of $[-b\mathbf{1}_N, dB_2]$.
- Note also that the first column of $M$ is orthogonal to $\mathbf{1}_{2N+1}$ by construction.

Let

$$v_1 = \sqrt{(N+1)/2} \cdot c_{N+1,n+1}, \qquad v_2 = \begin{cases} \sqrt{(N/2)}c_{N,n+1}, & \text{in Case (1)-(2)}, \\ \sqrt{(N/2)}s_{K,0}, & \text{in Case (3)}. \end{cases}$$

Let

$$u = -\frac{\sqrt{2}}{4(2N+1)} + \frac{x_0}{\sqrt{(N+1/2)}} \cdot \begin{bmatrix} v_1 \\ v_2 \end{bmatrix}, \qquad x_0 > 0 \text{ and } x_0^2 = \frac{1}{4}(1 - \frac{1}{2(2N+1)}). \qquad (\text{H.52})$$

In summary,

- When $m$ is even, the $u$ defined in (H.50) is orthogonal to all columns of the corresponding $M$ matrix in (H.48). Also, $\|u\|^2 = 1/2$, and the sum of all entries of $u$ is 0.

- When $m$ is odd, the $u$ defined in (H.52) is orthogonal to all columns of the corresponding $M$ matrix in (H.51). Also, $\|u\|^2 = 1/2$, and the sum of all entries of $u$ is 0.

- In either cases, if we fix a column in $M$, the the sum of all entries are 0.

In either case, let $Q_1 \in \mathbb{R}^{4N+2, m-2N-2}$ be the matrix where (a) the first two rows are the same, the next two rows are the same, and so on and so forth, and (b) row $1, 3, \ldots, 4N+1$ of $Q_1$ is the same as row $1, 2, \ldots, 2N+1$ of $M$, respectively. Also in either case, let $q \in \mathbb{R}^{4N+2}$ be the vector where (a) the first two rows are the same, the next two rows are the same, and so on and so forth, and (b) row $1, 3, \ldots, 4N+1$ of $q$ is the same as row $1, 2, \ldots, 2N+1$ of $u$, respectively. Let

$$Q_2 = [h_{4N+2,1}, h_{4N+2,2}, \ldots, h_{4N+2,2N+1}].$$

Let

$$Q_1^* = \begin{bmatrix} Q_1 \\ 0 \end{bmatrix}, \quad Q_2^* = \begin{bmatrix} Q_2 & u \\ 0 & \sqrt{2}/2 \end{bmatrix}, \quad \text{and} \quad Q_0^* = [c_{K,0}, Q_1^*, Q_2^*] \in \mathbb{R}^{K, m+1}.$$

It is seen that

$$(Q_0^*)' Q_0^* = I_{m+1}.$$

Now, let

$$Q^* = [Q_0^*, G^*],$$

where $G^* \in \mathbb{R}^{K-m-1}$ is any matrix so that $Q^*$ is a $K \times K$ orthogonal matrix. For any $a \geq a_{K,m}^*$, write

$$Q^* J_{K,m,a} (Q^*)' = (I) + (II) + (III),$$

where

$$(I) = a c_{K,0} c_{K,0}' - a_{K-1,m-1} c_{K-1,0} c_{K-1,0}',$$
$$(II) = (a_{K-1,m-1} c_{K-1,0} c_{K-1,0}' - 2 Q_1^* (Q_1^*)'),$$

and

$$(III) = I_K - 2 Q_2^* (Q_2^*)'.$$

Now, first, by the proof of Theorem 10,

$$(II) \text{ is non-negative.}$$

Second, by construction,

$$(III) = \begin{bmatrix} -2qq' & -\sqrt{2}q' \\ -\sqrt{2}q & 0 \end{bmatrix},$$

where by basic algebra, $\|q\|_\infty \leq 1/\sqrt{2N+1}$. It follows that all entries of $(III)$ are no smaller than

$$-1/\sqrt{N+1/2}.$$

Last, since $(m-1) \geq (K-1)/2$ in our range of interest, $a_{K-1,m-1} = 2(m-1) - (K-1) = (2m - K - 1)$. Recall that $\tilde{a}_{K,m} = \frac{K}{K-1}(2m - K - 1) + \frac{2K}{\sqrt{K-1}} = \frac{K}{K-1} a_{K-1,m-1} + \frac{K}{\sqrt{N}}$. Therefore, for any $a \geq \tilde{a}_{K,m}$, all entries of $(I)$ are no smaller than

$$a/K - a_{K-1,m-1}/(K-1) \geq \frac{1}{K}\left[\frac{K}{K-1} a_{K-1,m-1} + \frac{K}{\sqrt{N}}\right] - \frac{a_{K-1,m-1}}{K-1} \geq \frac{1}{\sqrt{N}}.$$

Combining these, $Q J_{K,m,a} Q'$ is non-negative for all $a \geq \tilde{a}_{K,m}$ and completes the proof of Case (a).

Consider Case (b). In this case,

$$K = 4N+3, \qquad m = 2N+3, \qquad N \geq 2,$$

and

$$a_{K,m} = 3, \qquad a_{K,m}^* = \frac{2K}{K-1} + \frac{2K}{\sqrt{K-1}}.$$

Let

$$u = \begin{bmatrix} a\mathbf{1}_{2N+2} \\ -b\mathbf{1}_{2N} \end{bmatrix}, \qquad \text{where } a = \sqrt{\frac{K-3}{2(K-1)(K+1)}}, \quad b = \sqrt{\frac{K+1}{2(K-1)(K-3)}}.$$

and let $q \in \mathbb{R}^{4N+2}$ be the vector such that (a) the first two rows are the same, the next two rows are the same, and so on and so forth, and (b) the row $1, 3, 5, \ldots, 4N + 1$ or $q$ equals to the row $1, 2, \ldots, (2N + 1)$ of the vector

$$-\frac{\sqrt{2}}{4(2N+1)}\mathbf{1}_{2N+1} + x_0 c_{2N+1,1}, \text{where } x_0 > 0 \text{ and } x_0^2 = \tfrac{1}{4}(1 - \tfrac{1}{(4N+2)}).$$

We have

- all rows of $u$ sum to 0 and $\|u\|^2 = 1$.
- all rows of $q$ sum to 0 and $\|q\|^2 = 1/2$.
- $u \perp q$.
- both $u$ and $q$ are orthogonal to

Let

$$Q_2 = [h_{4N+2,1}, \ldots, h_{4N+2,2N+1}] \in \mathbb{R}^{4N+2,2N+1},$$

and

$$Q_0^* = [c_{K,0}, Q_1^*], \qquad \text{where} \qquad Q_1^* = \begin{bmatrix} u & Q_2 & q \\ 0 & 0 & 1/\sqrt{2} \end{bmatrix} \in \mathbb{R}^{K,(2N+3)};$$

note that $m = (2N + 3)$ in the current case. It is seen

$$(Q_0^*)'Q_0^* = I_{m+1},$$

so there is a matrix $G^* \in \mathbb{R}^{K,K-m-1}$ such that

$$Q^* = [c_{K,0}, G^*, Q_1^*]$$

is a $K \times K$ orthogonal matrix. Finally, for any $a \geq \tilde{a}_{K,m}$,

$$(Q^*)J_{K,m,a}Q^* = (I) + (II) + (III),$$

where

$$(I) = ac_{K,0}c'_{K,0},$$

$$(II) = -2\begin{bmatrix} uu' & 0 \\ 0 & 0 \end{bmatrix}.$$

and

$$(III) = (I_K - 2Q_2^*(Q_2^*)').$$

Now, first, by the construction, all entries of (II) are no smaller than

$$-2\frac{K+1}{K(K-3)} = -\frac{2(K+1)}{(K-1)(K-3)}.$$

Second, by the construction,

$$(II) = \begin{bmatrix} -2qq' & -\sqrt{2}q' \\ -\sqrt{2}q & 0 \end{bmatrix}$$

Therefore, all entries of (II) + (III) are no smaller than

$$-\frac{1}{K}\left[\frac{2K(K+1)}{(K-1)(K-3)} + \sqrt{2}K\|q\|_\infty\right], \tag{H.53}$$

where we have used $\|q\|_\infty^2 \leq 1/(2N+1) \leq 1/2$ when $N \geq 2$.

We now analyze (H.53). Denote the term in the bracket by $g$, and so

$$g = \frac{2K(K+1)}{(K-1)(K-3)} + \sqrt{2}K\|q\|_\infty = \frac{2K}{K-1} + \frac{8}{K-3}\frac{K}{K-1} + \sqrt{2}K\|q\|_\infty.$$

When $N = 2$, $K = 11$, and $4x_0^2 = 0.9$. Therefore, and $\|q\|_\infty^2 = \frac{1}{8(2N+1)^2} + x_0^2 \frac{2}{(2N+1)} + 2x_0 \frac{\sqrt{2}}{4(2N+1)} \sqrt{\frac{2}{2N+1}} \approx 0.137$. It can be directly verified that in this case,

$$g \leq \frac{2K}{K-1} + \frac{2K}{\sqrt{K-1}}.$$

When $N \geq 3$, $K = 4N + 3 \geq 15$, so $(K-1) \geq 14$ and $K - 3 > 8$. Therefore,

$$\frac{2K}{K-1} + \frac{8}{K-3}\frac{K}{K-1} \leq \frac{2K}{K-1} + \frac{K}{K-1} \leq \frac{2K}{K-1} + \frac{1}{2\sqrt{14}}\frac{2K}{\sqrt{K-1}}.$$

As the same time, since $x_0 \leq 1/2$ and $N \geq 2$,

$$\begin{aligned}
\|q\|_\infty^2 &= \frac{1}{8(2N+1)^2} + x_0^2 \frac{2}{(2N+1)} + 2x_0 \frac{\sqrt{2}}{4(2N+1)} \sqrt{\frac{2}{2N+1}} \\
&\leq \frac{1}{8(2N+1)^2} + \frac{1}{4}\frac{2}{2N+1} + 2\frac{1}{2}\frac{\sqrt{2}}{4(2N+1)} \sqrt{\frac{2}{2N+1}} \\
&= \frac{1}{2N+1}[\frac{1}{2} + \frac{1}{2\sqrt{2N+1}} + \frac{1}{8(2N+1)}] \\
&\leq (3/4)\frac{1}{2N+1} \\
&= (3/2)\frac{1}{K-1}.
\end{aligned}$$

Combining these,

$$g \leq \frac{2K}{K-1} + \frac{1}{2\sqrt{14}}\frac{2K}{\sqrt{K-1}} + \frac{\sqrt{3}}{2}\frac{2K}{\sqrt{K-1}} = \frac{2K}{K-1} + (\frac{1}{2\sqrt{14}} + \frac{\sqrt{3}}{2})\frac{2K}{\sqrt{K-1}},$$

Since $(1/(2\sqrt{14}) + \sqrt{3}/2 \leq 1$,

$$g \leq \frac{2K}{K-1} + \frac{2K}{\sqrt{K-1}}.$$

Combining these, when $N \geq 2$, all entries of $(II) + (III)$ are no smaller than

$$-\frac{1}{K}[\frac{2K}{K-1} + \frac{2K}{\sqrt{K-1}}].$$

Note that by our conditions, all entries of $(I)$ are

$$\frac{1}{K}[\frac{2K}{K-1} + \frac{2K}{\sqrt{K-1}}].$$

Combining these proves the claim in Case (b) and completes the proof of Theorem 14 for case (U2b).

## I   DETAILS ABOUT ESTIMATING $(\Theta, \Pi, P)$ IN EXAMPLE 1 OF SECTION 1.1

In this section, we include more details on estimating $(\Theta, P, \Pi)$ in Example 1 of Section 1.1.

Given the adjacency matrix $A$ and the number of community $K$ of the network, let $(\hat{\lambda}_k, \hat{\xi}_k)$ be the $k$-th eigen-pair of $A$ (where $\lambda_k$ is the $k$-th largest eigenvalue in magnitude). Following Jin et al. (2022+), we apply the Mixed-SCORE algorithm and let $\hat{v}_1, \hat{v}_2, \ldots, \hat{v}_K$ be the estimated vertices of the Simplex there. Let $\hat{V} = [\hat{v}_1, \hat{v}_2, \ldots, \hat{v}_K]$ and $\hat{b}_1 \in \mathbb{R}^K$ be the vector where

$$\hat{b}_1(k) = (\hat{\lambda}_1 + \hat{v}_k' \text{diag}(\hat{\lambda}_2, \hat{\lambda}_3, \ldots, \hat{\lambda}_K)\hat{v}_k)^{-1/2}, \qquad 1 \leq k \leq K.$$

Let $\hat{B} = \text{diag}(\hat{b}_1)[\mathbf{1}_K, \hat{V}']$ and $\hat{\Lambda} = \text{diag}(\hat{\lambda}_1, \ldots, \hat{\lambda}_K)$.

- We estimate $\Pi$ by the Mixed-SCORE algorithm Jin et al. (2022+).
- We estimate $P$ by $\hat{P} = \hat{B}\hat{\Lambda}\hat{B}'$.
- Let $\hat{\theta}_i = \|\hat{\Xi}'e_i\|_1 / \|\hat{B}'\hat{\Pi}'e_i\|_1$, where $e_i$ is the $i$-th standard basis vector of $\mathbb{R}^n$, $1 \leq i \leq n$. We estimate $\Theta$ by $\hat{\Theta} = \text{diag}(\hat{\theta}_1, \hat{\theta}_2, \ldots, \hat{\theta}_n)$.

## J  Two algorithms for checking whether Problem (1) is solvable

In Section 2.2, we use a DFT approach and show that Problem (1) is solvable in many cases, and the results are more complete when $K$ is even, but less complete when $K$ is odd. Especially, in the end of Section 2.2, we mentioned that a numerical approach is helpful for it may cover some cases where our theorems do not cover. Below, we introduce an approach, focusing on two cases:

- (a). $K$ is odd, $m$ is even, and $(K+1)/2 \le m \le K-3$.
- (b). $K$ is odd, $m$ is odd, and $(K+1)/2 \le m \le K-4$.

Note here that Problem (1) is solvable when $m \le K/2$ or $m = K-1$ and is not solvable when $m = K-2$. Note also in both cases, $a_{K,m} = (2m-K) > 0$ and

$$J_{K,m} = \mathrm{diag}(1+2m-K, 1, \ldots, 1, -1, \ldots, -1).$$

Now, for (a), let $n = (K-m-1)/2$ and

$$Q_0 = [c_{K,0}, Q_1], \qquad \text{where} \qquad Q_1 = [c_{K,1}, s_{K,1}, \ldots, c_{K,n}, s_{K,n}].$$

Note that $Q_0 Q_0 = I_{2n+1} = I_{K-m}$. For any $G \in \mathbb{R}^{K,m}$ such that $G = [Q_0, Q_1, G]$ is a $K \times K$ orthogonal matrix,

$$Q J_{K,m} Q' = (2m-K+2)s_{K,0}s'_{K,0} + 2Q_1 Q'_1 - I_K = \frac{2m-K+2}{K}\mathbf{1}\mathbf{1}' + 2Q_1 Q'_1 - I_K. \quad \text{(J.54)}$$

By similar arguments as in the proof of Theorem 9, all diagonal entries of $Q J_{K,m} Q'$ are 0, and in order for $Q J_{K,m} Q'$ to be doubly stochastic, we only need to check if the off-diagonals of $Q J_{K,m} Q'$ are non-negative. This gives the following algorithm:

**Algorithm (a).** Given $(K, m)$ as in (a), check if all off-diagonal entries of the matrix on the RHS of (J.54) are non-negative.

Consider (b). Let $n = (K-m-2)/2$. Let $Q_1^* = [q_0, q_1, q_2, \ldots, q_{2n}] \in \mathbb{R}^{N,2n+1}$, where for $1 \le j \le n$,

$$q_0 = \begin{bmatrix} \mathbf{1}_{(K+1)/2} \\ \mathbf{1}_{(K-1)/2} \end{bmatrix}, \qquad q_{2j-1} = \begin{bmatrix} c_{(K+1)/2,j} \\ c_{(K-1)/2,j} \end{bmatrix}, \qquad q_{2j} = \begin{bmatrix} s_{(K+1)/2,j} \\ s_{(K-1)/2,j} \end{bmatrix}.$$

We partition $Q_1^*$ by

$$Q_1^* = \begin{bmatrix} \mathbf{1}_{(K+1)/2} & B_1 \\ -\mathbf{1}_{(K-1)/2} & B_2 \end{bmatrix},$$

For $c_0 = \sqrt{(K-1)/(K(K+1))}$, $d_0 = \sqrt{(K+1)/(K(K-1))}$, $c = \sqrt{\frac{1}{K}\left(\frac{K+1}{2} + \frac{1}{2n}\right)}$ and $d = \sqrt{\frac{1}{K}\left(\frac{K-1}{2} - \frac{1}{2n}\right)}$, we introduce a new matrix $Q_1$ by

$$Q_1 = \begin{bmatrix} c_0 \mathbf{1}_{(K+1)/2} & cB_1 \\ -d_0 \mathbf{1}_{(K-1)/2} & dB_2 \end{bmatrix}.$$

Similarly, let $Q_0 = [c_{K,0}, Q_1]$. By similar arguments as in Lemma 7, $Q'_0 Q_0 = I_{K-m}$, so there is a matrix $G \in \mathbb{R}^{K,m}$ so that $Q = [Q_0, Q_1, G]$ is a $K \times K$ orthogonal matrix. Similarly as above, it is seen that

$$Q J_{K,m} Q' = (a_{K,m}+2)c_{K,0}c'_{K,0} + 2Q_1 Q'_1 - I_K = \frac{2m-K+2}{K}\mathbf{1}\mathbf{1}' + 2Q_1 Q'_1 - I_K. \quad \text{(J.55)}$$

Compared with (J.55), the formula is the same, except for that the definition of $Q_1$ is changed. By similar arguments as in Lemma 7 and Theorem 11, all diagonal entries of $Q J_{K,m} Q'$ are 0, so we only need to check if all of its off-diagonals are non-negative.

**Algorithm (b).** Given $(K, m)$ as in (b), check if all off-diagonal entries of the matrix on the RHS of (J.55) are non-negative.

## K  SOLUTIONS FOUND BY THE OPTIMIZATION-BASED APPROACH

We solve the convex program in Section 3 with CVXPY[1] Agrawal et al. (2018); Diamond & Boyd. (2016) version 1.2.0. We use the default solver selected by the software. Below we give numerical solutions our approach found for

$$(K, m) = (11, 7), \qquad (K, m) = (15, 9), \qquad (K, m) = (19, 11).$$

For each case, we list

- $\lambda(\widehat{M})$ (the Eigenvalues of $\widehat{M}$ found by our approach, which demonstrate $\widehat{M}$ approximately achieves the desired rank $K - m - 1$),

- the matrix $\widehat{Q_1} \in \mathbb{R}^{K \times (K-m-1)}$ formed by the leading Eigen-vectors of $\widehat{M}$,

- and $\widehat{Q} J_{K,m} \widehat{Q}$, where $\widehat{Q} = [c_{K,0}, \widehat{Q_1}, \widehat{Q_2}]$. The matrix $\widehat{Q} J_{K,m} \widehat{Q}'$ is expected to be (entry-wise) non-negative.

Here, $\widehat{Q_1}$ is computed using our optimization algorithm, and $\widehat{Q_2} \in \mathbb{R}^{K,m}$ is any matrix such that $\widehat{Q}$ is orthogonal. Note that first by our algorithm, if we let $\widehat{Q} = [c_{K,0}, \widehat{Q_1}]$, then $\widehat{Q}_0' \widehat{Q}_0$ is (approximately) $I_{K-m}$ so such a $\widehat{Q_2}$ exists. Second, since

$$\widehat{Q} J_{K,m} \widehat{Q} = (2 + 2m - K) c_{K,0} c'_{K,0} + 2 \widehat{Q_1} \widehat{Q_1} - I_K,$$

$\widehat{Q} J_{K,m} \widehat{Q}$ does not depend on $\widehat{Q_2}$ so there is no need to compute $\widehat{Q_2}$ in our algorithm.

We use the notation $\widehat{Q} J_{K,m} \widehat{Q}_{:,i:j}$ for the $i$-th to $j$-th columns of the matrix. Due to space limitation, we present the Eigenvalues in scientific notation rounded to 3 decimal precision, and round matrix entries to 3 decimal places.

**Case 1**. $(K, m) = (11, 7)$:

$$
\lambda(\widehat{M}) = \begin{bmatrix} 1.000\text{e}+00 \\ 1.000\text{e}+00 \\ 1.000\text{e}+00 \\ 7.868\text{e}-06 \\ 7.829\text{e}-06 \\ 4.222\text{e}-06 \\ 3.907\text{e}-06 \\ 2.907\text{e}-06 \\ 1.542\text{e}-06 \\ 6.748\text{e}-07 \\ 4.497\text{e}-07 \end{bmatrix}, \quad
\widehat{Q_1} = \begin{bmatrix} -0.309 & -0.119 & 0.404 \\ -0.154 & -0.107 & 0.487 \\ -0.037 & 0.512 & 0.095 \\ -0.171 & -0.403 & -0.285 \\ -0.378 & -0.280 & -0.226 \\ -0.022 & 0.351 & -0.386 \\ 0.206 & 0.131 & 0.462 \\ -0.380 & 0.258 & -0.248 \\ 0.370 & -0.365 & -0.052 \\ 0.429 & 0.266 & -0.134 \\ 0.446 & -0.246 & -0.117 \end{bmatrix},
$$

$$
\widehat{Q} J_{K,m} \widehat{Q}'_{:,1:6} = \begin{bmatrix} 0.000 & 0.969 & 0.432 & 0.426 & 0.573 & 0.073 \\ 0.969 & 0.000 & 0.450 & 0.315 & 0.410 & 0.010 \\ 0.432 & 0.450 & 0.000 & -0.000 & 0.153 & 0.742 \\ 0.426 & 0.315 & -0.000 & -0.000 & 0.938 & 0.399 \\ 0.573 & 0.410 & 0.153 & 0.938 & -0.000 & 0.449 \\ 0.073 & 0.010 & 0.742 & 0.399 & 0.449 & 0.000 \\ 0.669 & 0.814 & 0.662 & 0.016 & 0.017 & 0.181 \\ 0.428 & 0.275 & 0.700 & 0.518 & 0.710 & 0.844 \\ 0.271 & 0.367 & 0.044 & 0.652 & 0.402 & 0.223 \\ 0.018 & 0.135 & 0.670 & 0.170 & 0.041 & 0.726 \\ 0.144 & 0.256 & 0.148 & 0.567 & 0.307 & 0.353 \end{bmatrix},
$$

---

[1] https://www.cvxpy.org/

$$\widehat{Q}J_{K,m}\widehat{Q}'_{:,7:11} = \begin{bmatrix} 0.669 & 0.428 & 0.271 & 0.018 & 0.144 \\ 0.814 & 0.275 & 0.367 & 0.135 & 0.256 \\ 0.662 & 0.700 & 0.044 & 0.670 & 0.148 \\ 0.016 & 0.518 & 0.652 & 0.170 & 0.567 \\ 0.017 & 0.710 & 0.402 & 0.041 & 0.307 \\ 0.181 & 0.844 & 0.223 & 0.726 & 0.353 \\ -0.000 & 0.137 & 0.463 & 0.577 & 0.466 \\ 0.137 & 0.000 & 0.011 & 0.332 & 0.047 \\ 0.463 & 0.011 & 0.000 & 0.592 & 0.976 \\ 0.577 & 0.332 & 0.592 & -0.000 & 0.738 \\ 0.466 & 0.047 & 0.976 & 0.738 & 0.000 \end{bmatrix}.$$

**Case 2.** $(K, m) = (15, 9)$:

$$\lambda(\widehat{M}) = \begin{bmatrix} 1.000\mathrm{e}{+}00 \\ 1.000\mathrm{e}{+}00 \\ 1.000\mathrm{e}{+}00 \\ 1.000\mathrm{e}{+}00 \\ 1.000\mathrm{e}{+}00 \\ 8.856\mathrm{e}{-}06 \\ 7.683\mathrm{e}{-}06 \\ 6.521\mathrm{e}{-}06 \\ 5.401\mathrm{e}{-}06 \\ 4.552\mathrm{e}{-}06 \\ 3.979\mathrm{e}{-}06 \\ 2.284\mathrm{e}{-}06 \\ 1.534\mathrm{e}{-}06 \\ 5.690\mathrm{e}{-}07 \\ 1.816\mathrm{e}{-}07 \end{bmatrix}, \quad \widehat{Q}_1 = \begin{bmatrix} -0.015 & -0.226 & 0.081 & 0.200 & -0.485 \\ -0.142 & -0.381 & -0.103 & 0.357 & 0.172 \\ -0.161 & 0.453 & 0.183 & 0.256 & 0.053 \\ 0.177 & 0.070 & -0.332 & -0.431 & 0.029 \\ 0.451 & -0.212 & 0.102 & 0.194 & -0.192 \\ 0.046 & 0.533 & -0.203 & 0.060 & 0.050 \\ -0.327 & 0.148 & 0.444 & -0.077 & 0.041 \\ 0.365 & -0.184 & -0.057 & 0.138 & 0.380 \\ -0.405 & -0.154 & -0.373 & 0.045 & -0.066 \\ 0.234 & -0.178 & 0.169 & -0.359 & 0.299 \\ -0.191 & -0.053 & -0.460 & -0.114 & -0.264 \\ 0.380 & 0.370 & -0.072 & 0.138 & -0.168 \\ -0.030 & -0.103 & 0.427 & -0.101 & -0.360 \\ -0.154 & -0.047 & 0.132 & -0.537 & 0.041 \\ -0.229 & -0.037 & 0.062 & 0.230 & 0.472 \end{bmatrix},$$

$$\widehat{Q}J_{K,m}\widehat{Q}'_{:,1:8} = \begin{bmatrix} 0.000 & 0.469 & 0.214 & 0.042 & 0.696 & 0.034 & 0.277 & 0.083 \\ 0.469 & 0.000 & 0.197 & 0.000 & 0.419 & 0.016 & 0.180 & 0.611 \\ 0.214 & 0.197 & -0.000 & 0.000 & 0.112 & 0.763 & 0.700 & 0.139 \\ 0.042 & 0.000 & 0.000 & -0.000 & 0.217 & 0.510 & 0.013 & 0.377 \\ 0.696 & 0.419 & 0.112 & 0.217 & 0.000 & 0.112 & 0.020 & 0.637 \\ 0.034 & 0.016 & 0.763 & 0.510 & 0.112 & -0.000 & 0.275 & 0.248 \\ 0.277 & 0.180 & 0.700 & 0.013 & 0.020 & 0.275 & 0.000 & 0.000 \\ 0.083 & 0.611 & 0.139 & 0.377 & 0.637 & 0.248 & 0.000 & -0.000 \\ 0.437 & 0.652 & 0.204 & 0.373 & 0.000 & 0.282 & 0.210 & 0.099 \\ -0.000 & 0.214 & 0.007 & 0.605 & 0.400 & 0.083 & 0.358 & 0.679 \\ 0.499 & 0.350 & 0.092 & 0.647 & 0.147 & 0.406 & 0.031 & 0.033 \\ 0.362 & 0.000 & 0.573 & 0.438 & 0.623 & 0.791 & 0.095 & 0.393 \\ 0.758 & 0.136 & 0.316 & 0.091 & 0.536 & -0.000 & 0.687 & -0.000 \\ 0.126 & 0.016 & 0.118 & 0.650 & 0.017 & 0.154 & 0.624 & 0.106 \\ 0.000 & 0.740 & 0.564 & 0.035 & 0.063 & 0.323 & 0.530 & 0.595 \end{bmatrix},$$

$$\widehat{Q}J_{K,m}\widehat{Q}'_{:,9:15} = \begin{bmatrix} 0.437 & -0.000 & 0.499 & 0.362 & 0.758 & 0.126 & 0.000 \\ 0.652 & 0.214 & 0.350 & 0.000 & 0.136 & 0.016 & 0.740 \\ 0.204 & 0.007 & 0.092 & 0.573 & 0.316 & 0.118 & 0.564 \\ 0.373 & 0.605 & 0.647 & 0.438 & 0.091 & 0.650 & 0.035 \\ 0.000 & 0.400 & 0.147 & 0.623 & 0.536 & 0.017 & 0.063 \\ 0.282 & 0.083 & 0.406 & 0.791 & -0.000 & 0.154 & 0.323 \\ 0.210 & 0.358 & 0.031 & 0.095 & 0.687 & 0.624 & 0.530 \\ 0.099 & 0.679 & 0.033 & 0.393 & -0.000 & 0.106 & 0.595 \\ 0.000 & 0.000 & 0.872 & -0.000 & 0.109 & 0.320 & 0.443 \\ 0.000 & 0.000 & 0.031 & 0.155 & 0.357 & 0.732 & 0.378 \\ 0.872 & 0.031 & -0.000 & 0.272 & 0.177 & 0.377 & 0.066 \\ -0.000 & 0.155 & 0.272 & -0.000 & 0.267 & -0.000 & 0.029 \\ 0.109 & 0.357 & 0.177 & 0.267 & 0.000 & 0.544 & 0.021 \\ 0.320 & 0.732 & 0.377 & -0.000 & 0.544 & -0.000 & 0.215 \\ 0.443 & 0.378 & 0.066 & 0.029 & 0.021 & 0.215 & 0.000 \end{bmatrix}.$$

**Case 3**. $(K, m) = (19, 11)$:

$$
\lambda(\widehat{M}) = \begin{bmatrix}
1.000\text{e}{+}00 \\
1.000\text{e}{+}00 \\
1.000\text{e}{+}00 \\
1.000\text{e}{+}00 \\
1.000\text{e}{+}00 \\
1.000\text{e}{+}00 \\
1.000\text{e}{+}00 \\
2.895\text{e-}05 \\
2.516\text{e-}05 \\
2.136\text{e-}05 \\
2.030\text{e-}05 \\
1.869\text{e-}05 \\
1.393\text{e-}05 \\
1.300\text{e-}05 \\
9.276\text{e-}06 \\
5.556\text{e-}06 \\
3.805\text{e-}06 \\
2.971\text{e-}06 \\
2.046\text{e-}06
\end{bmatrix},
$$

$$
\widehat{Q_1} = \begin{bmatrix}
-0.221 & -0.094 & -0.114 & 0.174 & 0.035 & 0.295 & -0.424 \\
-0.232 & 0.079 & 0.416 & -0.279 & 0.194 & 0.073 & 0.119 \\
-0.081 & 0.271 & -0.319 & 0.197 & -0.068 & -0.159 & 0.343 \\
0.136 & 0.147 & -0.110 & 0.001 & 0.051 & -0.000 & -0.560 \\
-0.056 & -0.055 & 0.233 & -0.318 & -0.362 & -0.270 & -0.059 \\
-0.056 & 0.032 & 0.145 & 0.452 & -0.277 & -0.226 & 0.106 \\
-0.220 & 0.261 & 0.162 & -0.165 & 0.134 & 0.357 & 0.230 \\
-0.029 & -0.094 & 0.154 & -0.325 & -0.418 & -0.180 & -0.150 \\
0.184 & 0.496 & 0.099 & -0.067 & 0.235 & 0.126 & -0.053 \\
0.167 & -0.155 & -0.377 & -0.060 & 0.279 & -0.236 & 0.193 \\
0.121 & -0.082 & -0.244 & -0.037 & -0.319 & 0.341 & 0.261 \\
-0.264 & -0.201 & -0.299 & -0.108 & 0.179 & -0.157 & -0.318 \\
0.419 & -0.373 & 0.031 & -0.152 & -0.049 & 0.071 & 0.150 \\
-0.334 & -0.157 & 0.257 & 0.393 & 0.077 & -0.051 & 0.058 \\
0.094 & -0.004 & -0.159 & 0.080 & -0.268 & 0.504 & -0.049 \\
0.445 & -0.047 & 0.185 & -0.048 & 0.352 & -0.088 & -0.012 \\
0.092 & -0.380 & 0.253 & 0.331 & 0.195 & 0.047 & 0.044 \\
0.194 & 0.421 & -0.026 & 0.183 & -0.133 & -0.310 & -0.079 \\
-0.363 & -0.064 & -0.287 & -0.254 & 0.165 & -0.137 & 0.199
\end{bmatrix},
$$

$$
\widehat{Q} J_{K,m} \widehat{Q}'_{:,1:7} = \begin{bmatrix}
0.000 & 0.115 & -0.000 & 0.679 & 0.000 & 0.163 & 0.243 \\
0.115 & -0.000 & 0.000 & 0.017 & 0.457 & 0.047 & 0.793 \\
-0.000 & 0.000 & 0.000 & 0.000 & 0.064 & 0.558 & 0.298 \\
0.679 & 0.017 & 0.000 & 0.000 & 0.209 & 0.080 & 0.000 \\
0.000 & 0.457 & 0.064 & 0.209 & -0.000 & 0.356 & 0.123 \\
0.163 & 0.047 & 0.558 & 0.080 & 0.356 & -0.000 & 0.015 \\
0.243 & 0.793 & 0.298 & 0.000 & 0.123 & 0.015 & -0.000 \\
0.137 & 0.347 & 0.002 & 0.317 & 0.972 & 0.293 & 0.075 \\
0.178 & 0.473 & 0.304 & 0.521 & 0.045 & 0.044 & 0.624 \\
0.000 & 0.000 & 0.538 & 0.159 & 0.028 & 0.065 & -0.000 \\
0.225 & 0.000 & 0.455 & 0.000 & 0.186 & 0.218 & 0.378 \\
0.637 & 0.135 & 0.153 & 0.572 & 0.236 & -0.000 & 0.003 \\
-0.000 & 0.148 & -0.000 & 0.088 & 0.348 & 0.091 & 0.051 \\
0.444 & 0.424 & 0.269 & 0.014 & 0.151 & 0.713 & 0.293 \\
0.606 & 0.000 & 0.222 & 0.350 & 0.056 & 0.189 & 0.408 \\
0.000 & 0.351 & -0.000 & 0.380 & 0.129 & 0.063 & 0.144 \\
0.355 & 0.279 & 0.000 & 0.093 & 0.030 & 0.481 & 0.103 \\
0.043 & -0.000 & 0.611 & 0.520 & 0.340 & 0.624 & 0.035 \\
0.174 & 0.415 & 0.528 & 0.001 & 0.270 & -0.000 & 0.417
\end{bmatrix},
$$

$$\widehat{Q}J_{K,m}\widehat{Q}'_{:,8:13} = \begin{bmatrix}
0.137 & 0.178 & 0.000 & 0.225 & 0.637 & -0.000 \\
0.347 & 0.473 & 0.000 & 0.000 & 0.135 & 0.148 \\
0.002 & 0.304 & 0.538 & 0.455 & 0.153 & -0.000 \\
0.317 & 0.521 & 0.159 & 0.000 & 0.572 & 0.088 \\
0.972 & 0.045 & 0.028 & 0.186 & 0.236 & 0.348 \\
0.293 & 0.044 & 0.065 & 0.218 & -0.000 & 0.091 \\
0.075 & 0.624 & -0.000 & 0.378 & 0.003 & 0.051 \\
-0.000 & 0.007 & 0.000 & 0.287 & 0.296 & 0.388 \\
0.007 & 0.000 & 0.155 & 0.091 & -0.000 & 0.053 \\
0.000 & 0.155 & 0.000 & 0.279 & 0.528 & 0.510 \\
0.287 & 0.091 & 0.279 & -0.000 & 0.000 & 0.579 \\
0.296 & -0.000 & 0.528 & 0.000 & 0.000 & 0.072 \\
0.388 & 0.053 & 0.510 & 0.579 & 0.072 & -0.000 \\
0.072 & 0.000 & 0.049 & 0.000 & 0.270 & 0.000 \\
0.214 & 0.258 & 0.000 & 0.847 & 0.070 & 0.394 \\
0.075 & 0.568 & 0.526 & 0.000 & 0.109 & 0.646 \\
0.000 & 0.024 & 0.285 & 0.130 & 0.171 & 0.540 \\
0.292 & 0.590 & 0.237 & 0.073 & 0.067 & -0.000 \\
0.225 & 0.065 & 0.643 & 0.251 & 0.682 & 0.091
\end{bmatrix},$$

$$\widehat{Q}J_{K,m}\widehat{Q}'_{:,14:19} = \begin{bmatrix}
0.444 & 0.606 & 0.000 & 0.355 & 0.043 & 0.174 \\
0.424 & 0.000 & 0.351 & 0.279 & -0.000 & 0.415 \\
0.269 & 0.222 & -0.000 & 0.000 & 0.611 & 0.528 \\
0.014 & 0.350 & 0.380 & 0.093 & 0.520 & 0.001 \\
0.151 & 0.056 & 0.129 & 0.030 & 0.340 & 0.270 \\
0.713 & 0.189 & 0.063 & 0.481 & 0.624 & -0.000 \\
0.293 & 0.408 & 0.144 & 0.103 & 0.035 & 0.417 \\
0.072 & 0.214 & 0.075 & 0.000 & 0.292 & 0.225 \\
0.000 & 0.258 & 0.568 & 0.024 & 0.590 & 0.065 \\
0.049 & 0.000 & 0.526 & 0.285 & 0.237 & 0.643 \\
0.000 & 0.847 & 0.000 & 0.130 & 0.073 & 0.251 \\
0.270 & 0.070 & 0.109 & 0.171 & 0.067 & 0.682 \\
0.000 & 0.394 & 0.646 & 0.540 & -0.000 & 0.091 \\
-0.000 & 0.085 & 0.101 & 0.741 & 0.134 & 0.240 \\
0.085 & -0.000 & 0.005 & 0.195 & 0.101 & -0.000 \\
0.101 & 0.005 & -0.000 & 0.571 & 0.332 & 0.000 \\
0.741 & 0.195 & 0.571 & 0.000 & 0.000 & 0.000 \\
0.134 & 0.101 & 0.332 & 0.000 & 0.000 & 0.000 \\
0.240 & -0.000 & 0.000 & 0.000 & 0.000 & 0.000
\end{bmatrix}.$$

## L  CONNECTION OF NIEP AND SOCIAL NETWORK MODELING

In Section 1.1, we mentioned that that the NMF problem (2) is motivated by network modeling. We now discuss this with more details. Consider a symmetric connected network with $n$ nodes and let $A$ be the adjacency matrix:

$$A(i,j) = \begin{cases} 1, & \text{if there is an edge connecting nodes } i \text{ and } j, \\ 0, & \text{otherwise,} \end{cases} \qquad 1 \le i, j \le n.$$

Conventionally, self edges are not allowed, so all diagonal entries of $A$ are 0. We assume the network has $K$ perceivable communities

$$\mathcal{C}_1, \mathcal{C}_2, \ldots, \mathcal{C}_K.$$

In many network models, we assume that the upper triangular entries of $A$. Also, for a non-negative matrix $\Omega \in \mathbb{R}^{n,n}$,

$$A = \mathbb{E}[A] + (A - \mathbb{E}[A], \qquad \text{and} \qquad \mathbb{E}[A] = \Omega - \text{diag}(\Omega). \tag{L.56}$$

We say the network model is a rank-$K$ model if $\Omega$ is an irreducible non-negative matrix where

$$\text{rank}(\Omega) = K;$$

recall that $K$ is the number of communities. Many well-known network models (e.g., the Random Dot Product Graph (RDPG) model (Young & Scheinerman, 2007) and the generalized RDPG model (Rubin-Delanchy et al., 2021) are rank-$K$ models.

In these models, the parameters do not have explicit practical meanings. It is desirable to have models where the parameters have more explicit meanings. The Degree-Corrected Mixed-Membership (DCMM) model is one of such models, where we further assume

$$\Omega = \Theta \Pi P \Pi' \Theta,$$

See for example (Zhang et al., 2020; Jin et al., 2022+). Here, $\Theta = (\theta_1, \ldots, \theta_n)$ is an $n \times n$ diagonal matrix where $\theta_i > 0$ is the degree heterogeneity parameter for node $i$, $\Pi = [\pi_1, \ldots, \pi_n]'$ is an $n \times K$ matrix where each $\pi_i$ a $K$-dimensional weight vector and represents the membership of node $i$, and $P$ is a $K \times K$ non-negative matrix, representing the baseline connecting probabilities between different communities.

Conventionally, we assume the ranks of $\Pi$ and $P$ are $K$, so a DCMM is also a rank-$K$ model. However, compared to other rank-$K$ models, all parameter matrices $(\Theta, \Pi, P)$ in the DCMM model have practical meanings and are easy to interpret. These make the DCMM model especially appealing in practice. An interesting question is then

$$\text{When is a rank-}K \text{ network model also a DCMM model?} \tag{L.57}$$

This is the NMF problem in (2).

By our results on NIEP in Sections 2-3 and on NMF in Section 4, have the following results.

**Lemma L.1** *We can always rewrite a rank-$K$ model as a DCMM model if either one of the following conditions holds.*

- *$K = 2$.*

- *$K \geq 3$ but Condition (B\*) (e.g., (12) in Section 4) holds.*

For real applications, we may consider the 5 networks below. The first 4 networks have 2 communities, and it is believed that a rank-2 model is appropriate, but it is not known that whether a DCMM model is also appropriate. By our lemma above, we conclude a DCMM model is appropriate, as long as a rank-2 model is appropriate. The UKfaculty network has 3 communities, and it is believed that a rank-3 model is appropriate. It was argued by Jin (2022) that Condition (B\*) holds in this case, so a DCMM model is also appropriate.

| Dataset | Source | #Nodes | #Edges | $K$ |
|---------|--------|--------|--------|-----|
| Weblogs | Adamic & Glance (2005) | 1222 | 16714 | 2 |
| Karate | Zachary & Wayne (1977) | 34 | 78 | 2 |
| Dolphins | Lusseau *et al.* (2003) | 62 | 159 | 2 |
| Polbooks | Krebs (unpublished) | 92 | 374 | 2 |
| UKfaculty | Nepusz *et al.* (2008) | 79 | 552 | 3 |

Note that Jin (2022) focused on the case of $m \leq K/2$. For the case of $m > K/2$, it remained unclear that under what conditions we can rewrite a rank-$K$ model as a DCMM model; such a more challenging case is addressed in the current paper. Note also that that the framework can be extended to weighted networks and asymmetrical networks (such as citation networks and bipartite networks).

Aside from the network modeling, similar settings also arise in topic modeling. Suppose we have n text documents, each with $N_i$ words, $1 \leq i \leq n$, and the dictionary size is $p$. Let $X_i$ be the $p$-dimensional word count vector for document $i$ and let $X = [X_1, X_2, \ldots, X_n]$. If we assume these documents discuss only $K$ topics, then it is reasonable to assume an rank-$K$ model, where we assume

$$\text{rank}(\mathbb{E}[X]) = K.$$

The Hoffmann's model is a well-known topic model (e.g., Ke & Wang (2022)), where we additionally assume for a $p \times K$ non-negative matrix $A$ and a $K \times n$ non-negative matrix $W$,

$$\mathbb{E}[X] = AW.$$

A natural question is then when we can rewrite a rank-$K$ topic model as a Hoffmann's topic model. Our theory can be directly extended to address this question.