# OpenReview forum: "Sharp results for NIEP and NMF"
_ICLR.cc/2024/Conference — Submitted to ICLR 2024_

### Official Review · Reviewer_Mjse · 2023-10-27

**Soundness:** 4 excellent
**Presentation:** 3 good
**Contribution:** 4 excellent
**Rating:** 8
**Confidence:** 1

**Summary:**

This paper focuses on a specific instance of the doubly stochastic NIEP with a certain form of predefined diagonal matrix. The author offers sharp result for this particular NIEP and its associated NMF problem. The constructive proof enables the development of a practical NMF algorithm. The paper also explores a comprehensive characterization of scenarios encountered in the context of the doubly stochastic NIEP, marking a significant advancement, particularly in challenging cases where $m$ is large.

**Strengths:**

The obtained results are novel and precise, making them particularly intriguing for addressing the NIEP. The approach relying on Haar and discrete Fourier Transform is not only interesting but also holds promise, hinting at a potential in tackling the remaining challenges posed by this difficult inverse problem.

**Weaknesses:**

I could not identify any apparent limitations within my limited knowledge in this field; see questions part for details.

**Questions:**

I apologize for my limited familiarity with the solvability of the NIEP and NMF problems. I have a straightforward question: Based on my understanding of Schur decomposition, it seems that a nonnegative matrix $A$ with a given spectrum can be expressed as $A = U(\Lambda+V)U^\top$, where $U$ is orthogonal, and $V$ is upper triangular. If this is the case, does the solvability of the nonlinear system $U(\Lambda+V)U^\top\ge0$ offer a potential approach to address the NIEP problem from an optimization perspective?

---

> ### Author Response · Authors · 2023-11-18
>
> **Response to your main point**.   We thank you for a very positive feedback for our paper.  We are especially
> glad that you think we have made a significant advancement for the NIEP problem,  particularly in challenging cases where the size of the matrix is large. You have raised the following main point.
> - I apologize for my limited familiarity with the solvability of the NIEP and NMF problems. I have a straightforward question: Based on my understanding of Schur decomposition, it seems that a nonnegative matrix  with a given spectrum can be expressed as
> $A = U (\\Lambda + V) U'$,  where $U$
>  is orthogonal, and $V$
>  is upper triangular. If this is the case, does the solvability of the nonlinear system
>  $U (\\Lambda + V) U' \\geq 0$
>  offer a potential approach to address the NIEP problem from an optimization perspective?
>
> Here is our response.  Yes, you are right, but this is a non-convex problem: the set of
> $K \times K$ orthogonal matrices (which is called the Stiefel manifold in the literature)
> is non-convex, and the optimization problem is also non-convex, so computationally,
> this is a challenging problem. For example, we may put an $\\epsilon$-net on the Stiefel manifold (so that for any point on the Stiefel manifold, there is a point in the net within a distance of $\\epsilon$) and only
> check your inequality when $U$ is a net-point.   However, such an $\\epsilon$-net will have at least
> $C(1/\\epsilon)^{O(K^2)}$ points, so this approach is computationally infeasible for relatively large $K$.

---

### Official Review · Reviewer_QrTg · 2023-11-03

**Soundness:** 1 poor
**Presentation:** 1 poor
**Contribution:** 3 good
**Rating:** 5
**Confidence:** 3

**Summary:**

The authors give results for the Non-negative Inverse Eigenvalue Problem (NIEP), which they define as follows:
for integers $K, m$ such that $1 \leq m \leq K-1$, let $a_{K, m} = \max(0, 2m - K)$ and let $J_{K, m} = diag(1 + a_{K, m}, 1, \dots,1, -1, \dots, -1)$, then is there a orthogonal $K \times K$ matrix $Q$ such that $QJ_{K, M}Q^{-1}$ is doubly stochastic?

The authors give results for when the problem is solvable and not for various values of $m$ and $K$ by connecting it to a related Non-negative Matrix Factorization problem.

**Strengths:**

The authors seem to shed light on an important problem in linear algebra with applications in NMF and network modeling.

**Weaknesses:**

Overall the writing of the paper is quite poor. Even though it seems like the result could be important, very little exposition is given to i) stating the result the paper shows clearly (for e.g. in a clean theorem statement), ii) its connection to related/relevant works to this paper and iii) it's relevance to other problems in machine learning/applications. Especially i and ii. This makes it hard to judge the paper on its merits.

A clear introduction, statement of the results and comparison to relevant works can significantly increase the quality of this paper.

**Questions:**

In order to address the above concerns, can you please çlearly state --
1) What is the connection between the NMF problem and the NIEP problem?
2) What is the result that this paper shows and what are already known results?

---

> ### Author Response · Authors · 2023-11-18
>
> **Response to your main points**.  Thanks for your valuable comments. You have raised two major points.
> - What is the connection between the NMF problem and the NIEP problem?
> - What is the result that this paper shows and what are already known results?
>
> Response to your main point 1.  Thanks for your valuable comments.
> To clarify,   the whole section of Section  1.1 is for explaining the relationship between NMF and NIEP. In detail,  if  the NIEP is solvable for $(K, m)$, then
> the NMF is solvable if Condition (A) is satisfied with $a \geq a_{K, m}$, and
> Condition (A) cannot be further relaxed in the sense that $a_{K, m}$ is the smallest possible value of $a$ there. When the NIEP is not solvable, then the NMF is solvable if Condition (A) is satisfied with an $a \geq \tilde{a}\_{K, m}$  where
> $\tilde{a}\_{K, m}  > a_{K,m}$ is given explicitly in Section 4. In this case,  we are not sure if $\tilde{a}_{K, m}$  is the smallest possible $a$ there.  Additionally, we have also added Section L in the supplement to explain the relationship between the NMF problem and network modeling.
>
>
> Response to your main point 2. Thanks for your valuable comment.  We have revised several places in Section 1 to improve literature review, overview of our results, and comparison with existing literature.  To further clarify,
> our main results are as follows.
> - The orthodox NIEP concerns the following problem:  given $K$ numbers $\sigma_1,\sigma_2,\ldots, \sigma_K$, when do we have a $K \times K$ non-negative matrix $M$ with $\sigma_1, \sigma_2, \ldots, \sigma_K$ as its eigenvalues?
> The existing literature on the NIEP have been largely focused on general $(\sigma_1, \sigma_2 \ldots, \sigma_K)$ but
> small $K$ (e.g., $K \leq 4$) (e.g., see the two survey papers we added  in the first paragraph of
> the introduction). Especially, the orthodox NIEP and all of the subproblems we mentioned in the first
> paragraph of our paper remain unsolved for $K \geq 5$.  See Page 1 of the paper for details.
> - Since the orthodox NIEP is too hard (and people can only crack the case for $K \leq 4$ in the past  70 years),  it is of interest to focus on some special cases (especially those motivated by real applications),  where the results may be more fruitful.
> - Motivated by the NMF problem and network modeling, we  focus on a special case where
> $(\sigma_1,\sigma_2, \ldots, \sigma_K) = (1 + a_{K, m}, 1, \ldots, 1, -1, \ldots, -1)$  but $K$ is arbitrary. Here,  $1 \leq m \leq (K-1)$, and $a_{K,m} = \max\{0, 2m-K\}$.
> We show that for $9$ different cases (S1)-(S9), the NIEP is solvable.  For example, one of these cases is $K$ is a multiple of $4$ and $m$ is arbitrary; we show that in this case the NIEP is always solvable.   We also identify two cases (N1)-(N2)  and proved that the NIEP is not solvable in these cases. Table 1 in Section 3 summarizes our results for all pairs of $(K, m)$ with $K \leq 20$ (but
> our theorems are for general $(K, m)$).
> - Moreover, using the results on NIEP and the connection  between NMF and NIEP in Section 1.1,  we derive sharp results for NMF in Section 4. There is a rich literature on solving general NMF problems of the form $V \approx W H$, e.g. [1] and the ensuing work, but they cannot be directly applied to our NMF problem (2), which has a more constrained factorization. We will add these discussion to the paper.
>
> In response, we will make the following changes.
> - We have revised the first paragraph of Section 1 to clarify the difference between orthodox NIEP and our NIEP. Especially,  we have added more references to the literature on the NIEP.
> On Page 1, we have revised the first paragraph and added several
> references. Existing approaches on NIEP include the Fiedler's approach and Soules' approach,
> but these approaches focus on the orthodox NIEP and are very different from ours.
> We focus on a special NIEP, where the results are new (especially as the NIEP
> remain unsolved for $K \geq 5$ while our $K$ is arbitrary).  See the two NIEP survey papers we added on Page 1 for details.
> - We have  revised Section 1.1 slightly to clarify the relationship between NMF and NIEP.
> - We will revise our results summary in Section 1.2
> - We have added Section L in the supplement to explain the relationship between the NMF problem and network modeling.  We have also discussed the modeling of $5$ real networks.
>
> [1] Daniel D Lee and H Sebastian Seung. Learning the parts of objects by non-negative matrix factorization.Nature, 401(6755):788–791, 1999

---

### Official Review · Reviewer_dPXa · 2023-11-08

**Soundness:** 3 good
**Presentation:** 3 good
**Contribution:** 3 good
**Rating:** 6
**Confidence:** 4

**Summary:**

The orthodox non-negative inverse eigenvalue problem (oNIEP) states as follows: the sufficient and necessary conditions such that the list of $K$ complex numbers $(\lambda_1, \cdots, \lambda_K)$ is the spectrum of a $K \times K$ non-negative matrix $A$. Since it is difficult, this paper considers one of its special cases: for the integer pair $(K, m)$, under what conditions, there exists a $K \times K$ orthogonal matrix $Q$ that can be efficiently computed, such that $Q J_{K, m} Q^\top$ is doubly stochastic, where $J_{K, m} = \text{diag}(1+a_{K,m}, 1, \cdots, 1, -1, \cdots, -1)$ and $a_{K, m} =\max \\{ 0, 2m - K \\}$. The proposed problem is closely related to the non-negative matrix factorization (NMF) and network modeling. This paper explores the solvability of different $(K, m)$ pairs by utilizing some novel techniques, including Haar basis, discrete Fourier transform (DFT), and Fiedler's approach.

**Strengths:**

(1) This paper researches one special case of the oNIEP that is closely related to NMF and network analysis, which are fundamental areas in machine learning.
(2) For the proposed problem, this paper takes advantage of some novel techniques (Haar basis, DFT, and Fielder's approach) to explore the solvability of different integer pairs $(K, m)$. The proofs are technically solid.
(3) The proposed algorithms (Algorithm 1 and Algorithm 2) for NMF explicitly compute the matrix $Q$, and the results are sharp.

**Weaknesses:**

(1) This paper is very technical and it would be better if it could introduce more related work in the Introduction section about NIEP and NMF, etc.
(2) This paper analyzes the solvability of the special NIEP and provides the algorithms for NMF, but does not give the time complexity of the proposed method, which lacks comparisons with some prior methods with respect to time efficiency.
(3) Although this paper presents the application of the proposed problem in NMF and network modeling in Section 4, it is still not very clear the specific applications in NMF and DCMM. It would be better if some specific experiments were implemented for NMF and network analysis.

Some typos:
1) Page 7, Section 3, paragraph 1, line 3, Table 2.5 -> Section 2.5
2) Page 8, the line above Algorithm 2, develop an more -> develop a more
3) Page 9, Section 5, line 3, delete "analysis of" in "analysis of network analysis"

**Questions:**

See Weaknesses.

---

> ### Author Response · Authors · 2023-11-18
>
> **Response to your main points**.  Many thanks for your very positive feedback on our paper. We are especially glad that you think our approaches are novel, our proofs are solid, and our paper is well written. You have raised the following main points.
> -  This paper is very technical and it would be better if it could introduce more related work in the introduction section about NIEP and NMF, etc.
> -  This paper analyzes the solvability of the special NIEP and provides the algorithms for NMF, but does not give the time complexity of the proposed method, which lacks comparisons with some prior methods with respect to time efficiency.
> -  Although this paper presents the application of the proposed problem in NMF and network modeling in Section 4, it is still not very clear the specific applications in NMF and DCMM. It would be better if some specific experiments were implemented for NMF and network analysis.
>
> Response to your main point 1.  Thanks for your comments. In response, first, in the last few lines of the first paragraph of the introduction, we have add a couple of sentences and four references to orthodox NIEP,  including two survey papers on NIEP.
> Second, we have added a long discussion on how the NMF problem in equation (2)
> is motivated by social network modeling. Due to space limit, we have put it in the Section L of the
> supplement.
>
> Response to your main point 2. Time efficiency of the proposed algorithm: with explicit construction of the $Q$ matrix, the dominating computation in both Algorithms 1 and 2 are computing the top $K$ Eigen value and vector pairs of a $n$-by-$n$ real symmetric matrix, which can be solved in O($Kn^2$) time by the Lanczos method or power iteration, and there exist quite a few numerically stable and well optimized software implementations. Standard methods for NMF such as the multiplicative update method of [1] are iterative optimization, and usually run in $O(T K n^2)$ time, where $T$ is the number of iterations. Hence our proposed method with explicit construction of the $Q$ matrix has similar time complexity to existing methods. However, [1] and other optimization-based NMF methods yield factorizations of the form $V \approx W H$, where $W$ and $H$ are allowed to be different. To solve our NMF problem in (2), these techniques need to be modified to enforce extra constraints and may run slower. When constructing $Q$ via numerical optimization as in Lemma 13, the time complexity can be significantly higher. We will add these discussions to the paper.
>
> Response to your main point 3.  Thanks and this is a very interesting point. In Section L of the supplement, we have added a section explaining how the NMF problem is related
> to network modeling. Consider a network with $K$ communities.
> The rank-$K$ model is one of the most frequently used model
> for social works, including block models, random dot product graph (RDGP) models,
> and generalized RDGP models as special cases. The Degree-Corrected Mixed-Membership
> model is also a rank-$K$ model, but where the parameters all have explicit meanings and are
> more interpretable in practice. An interesting question is therefore when a rank-$K$ model is also a DCMM model.
> We show that
> - When $K = 2$, a rank-$K$ model is always a DCMM model.
> For the frequently seen networks weblog, dolphin, karate, polbooks for example.
> In the literature, we usually assume they have $2$ communities, and
> so a rank-$2$ model are appropriate. Using our results, whenever a rank-$2$ model
> is appropriate for these networks, a DCMM model is also appropriate for them.
> - For networks with $K \geq 3$ communities,  we may similarly model them with a rank-$K$ model. And to check if we can write them as a DCMM model, all we need is to check whether
> Condition (B*) in Section 4 is satisfied; we can check this by estimating $\Omega$ first. See Section 4 of  (Jin, 2022) for example, where it was argued that UKFaculty network satisfies Condition (B*) with $K = 3$  and so we can model it with a DCMM model with $K = 3$. See the table below and
> Section L of the supplement.
>
> | Dataset | Source |  \#Nodes | $K$  | \#Edges  |
> | ----------- | ---------- | ------------- | ------- | ------------- |
> | Weblogs | Adamic &  Glance (2005) |  1222 | 2 | 16714  |
> | Karate |  Zachary & Wayne (1977) |  34 | 2 | 78  |
> | Dolphins | Lusseau et al. (2003) | 62  | 2  | 159  |
> | Polbooks | Krebs (unpublished)  | 92  | 2 | 374   |
> | UKfaculty | Nepusz et al. (2008)  |  79 | 3 | 552    |
>
>
> **Response to your minor points**.   We thank you for pointing out these typos,   and we have  fixed them in the revision.
>
> [1] Daniel D Lee and H Sebastian Seung. Learning the parts of objects by non-negative matrix factorization.Nature, 401(6755):788–791, 1999

---

> > ### Comment · Reviewer_dPXa · 2023-11-20
> >
> > Thank you for your detailed feedback. From the technical perspective, this paper seems solid and has enough contributions. Like some other reviewers, I am also concerned with the motivation and the application of the proposed method.  Your explanation about the relationship between NIEP and NMF, network analysis makes sense. Thank you for providing the time complexity of the proposed algorithm.
> >
> > Overall, I tend to give a positive evaluation of this paper. In Appendix L and your reply to the third weakness, you explained that one of the applications of the proposed method is identifying suitable models in real network analyses. Does there exist some other existing work on this application? If yes, what is your advantage over them?
> >
> > Some minor typos: (1) At the bottom of Page 8, in the sentence "This suggests and interesting conjecture", and -> an; (2) In Appendix K, there is ?? after CVXPY.

---

> > > ### Author Response · Authors · 2023-11-20
> > > **Replying to official comment by Reviewer dPXa**
> > >
> > > Thanks for your prompt feedback and positive evaluation! For your first question, the application to network modeling was discussed recently in Jin (2022), but the major difference is that, Jin (2022) only covers the case of $m \leq K/2$ for technical reasons. Therefore, for  applications with $m > K/2$, it remains unclear that under what conditions, we can rewrite a rank-$K$ network model as a DCMM model. This is a much more challenging problem, and is addressed in the current paper.  Also, Jin (2022) focused on binary unweighted networks. We note that the framework can be extended to weighted networks and asymmetrical networks (such as citation networks and bipartite networks).
> > >
> > > Aside from the network modeling, similar settings also arise in topic modeling. Suppose we have n text documents, each with $N_i$ words, $1 \leq i \leq n$, and the dictionary size is $p$. Let $X_i$ be the $p$-dimensional word count vector for document i and let $X = [X_1, X_2, \ldots, X_n]$. If we assume these documents discuss only K topics, then it is reasonable to assume an rank-$K$ model, where we assume
> > > $$
> > > \mathrm{rank}(\mathbb{E}[X])  =K.
> > > $$
> > > The Hoffmann's model is a well-known topic model, where we additionally assume
> > > $$
> > > \mathbb{E}[X] = A W,
> > > $$
> > > for a $p \times K$ non-negative matrix $A$ and a $K \times n$ non-negative matrix $W$. A natural question is then when we can rewrite a rank-$K$ topic model as a Hoffmann's topic model. Our theory can be directly extended to address this questions.
> > >
> > > We will add all these comments to the end of the supplement.
> > >
> > > We also thank you for the suggested minor changes. We will correct them slightly later today.

---

> > > > ### Comment · Reviewer_dPXa · 2023-11-21
> > > >
> > > > Thank you for your clarification. I will retain my score.

---

> > > > > ### Author Response · Authors · 2023-11-21
> > > > > **Response to Reviewer dPXa**
> > > > >
> > > > > Thanks. I may be wrong, but it seems that you have significant lowered your rating after our revision. We have been trying very hard to address your raised issue. Could you let us know is there any specific area that you are still not satisfied?
> > > > >
> > > > > As I reply to another reviewer, I think it is a win-win situation for machine learning journal to publish difficult, solid, and yet closely related mathematical results, and we think our papers is a paper of this kind. From the machine learning perspective, I think this is something the community should encourage. Hope this makes sense to you, thanks!

---

> > > > > > ### Comment · Reviewer_dPXa · 2023-11-22
> > > > > >
> > > > > > Yeah, from my personal point of view, I tend to accept this paper, but my score would be kind of conservative since I am not an expert in this field.

---

### Official Review · Reviewer_gqyq · 2023-11-09

**Soundness:** 3 good
**Presentation:** 2 fair
**Contribution:** 3 good
**Rating:** 5
**Confidence:** 3

**Summary:**

In this paper, the authors consider a special case of the Non-negative Inverse Eigenvalue Problem (NIEP) motivated by non-negative matrix factorization (NMF) and network modeling.
For various choices of the pair $(K,m)$, with $K$ the size of the matrix and $m$ the number of (diagonal) entries equal to $-1$, the authors demonstrate that the NIEP is solvable and provide efficient algorithm.
Some discussions are made in Section 4 on the application to NMF and network modeling.

**Strengths:**

The paper is technically strong and I believe, makes a significant contribution to advance the art of NIEP, by wisely combining some existing ideas (e.g., Haar and Discrete Fourier Transform approach) in a clearly non-trivial manner.

**Weaknesses:**

My major complaint about this paper is its presentation.
The paper contains ton of information and is written in such a way that is not easy to follow.
It would be great if the authors could make more efforts in organizing and presenting the technical in a way more accessible to general ML/AI audience.

**Questions:**

I do not have specific questions, but the following comments:

1. Section 1.2: "We also show that in 2 two different cases, (N1)-(N2), Problem (1) is not solvable": redundancy here?
2. Section 2: "Theorems 2-4 are proved in the supplement" perhaps say also in which specific section of the supplement?
3. after Theorem 9: For Case (d))
4. after Lemma 13: "This motivates the following algorithm", could the algorithm be put into an "algorithm" environment here? or is it just not necessary?
5. more generally, the authors propose to tackle Problem (1) using a construction approach by exploiting Haar and DFT basis. To me this choice of basis may not be unique, could the authors comment on the possible extension of the proposed construction?

---

> ### Author Response · Authors · 2023-11-18
>
> **Response to your main point**. You have made the following main point.
>
> - The paper contains ton of information and is written in such a way that is not easy to follow. It would be great if the authors could make more efforts in organizing and presenting the technical in a way more accessible to general ML/AI audience.
>
> Thanks  for a very valuable comment.  To respond, we have made the following changes.
> - We have greatly revised Section 2 (Page 4 and Page 5-6, marked in blue) to
> discuss the key ideas underlying our proofs.
> - We have added Section L in the supplement to explain the connection between the NMF problem and the social network modeling problem.  Especially, we discuss the impact our results
> on modeling $5$ well-known networks: weblog, dolphin, karate, polbooks, and UKFacutly.
>
> **Response to your other points**. You have made the following other points.
> - Section 1.2: ``We also show that in 2 two different cases, (N1)-(N2), Problem (1) is not solvable``: redundancy here?
> -  Section 2:  ``Theorems 2-4 are proved in the supplement`` perhaps say also in which specific section of the supplement?
> -  After Theorem 9: For Case (d))
> -  After Lemma 13: ``This motivates the following algorithm``, could the algorithm be put into an algorithm environment here? or is it just not necessary?
> - More generally, the authors propose to tackle Problem (1) using a construction approach by exploiting Haar and DFT basis. To me this choice of basis may not be unique, could the authors comment on the possible extension of the proposed construction?
>
> Thanks for your very careful read and valuable comments. For 1-3: we have revised as suggested.
> For 4: we can but we didn't do it due to the space constraint. We will try to revise the paper to accommodate that. For 5: this is a very interesting point and for some of the cases, we have indeed proposed different approaches.  For example, consider the case of $K = 2^d$, where $d \geq 1$ is an integer.  In this case, we can use the Haar construction as in Theorem 6, and we can also use the Haar-DFT combined construction as in Theorem 10. How to extend and combine the proposed approach  to broader cases  is a very interesting research problem for the future work.

---

> > ### Comment · Reviewer_gqyq · 2023-11-22
> >
> > I thank the authors for their clarification and revision.
> >
> > One last comment: note that the page limit of ICLR is 9 for both the initial and final camera ready version. And the revised version has now about half a page more, so it would be great to remove something to the appendix to reduce back to 9 pages (at least).

---

> > > ### Author Response · Authors · 2023-11-22
> > > **Reply to Reviewer gqyq**
> > >
> > > Thanks for the comments. We will shorten it if the paper is accepted. One way is to remove Theorem 6 and the paragraph
> > > above it to the appendix. Theorem 6 can be viewed as a special case of Theorem 10, but it is interesting for its own sake as the construction of matrix Q there is completely different that of Theorem 6, so the idea there can be very useful for future work. But if we have to remove a part to the appendix, we can choose to move this particular part.

---

### Official Review · Reviewer_SvT9 · 2023-11-10

**Soundness:** 2 fair
**Presentation:** 1 poor
**Contribution:** 3 good
**Rating:** 6
**Confidence:** 3

**Summary:**

The paper studies chiefly the **NIEP** (Non-negative Inverse Eigenvalue Problem). This is something of a pure math question. First, recall that a matrix is _double-stochastic_ if it has non-negative entries, each row sums to 1, and each column sums to 1. Then, for all pairs $(K,m)$ where $K=1,...,\infty$ and $m = 1,...,m-1$, we want to know if there exists a rotation matrix $Q \in R^{k \times k}$ such that $Q J_{K,m} Q^\intercal$ is double-stochastic, where $J_{K,m} = \text{diag}(1+\text{max}\\{0,2m-K\\}, 1, ...,1, -1, ..., -1)$ is a diagonal matrix with $K$ rows and columns and exactly $m$ many -1's in it.

For the NIEP, the authors show that a large range of $(K,m)$ pairs either do or do not admit such a matrix $Q$. Many different constructions for such $Q$ are provided, with motivations behind these constructions varying between the _Haar Basis_, the _Discrete Fourier Transform_, a mixture of the Haar Basis and DFT, and _Fielder's Approach_.

Several $(K,m)$ pairs are not covered by this analysis, and the authors use a heuristic algorithm to show that some of these pairs likely admit a $Q$ matrix. These are the only experiments in the paper.

The NIEP is related to the **NMF** (Non-negative Matrix Factorization) Problem. Here, we are given a non-negative rank-$K$ matrix $\Omega \in R^{n \times n}$. We want to find a non-negative tall-and-skinny matrix $\Pi \in R^{n \times K}$ and non-negative diagonal matrix $P\in R^{K \times K}$ such that $\Omega = \Pi P \Pi^\intercal$. This problem is computational, and not a pure math question. By studying these various cases for the NIEP, it becomes clear that NMF is solvable in many regimes with a simple algorithm.

**Strengths:**

The paper studies a mathematically interesting question (NIEP), and seems to use novel constructions to answer this question. There's a lot of care and precision put into characterizing how to build these $Q$ matrices, and a really wide variety of tools are called in to participate in this task.

NMF is also a problem that's known to be interesting. If I ask around the office, people have heard of it, and know that it matters. So, there's certainly significance in improving our knowledge of when this problem is solvable, and how to solve it.

The relationship between NIEP and NMF seems to be nice and crisp. A good amount of attention is paid to showing that the authors' solution to the pure math NIEP problem implies simple algorithmic solutions to the NMF problem. This focus on pragmatism in a theoretical paper is also good and appreciated.

The writing in the paper is clear (though almost always focuses on the wrong subject matter; see "weaknesses"). I appreciate the effort taken to represent many different cases in a single paper, though I do believe this somewhat misses the point of what the paper should be trying to convey.

**Weaknesses:**

This paper cannot be accepted in its current state. At its core,
- The paper lacks a clear discussion in its own right that the NIEP and NMF problems are well motivated. There are exactly 2 sentences that motivate the problem studied. Instead of actually motivating the problem, these sentences instead say that someone else wrote down a good motivation. This is vastly insufficient evidence, and the paper should motivate its own problem clearly.

$\phantom{.}$

- The body of the paper contains many constructions for $Q$ matrices that solve the NIEP for various $(K,m)$ pairs. It does not, however, justify _at all_ why these constructions should work, or **any intuition at all**.
    - For a theoretical paper, this is devastating. It leaves me with no confidence that the results are correct, and the appendix is out of scope of the typical review. _(I read a bit of the appendix, to get a flavor of Theorem 2 for instance, but it was fairly cold and not laden with any helpful intuition)_
    - It would be much better to describe constructions in the appendix, and intuitively explain the ideas behind constructions in the body of the paper.

At the core of the issue of the paper are the above two issues: the paper lacks motivation, and makes no attempt to convince me of its correctness.

A lesser issue is that I do not understand if this paper is a good fit for ICLR, or really any other ML conference. The body of the paper feels a like pure math with a coat of paint on it to give it the rough shape of an ML paper. This harkens back to the motivation a bit (an unmotivated theoretical paper looks like pure math to me), but also the NIEP feels not-ML-like to me. I hesitate to get to gate-keepy about this, so I won't strongly hold this against the paper. A stronger connection to the practice of ML would go a long way.

---

I'll backup my bullet points above with some concrete evidence.

The two sentences in my first point can be found after problem (2) on page 2. Admittedly, there's some motivating sentences in the conclusion too, but that's far too late and don't actually discuss any additional motivations.

The lack of intuition and justification can be seen throughout the paper, on pages 2-6 and page 8. The most brutal examples might be the following three:
- On page 3, the first paragraph of section 1.2 states that 4 different techniques are used to design $Q$ matrices. No explanation for why these methods work is ever given. No discussion of what similar methods would or wouldn't work. No lessons to learn at all.
- On page 4, Theorems 2, 3, and 4 are all stated without any justification or argument whatsoever. They are simply stated.
- The top half of page 5 is devoted to define some intricate constructions for $Q$ matrices related to the discrete fourier transform. These constructions are followed by two theorems which state that these $Q$ matrices resolve the NIEP for some $(K,m)$ pairs. No reason connecting the construction to the theorem is given; no reason that these intricate $Q$'s are useful.

I believe that the subject matter of this paper could be published at a good venue. In the current presentation, I cannot accept this to ICLR, and I remain unclear if any ML conference is a good fit. But these seems like good research results that belong in some published venue.

**Questions:**

These are general minor edits, typos, ect. Feel free to ignore all of these if you want to.

1. [throughout the paper] Be sure that when you cite someone, you keep parenthesis around the citation. Not having those parenthesis makes a lot of sentences kinda hard to read. (e.g. first sentence of the introduction).
1. [throughout the paper] The language of NIEP being "solvable" is deeply confusing language. In the view of computer science, it's a deterministic problem -- for each $(K,m)$ pair there exactly 1 hardcoded answer, so the problem is "solvable". The question is really if $Q$ exists for this particular $(K,m)$ pair, which should not use the word "solvable". Instead say something like "$J_{K,m}$ can be made doubly stochastic". Or use some other phrase that sounds information-theoretic and not computational.
1. [throughout the paper] The notation of $4 | K$ is not typical in any literature I've worked in. Define this if your going to use it.
1. [page 2] Discuss condition (B) more. It's really unintuitive, and looks almost impossible to satisfy at a glance. You describe this as the "sharpest" possible bound, but I have no clue what that actually means. Assume your audience hasn't ever seen NMF before, but is interested in learning more about it.
1. [page 2] Is there a reason to omit the proof of lemma 1 instead of appendicizing it? Why not just appendicize?
1. [page 2, paragraph before "The Matrix Q"] I really like this paragraph. Good paragraph and summary!
1. [page 4] This "particularly hard and takes a long time to figure out" is not really paper-writing-language. I think this is called being "overly editorialized". You can call it intricate or something, but really you don't need to include this parenthetical at all. Alternatively, you could actually describe what made the proof so tricky and clever.
1. [page 4] I think that $H^*$ needs to be composed of $[h_{K,2}, ..., h_{K,m+1}]$ in order for $U$ to be orthogonal?
1. [page 4] Why include Theorem 6 if its implies by Theorem 10. Just cut theorem 6.
1. [page 6] The "the reason why" sentence really isn't a reason. The sentence doesn't explain why your construction _conceptually_ fails here.
1. [page 8]  Why do you pick $tr(M \hat M_0)$ to be the optimization objective? Also, why not use a uniform-at-random orthonormal matrix $\hat M_0$ -- it's not really going to make a difference in practice but seems more conceptually sound.

Do not assume that the readers know:
- What the Haar basis matrix is
- What a Perron root is
- What the Stiefel manifold is

and be sure to define these terms before you use them.

**Details Of Ethics Concerns:**

M/A

---

> ### Author Response · Authors · 2023-11-18
>
> **Response to your summary**.  Thanks and we are especially glad that you think
> - (a) we study an interesting question using novel constructions, (b) there is a lot of care and precision we put into characterizing the $Q$ matrix, and (c) people heard about our problem and know why it matters.
> - the relationship between NMF and our problem (NIEP) is nice and crisp.
> - the writing of our paper is clear, and the practice side of the paper is appreciated.
> -  the paper could be published in a good venue.
>
> We thank you for many very positive comments,
> which we very much appreciate. We are also a little surprised to see that you gave our paper a
> very low rating.  We have revised our papers (especially in Section 2, marked in blue).  We hope these changes have satisfactorily addressed your concern.
>
> **Response to your main points**. In particular, you have raised the following points.
> - Main point 1.  The paper lacks a clear discussion in its own right that the NIEP and NMF problems are well motivated ...
> - Main point 2  The body of the paper contains many constructions for $Q$
>  matrices that solve the NIEP for various $(K, m)$ pairs. It does not, however, justify at all why these constructions should work, or any intuition at all ....
> - Main point 3.  A lesser issue is that I do not understand if this paper is a good fit for ICLR, or really any other ML conference.
>
> For your main point 1, it seems that there is a major misunderstanding. The NIEP problem is motivated by an NMF problem in equation (2), and the NMF problem is motivated by the problem of social network modeling:
>
> Network modeling -->  The NMF problem in equation (2) --> NIEP,
>
> In Section 1.1, we have used a whole page to explain how the NIEP is motivated by the
> NMF problem in equation (2).  The two sentences you mentioned is for explaining how the NMF problem in equation (2)
> is motivated by network modeling. For the latter, we have now added a more detailed explanation, but for space limit, we put it in the appendix (Section L, the last page), where we also discuss modeling for $5$ real social networks.
>
> For your main point 2: thanks and your comments have greatly helped us in revising our paper. In particular,
> - we have rewritten the first part of Section 2 (marked in blue) on page 4,
> explaining a key idea in our proofs. We also used the key idea to explain the proof for Theorem 2.
> - we have rewritten Section 2.2 (marked in blue) on Page 5-6, explaining the key idea
> underlying Lemma 7 and Theorem 8 there.
>
> We encourage you to read the changes. We hope these changes have explained some of the key ideas in our proofs  and assure you that our proofs are correct and rigorous.
>
> For your main point (3),  we think being diverse, inclusive, and interdisciplinary is one of the main reasons why ML conferences become so successful in the past decades.
> - It is a win-win to publish papers of this kind in ML conferences: on one hand, we can  attract talented mathematicians to tackle ML problems in areas such as NMF and network analysis,   and on the other hand,  we can enrich mathematics by providing mathematicians with new ideas and new problems.
> - Both NMF and network analysis are important ML research topics, and
> in the past decades, ML conferences have published many interesting papers in these areas.
> We have spent almost a whole page (Page 2)  explaining how the problem (i.e., NIEP)  grows out from NMF problem in equation (2),
> and in the appendix (Section L),  we have added discussion on how the NMF problem (2) is motivated by network modeling.
> See our response to your main point (1) above.
>
> **Response to your other points and minor points**.  Thanks. For your other point:
> we think they are already addressed in our response to your main points.  For the minor points:  thanks, and we will revise as suggested if not already.

---

> > ### Author Response · Authors · 2023-11-22
> > **Reply to Reviewer SvT9**
> >
> > Thanks again for many very positive comments. In the last round of the review, you have some concerns about the presentation. To address your concern, we have revised the paper in many places. Especially, we have revised Section 2 (marked blue) to shed light on the some of the main ideas in our proofs.  Please let us know if we have satisfactorily addressed your concern.  Thanks.

---

### Author Response · Authors · 2023-11-18

Thank you for the thoughtful comments and questions.
We are especially glad that you think we have made
interesting progresses on a difficult problem.

Some of you think our presentation  can be further improved.
For example, Reviewer SvT9 suggest that we provide some insight on the proofs, and
Reviewer dPXa suggests that we further explain the connection between NIEP, NMF, and
network modeling.
 We agree and we have made the following major changes.

- In the first part of Section 2 (marked in blue), we have revised our paper and explain a key idea for our proofs.
- In Section 2.2 (marked in blue), we have revised our paper and explain the key idea behind the
DFT construction.
- In Section L of the supplement, we have add a relatively long section explaining the
connection between NMF and network modeling. Note that the whole Section 1.1 is for explaining the  connection between NIEP and NMF.
- Especially, in Section L of the supplement, we discuss how our results impact network modeling by consider $5$ well-known real networks: Weblog, karate, dolphin, polbooks,  and UKFaculty.

We have also other places of our paper (e.g.,  Section 1)  to improve our literature review and our result summary.
We hope these changes have adequately addressed your raised issues.

---

### Meta-Review · Area_Chair_nEUB · 2023-12-06

**Metareview:**

The paper studies a special case of the non-negative inverse eigenvalue problem, which is motivated by applications to non-negative matrix factorization and network modeling. The reviewers appreciated the main contribution of the paper, but there were significant concerns regarding the presentation of the paper and its accessibility to a general ML audience. Overall, the paper had only limited support. We encourage the authors to further revise the paper based on the reviewer feedback.

**Justification For Why Not Higher Score:**

Although the paper studies a problem that is relevant to the community, the paper's presentation does not seem to be suitable for a general ML audience. Additionally, the motivation and applicability of this work is unclear.

**Justification For Why Not Lower Score:**

N/A

---

### Decision · Program_Chairs · 2024-01-16

Reject